# Triplication of the interferon receptor locus contributes to hallmarks of Down syndrome in a mouse model

Katherine A. Waugh[1,2], Ross Minter [1], Jessica Baxter[1], Congwu Chi[3,4,5], Matthew D. Galbraith[1,2], Kathryn D. Tuttle[1], Neetha P. Eduthan[1], Kohl T. Kinning [1], Zdenek Andrysik [1,2,15], Paula Araya[1,15], Hannah Dougherty[1,15], Lauren N. Dunn [1,6,15], Michael Ludwig[1,2,15], Kyndal A. Schade[1,15], Dayna Tracy[1,15], Keith P. Smith[1], Ross E. Granrath[1], Nicolas Busquet[7,8], Santosh Khanal[1,2], Ryan D. Anderson[9], Liza L. Cox[9], Belinda Enriquez Estrada[1], Angela L. Rachubinski [1,10], Hannah R. Lyford[1], Eleanor C. Britton[1], Katherine A. Fantauzzo [11], David J. Orlicky[12], Jennifer L. Matsuda[13], Kunhua Song[1,3,4,5], Timothy C. Cox [9,14], Kelly D. Sullivan [1,6,16] ✉ & Joaquin M. Espinosa [1,2,16] ✉

Down syndrome (DS), the genetic condition caused by trisomy 21, is characterized by variable cognitive impairment, immune dysregulation, dysmorphogenesis and increased prevalence of diverse co-occurring conditions. The mechanisms by which trisomy 21 causes these effects remain largely unknown. We demonstrate that triplication of the interferon receptor (*IFNR*) gene cluster on chromosome 21 is necessary for multiple phenotypes in a mouse model of DS. Whole-blood transcriptome analysis demonstrated that *IFNR* overexpression associates with chronic interferon hyperactivity and inflammation in people with DS. To define the contribution of this locus to DS phenotypes, we used genome editing to correct its copy number in a mouse model of DS, which normalized antiviral responses, prevented heart malformations, ameliorated developmental delays, improved cognition and attenuated craniofacial anomalies. Triplication of the *Ifnr* locus modulates hallmarks of DS in mice, suggesting that trisomy 21 elicits an interferonopathy potentially amenable to therapeutic intervention.

Trisomy of human chromosome 21 (trisomy 21) occurs 1 in ~700 live births, causing Down syndrome (DS)[1,2]. People with DS experience variable developmental delays, cognitive impairments and craniofacial abnormalities, as well as higher rates of congenital heart defects (CHD), autoimmune disorders and diverse neurological conditions, including Alzheimer's disease, while also displaying lower rates of solid malignancies and hypertension[3–5]. Despite many research efforts, the mechanisms driving these hallmarks of DS are largely unknown.

Interferon (IFN) signaling is hyperactive in DS[6]. Upon receptor binding, IFN ligands induce the Janus kinase/signal transducer and activator of transcription (JAK/STAT) signaling pathway and downstream transcriptional programs mediating restriction of viral replication, decreased cell proliferation, apoptosis, metabolic reprogramming and immune activation[7]. Notably, four of six IFN receptor genes (*IFNRs*) reside on human chromosome 21 (HSA21), which are as follows: *IFNAR1/IFNAR2, IFNGR2* and *IL10RB*, which recognize type I, II and III IFNs, respectively[6,8]. Cells with trisomy 21 display hypersensitivity

to IFN stimulation[6,9–11], which is rescued in vitro by reducing *IFNR* copy number[10]. Furthermore, multiple constitutive trisomies have been shown to elevate IFN signaling through the accumulation of cytosolic double-stranded DNA and activation of the cyclic guanosine monophosphate–adenosine monophosphate (GMP–AMP) synthase–stimulator of *IFN* gene (cGAS-STING) pathway[12]. Notably, mutations leading to overactive IFN signaling cause interferonopathies, a group of monogenic disorders that share key traits with DS[13,14]. Therefore, elucidating the mechanism driving IFN hyperactivity in DS and its contribution to various phenotypes could identify targeted therapeutics for this population.

Here we used transcriptome and cytokine analyses in a large cohort of individuals with DS to define associations between overexpression of HSA21 genes and inflammatory markers, which revealed that few triplicated genes, including the four *IFNRs*, associate with IFN hyperactivity and inflammation. We then employed genome editing to correct the dosage of the *Ifnr* locus in a mouse model of DS, which revealed that the *Ifnr* locus contributes to multiple key phenotypes in mice, with potential therapeutic implications for the management of this condition.

## Results

### Inflammatory markers correlate with *IFNR* expression

Using matched whole-blood transcriptome and plasma immune marker data from 304 individuals with Down syndrome (163 male and 141 female) versus 96 euploid controls (44 male and 52 female), we completed a correlation study between overexpression of HSA21 genes and immune markers across the lifespan (Methods, Extended Data Fig. 1a,b and Supplementary Table 1). Expectedly, the transcriptome analysis detected upregulation of most genes encoded on HSA21, with a mean fold-change of ~1.5 (Fig. 1a and Supplementary Table 2). Nevertheless, there was a wide range of expression of the triplicated genes among individuals with and without DS (for example, *IFNAR1* and *DYRK1A*; Fig. 1b). This analysis also identified thousands of differentially expressed genes (DEGs) encoded elsewhere in the genome (for example, *MYD88* and *COX5A*; Fig. 1a,b). Gene set enrichment analysis (GSEA) extended previous observations demonstrating activation of the IFN transcriptional response in DS[6]. Among the top 10 gene sets substantially enriched in trisomy 21, seven correspond to IFN signaling and inflammatory pathways (Fig. 1c and Supplementary Table 2). To define which HSA21 genes were associated with signaling pathways dysregulated in DS, we correlated their mRNA expression with the rest of the transcriptome via Spearman analysis using only trisomy 21 samples and analyzed the matrices of ranked rho ($\rho$) values by GSEA (Extended Data Fig. 1c). While most HSA21 genes had negative correlations with gene signatures of inflammation, a few had consistent significant positive correlations, including the four *IFNRs* and IFN-stimulated genes (ISGs) encoded on HSA21, such as *MX1* and *MX2* (Extended Data Fig. 1c,d).

For example, whereas expression of *IFNAR1* positively correlated with multiple inflammatory pathways, *DYRK1A* expression did not (Fig. 1d,e). Multiple ISGs not encoded on HSA21 (for example, *MYD88*, *STAT3* and *TRIM25)* showed strong positive correlations with *IFNRs* but not with most HSA21 genes (Fig. 1c–f and Extended Data Fig. 1c). In contrast, genes in the oxidative phosphorylation signature elevated in DS (for example, *COX5A*) were negatively correlated with *IFNR* expression, correlating instead with the expression of other HSA21 genes, such as *ATP5PO* and *SOD1* (Fig. 1c,f and Extended Data Fig. 1c–e). Thus, not all HSA21 genes are overexpressed in a concerted fashion in DS, with different individuals overexpressing different patterns of HSA21 genes, which in turn associate with the dysregulation of different pathways. For example, among HSA21 genes, *IFNAR1* is co-expressed with *IFNGR2* but anticorrelated with *ATP5PO*, whereas *DYRK1A* is co-expressed with *ZBTB21* but anticorrelated with *CSTB* (Extended Data Fig. 1e).

We then defined correlations between circulating protein levels of the inflammatory marker C-reactive protein (CRP) and the pro-inflammatory cytokine interleukin 6 (IL6) versus expression of HSA21 genes among people with trisomy 21. Expression of only a few HSA21 genes correlated positively with CRP and IL6, including the four *IFNRs* (Fig. 1c,g, Extended Data Fig. 1c,f and Supplementary Table 3). Whereas *IFNAR1* expression correlates positively with levels of CRP and IL6, *DYRK1A* expression is negatively correlated with both immune markers (Fig. 1h and Extended Data Fig. 1f).

Altogether, these results indicate that the inflammatory state observed in DS is associated with overexpression of select HSA21 genes, including all four *IFNRs*, and is unlikely to be solely a general effect of the aneuploidy.

### The *Ifnr* locus contributes to global transcriptome changes

The B6.129S7-Dp(16Lipi-Zbtb21)1Yey/J mouse model of DS, herein 'Dp16', carries a segmental duplication of mouse chromosome 16 (MMU16) causing triplication of ~120 protein-coding genes orthologous to those on HSA21, including the *Ifnr* cluster[15,16]. Dp16 mice display key phenotypes of DS including hyperactive IFN signaling, a dysregulated antiviral response, increased prevalence of heart defects, developmental delays, cognitive impairments and craniofacial anomalies[6,16–21].

To test if the *Ifnr* locus contributes to DS phenotypes, we used genome editing technology to delete one copy of the entire gene cluster. Given that all four *Ifnrs* employ JAK/STAT signaling, creating the potential for genetic redundancy, we designed a strategy to delete the 192 kb genomic segment encoding all four *Ifnrs* in wild-type (WT) mice (Methods, Fig. 2a and Supplementary Table 4). Heterozygous knockout was confirmed in potential founders and whole genome sequencing (WGS) confirmed the heterozygous deletion without other substantial genomic alterations (Fig. 2a and Extended Data Fig. 2a–d). Heterozygous progeny of this strain (WT$^{1 \times Ifnrs}$) was then intercrossed with Dp16 to correct *Ifnr* copy number from three to two in a fraction

**Fig. 1 | Overexpression of *IFNRs* associates with inflammatory signatures in DS. a**, Manhattan plots of human chromosomes 3 and 21 (HSA3 and HSA21) displaying results of whole-blood transcriptome analysis for individuals with trisomy 21 (T21, *n* = 304, 163 male and 141 female) versus euploid controls (D21, *n* = 96, 44 male and 52 female). Red points mark DEGs identified by DESeq2. **b**, Sina plots displaying results for representative DEGs. Boxes represent interquartile ranges and medians, with notches approximating 95% CIs; *q* values determined by DESeq2 with Benjamini–Hochberg correction. **c**, Heatmaps displaying top left–NES from GSEA of transcriptome changes in individuals with DS. Only the top ten positively enriched pathways by NES are shown; top right–NES from GSEA of transcriptome signatures associated with expression of HSA21 genes surrounding the *IFNR* cluster (red). Spearman correlations were defined for HSA21-encoded mRNAs versus all other mRNAs using only trisomy 21 samples and $\rho$ values as the GSEA ranking metric; middle–Spearman correlations between mRNAs encoded on HSA21 versus mRNAs for indicated DEGs encoded elsewhere in the genome among individuals with DS; bottom–Spearman correlations between mRNAs encoded on HSA21 and plasma levels of CRP and IL6 in individuals with DS. The asterisks indicate *q* < 0.1 from either GSEA or Spearman correlations with permutation test and Benjamini–Hochberg correction. **d,e**, Volcano plots of Spearman correlations for (**d**) *IFNAR1* or (**e**) *DYRK1A* mRNA abundance versus all other mRNAs among individuals with DS. Heatmaps display NES from GSEA of ranked Spearman $\rho$ values. **f**, Scatter plots displaying relationships between expression of *IFNAR1* versus indicated DEGs among individuals with DS. **g**, Volcano plot of Spearman correlations for CRP levels versus mRNAs encoded on HSA21 among individuals with DS (*n* = 249, 137 male and 112 female). **h**, Scatter plots displaying relationships between CRP and two example mRNAs encoded on HSA21. In **f** and **h**, individual points are colored by local density and blue lines represent linear regression fits with 95% CIs in gray; $\rho$ and *q* values from Spearman correlation with permutation test and Benjamini–Hochberg correction. RPKM, reads per kilobase per million; NES, normalized enrichment scores; CI, confidence interval.

of Dp16 offspring (Dp16$^{2xIfnrs}$; Fig. 2b). WT$^{1xIfnrs}$ mice were viable and fertile, with no obvious phenotypes, but additional characterization will be valuable to define the impacts of monosomy (or nullisomy) of the *Ifnr* locus.

To characterize the impacts of *Ifnr* copy number on gene expression programs dysregulated in Dp16, we completed transcriptome analysis across different tissues relevant to DS phenotypes, including adult mesenteric lymph nodes, embryonic and adult heart tissues,

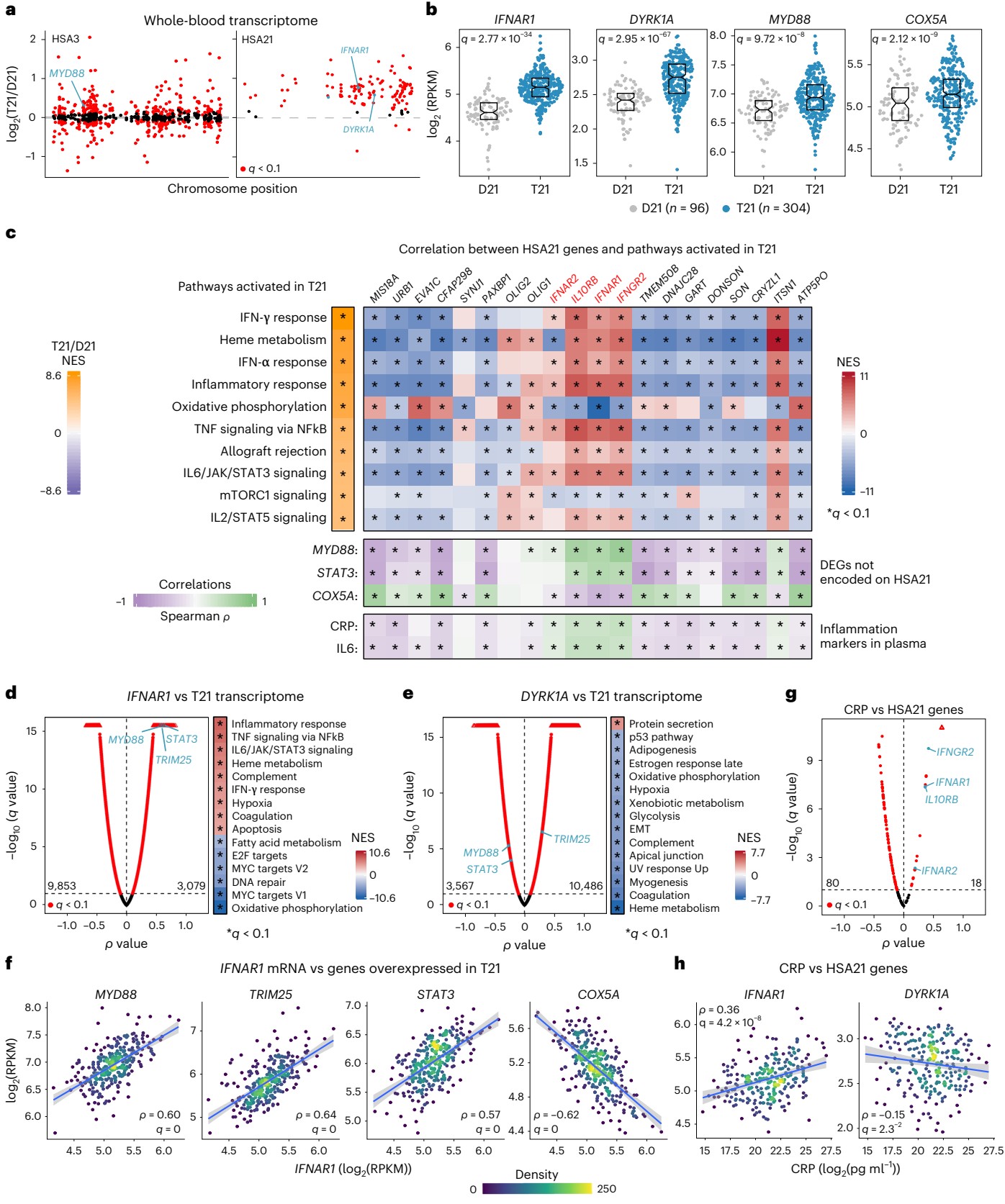

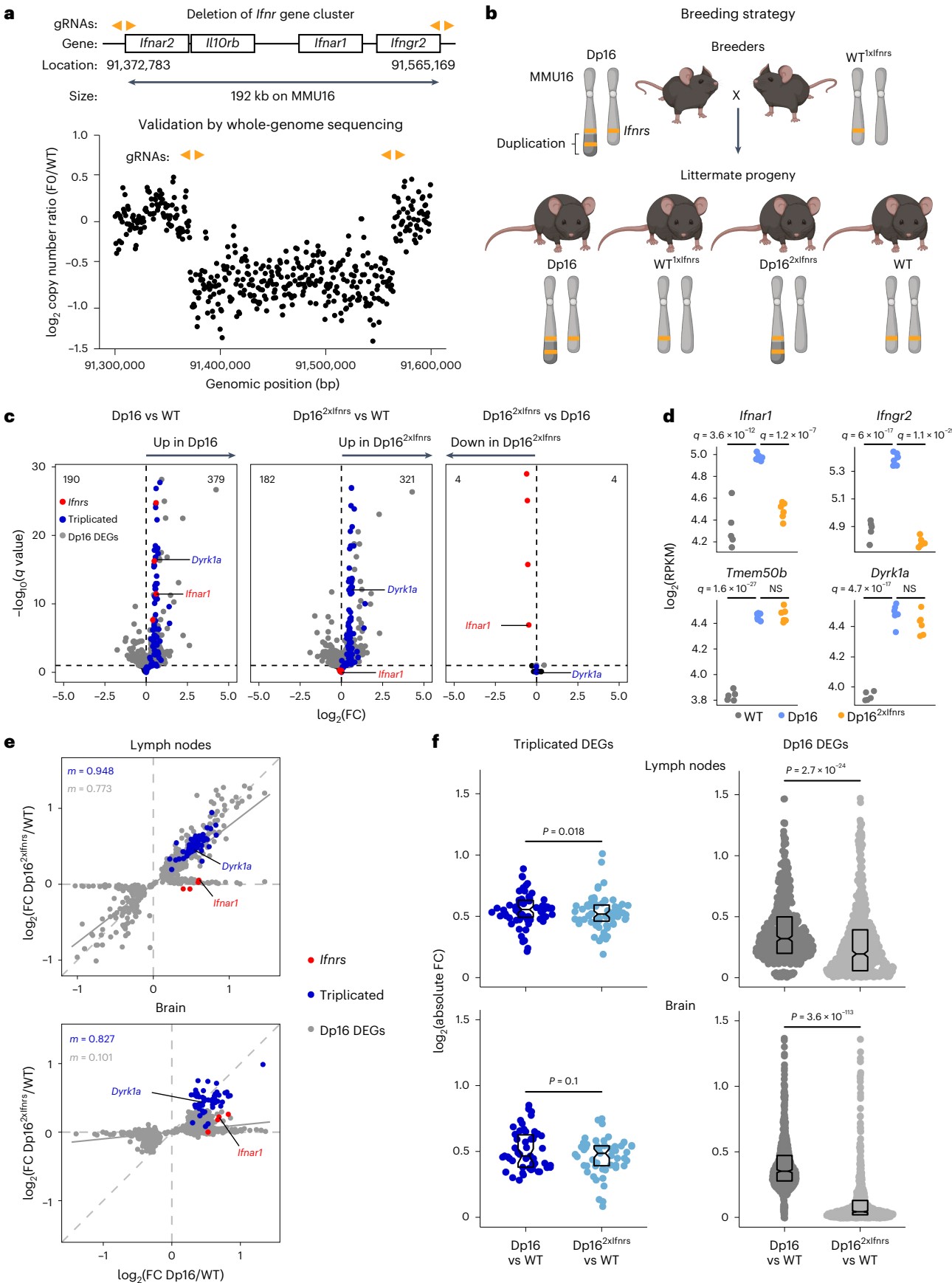

**Fig. 2 | Triplication of the *Ifnr* locus contributes to global dysregulation of gene expression in a mouse model of DS. a**, Top−diagram indicating genomic locations of the mouse *Ifnr* gene cluster on MMU16 and gRNAs (orange arrowheads) employed for genome editing using CRISPR−Cas9 technology. Positions are indicated in base pairs (bp) for the GRCm38 assembly of the *M. musculus* genome. Bottom−copy number variant analysis from WGS for a candidate founder (F0) bearing a deletion relative to a WT control. **b**, Breeding strategy to correct copy number of the *Ifnr* gene cluster in the Dp16 mouse model of DS. **c**, Volcano plots showing transcriptome analysis of mesenteric lymph nodes obtained from naïve adult WT (*n* = 5, 2 male and 3 female), Dp16 (*n* = 6, 3 male and 3 female), and Dp16[2xIfnrs] (*n* = 6, 3 male and 3 female), highlighting expression of *Ifnrs* (red), other MMU16 genes triplicated in Dp16 (blue), with DEGs encoded elsewhere in the genome in gray, and all other genes in black.

**d**, Expression levels in RPKM for representative MMU16-encoded mRNAs from mesenteric lymph nodes. *q* Values defined by DEseq2 after Benjamini−Hochberg correction. **e**, Scatter plots comparing mRNA fold-changes for Dp16 DEGs in mesenteric lymph nodes (top, sample sizes as in **c**) and brain (bottom, WT (*n* = 6, 2 male and 4 female), Dp16 (*n* = 5, 2 male and 3 female), and Dp16[2xIfnrs] (*n* = 7, 4 male and 3 female)) for Dp16/WT and Dp16[2xIfnrs]/WT, with *Ifnrs* highlighted in red, Dp16 triplicated genes in blue, nontriplicated Dp16 DEGs in gray, and slope (*m*) colored accordingly. Solid gray lines represent linear fits for the nontriplicated Dp16 DEGs. **f**, Sina plots displaying absolute fold-changes for DEGs triplicated in Dp16 (blue, excluding the four *Ifnrs*), and nontriplicated Dp16 DEGs (gray) across the genome for Dp16 versus WT or Dp16[2xIfnrs], comparisons, with *P* values for two-sided paired Wilcoxon rank tests, boxes representing interquartile ranges and medians, and notches approximating 95% CIs. gRNA, guide RNA.

embryonic neural crest-derived facial mesenchyme and adult whole brain tissue. These analyses consistently highlighted overexpression of the triplicated genes in Dp16 and rescue of *Ifnr* overexpression in Dp16[2xIfnrs], while preserving overexpression of other triplicated genes (Fig. 2c,d and Extended Data Fig. 2e,f). For example, in the lymph nodes, the genomic deletion clearly rescued overexpression of *Ifnar1* and *Ifngr2* without affecting the expression of the nearby triplicated genes *Tmem50b* or *Dyrk1a* (Fig. 2d). Notably, these analyses identified hundreds of DEGs in Dp16 tissues across the genome (Fig. 2c, Extended Data Fig. 2e and Supplementary Tables 5−10). Comparison of the fold-changes for these DEGs in Dp16/WT versus Dp16[2xIfnrs]/WT revealed significant attenuation of gene expression changes in Dp16[2xIfnrs] in every tissue examined, albeit to variable degrees (Fig. 2e,f and Extended Data Fig. 2e). As described later, this dampening of gene expression changes in Dp16[2xIfnrs] affects specific signaling pathways in each tissue. In contrast, triplicated MMU16 genes are largely insensitive to *Ifnr* copy number (Extended Data Fig. 2e).

Altogether, these results demonstrate that *Ifnr* triplication contributes to dysregulated gene expression programs throughout the genome and that Dp16[2xIfnrs] mice provide a model to define the contribution of the *Ifnr* locus to DS phenotypes.

### Triplication of the *Ifnr* locus exacerbates immune responses

We previously demonstrated that immune cells from Dp16 are hypersensitive to IFN stimulation[21]. Furthermore, upon chronic exposure to the viral mimetic polyinosinic-polycytidylic acid (poly(I:C)), Dp16 mice experience exacerbated weight loss and death, which is rescued by JAK1 inhibition[21]. We thus investigated the role of the *Ifnr* locus on these phenotypes. Correction of *Ifnr* dosage rescued protein overexpression for all four *Ifnrs* (Fig. 3a and Extended Data Fig. 3a−e). When stimulated ex vivo with IFN-α or IFN-γ, white blood cells (WBCs) from Dp16 show substantially elevated levels of phospho-STAT1, but this

phenotype is rescued in Dp16[2xIfnrs], which showed even lower levels than WT (Fig. 3b and Extended Data Fig. 3f).

Transcriptome analysis of mesenteric lymph nodes identified several gene sets substantially dysregulated in Dp16 and attenuated in Dp16[2xIfnrs], including signatures of increased cell proliferation (E2F targets and G2/M Checkpoint), increased IL2/STAT5 signaling and decreased oxidative phosphorylation (Fig. 3c,d). Although changes in individual genes are often modest across the three genotypes, the results nonetheless indicate that *Ifnr* overexpression is accompanied by abnormal gene expression in the immune system of Dp16, even in the absence of immune stimulation. Notably, key inflammatory signatures are still elevated in Dp16[2xIfnrs] relative to WT mice, such as IL2/STAT5 signaling and IL6/JAK/STAT3 signaling (Fig. 3c), indicating that much immune dysregulation occurs without triplication of the *Ifnr* locus in this setting, suggesting roles for other triplicated genes.

To investigate the effects on the organismal antiviral response, we challenged mice with chronic poly(I:C) treatment. Dp16 mice lost substantially more weight than WT littermates and had to be removed much earlier at the humane endpoint of 15% weight loss, but Dp16[2xIfnrs] did not differ from controls (Fig. 3e,f). Thus, although Dp16[2xIfnrs] retain many global gene expression changes in the immune compartment relative to Dp16, their lethal inflammatory response is clearly normalized. Analysis of cytokine induction revealed overproduction of TNF in Dp16 relative to controls, but this was not observed in Dp16[2xIfnrs] (Fig. 3g and Extended Data Fig. 3g). TNF can mediate inflammation-driven cachexia[22], and its levels correlated with weight loss in our paradigm (Fig. 3h).

Altogether, these results indicate that triplication of the *Ifnr* locus contributes to select gene expression changes in the immune system, mediating hypersensitivity to IFN stimulation and a dysregulated antiviral response in vivo.

**Fig. 3 | Triplication of the *Ifnr* locus drives increased IFNR expression and exacerbated antiviral responses in a mouse model of DS. a**, gMFI relative to WT mice, as measured by flow cytometry, for IFNR proteins on CD45[+] WBCs from heterozygous *Ifnr* knockout mice (WT[1xIfnrs], *n* = 5, 3 male and 2 female), WT (*n* = 8 for IFNAR1, 2 male and 6 female, *n* = 16 for IFNGR2, 10 male and 6 female), Dp16 (*n* = 7, 5 male and 2 female) and Dp16[2xIfnrs] (*n* = 6 for IFNAR1, 4 male and 2 female; *n* = 8 for IFNGR2, 6 male and 2 female). Horizontal dashes indicate mean values. Significance determined by two-sided Mann−Whitney test. **b**, gMFI relative to WT, as measured by flow cytometry, for phosphorylated STAT1 in WBCs at baseline or after 30 min stimulation with IFN-α or IFN-γ. Number of animals−unstimulated WT (*n* = 25, 16 male and 9 female), Dp16 (*n* = 15, 7 male and 8 female) and Dp16[2xIfnrs] (*n* = 8, 8 male); +IFN-α WT (*n* = 23, 14 male and 9 female), Dp16 (*n* = 12, 3 male and 9 female) and Dp16[2xIfnrs] (*n* = 9, 9 male); +IFN-γ WT (*n* = 25, 16 male and 9 female), Dp16 (*n* = 12, 12 male) and Dp16[2xIfnrs] (*n* = 7, 7 male). Significance determined by two-sided Mann−Whitney test. **c**, Heatmap displaying NES from GSEA with Hallmark gene sets of transcriptome fold-changes for the indicated comparisons in mesenteric lymph nodes, sorted by NES for Dp16/WT; *q* < 0.1 defined by GSEA with Benjamini−Hochberg correction.

**d**, Heatmaps (top) representing median RPKM expression *z* scores per genotype and sina plots (bottom) for example genes from the indicated gene sets; *q* values determined by DESeq2 with significance defined as *q* < 0.1 after Benjamini−Hochberg correction. **e**, Kaplan−Meier plot comparing survival across genotypes during chronic stimulation with the TLR3 agonist poly(I:C); significance determined by Mantel−Cox log-rank test. **f**, Percentage weight loss normalized to total number of poly(I:C) doses. For **e** and **f**, WT sham (*n* = 9, 2 male and 7 female), WT poly(I:C) (*n* = 13, 6 male and 7 female), Dp16 poly(I:C) (*n* = 9, 6 male and 3 female), Dp16[2xIfnrs] poly(I:C) (*n* = 13, 6 male and 7 female). **g**, TNF protein in serum on day 3 of poly(I:C) exposure, WT sham (*n* = 6, 2 male, 4 female), WT poly(I:C) (*n* = 7, 3 male, 4 female), Dp16 poly(I:C) (*n* = 6, 5 male, 1 female) and Dp16[2xIfnrs] poly(I:C) (*n* = 6, 3 male and 3 female). In **f** and **g**, horizontal dashes indicate group means and *P* values for pairwise comparisons were determined by two-sided Mann−Whitney test, with significance set at *P* < 0.05. **h**, Scatter plot comparing TNF concentration and percent weight loss on day 3 of poly(I:C) exposure (*n* = 25 animals, numbers by genotype and sex as in **g**), with simple linear regression fit line. ρ and *P* values from Spearman correlation with permutation test. gMFI, geometric mean fluorescent intensities.

## Triplication of the *Ifnr* locus contributes to heart defects

Around half of newborns with DS are born with CHDs[3]. To test if the *Ifnr* locus contributes to this phenotype, we evaluated heart malformations in WT, Dp16 and Dp16^2xIfnrs embryos through histological evaluation at embryonic day (E) 15.5 (Fig. 4a–c and Extended Data Fig. 4a). In agreement with previous reports[16,23,24], Dp16 mice displayed elevated frequency of atrial septal defects (ASD) and/or ventricular septal defects (VSDs), but this phenotype was corrected in Dp16^2xIfnrs (Fig. 4d). Western

blot analysis showed elevated phospho-STAT1 in the developing heart tissue of Dp16, but not in Dp16^2xIfnrs (Extended Data Fig. 4b).

To investigate potential underlying mechanisms, we completed transcriptome analysis of heart tissue at E12.5, E18.5 and adulthood. At all three time points, Dp16 shows overexpression of most triplicated genes, along with global dysregulation of key signaling pathways, with significant attenuation of genome-wide changes in Dp16^2xIfnrs, most prominently at E12.5 (Fig. 4e and Extended Data Fig. 2e). Pathway

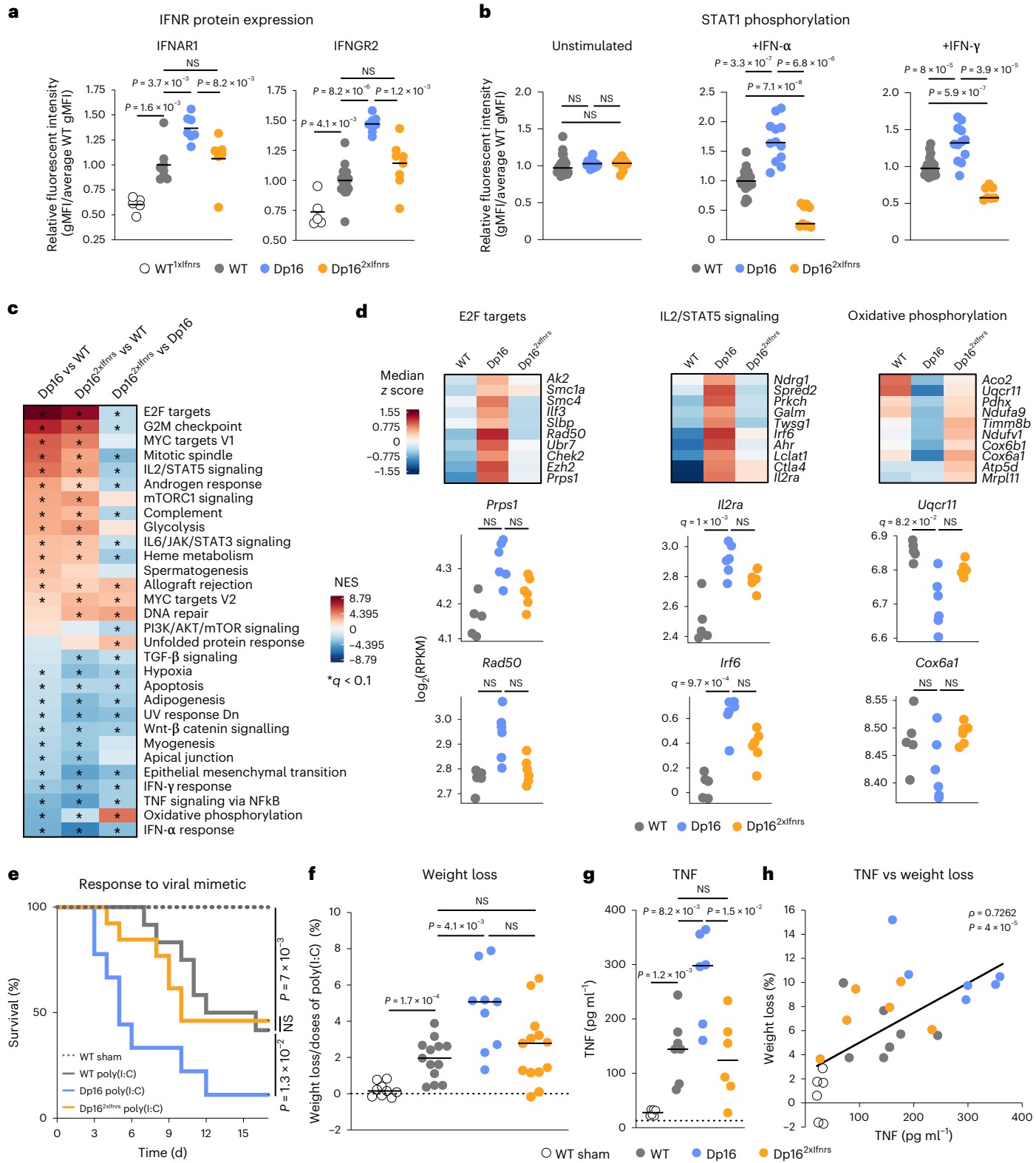

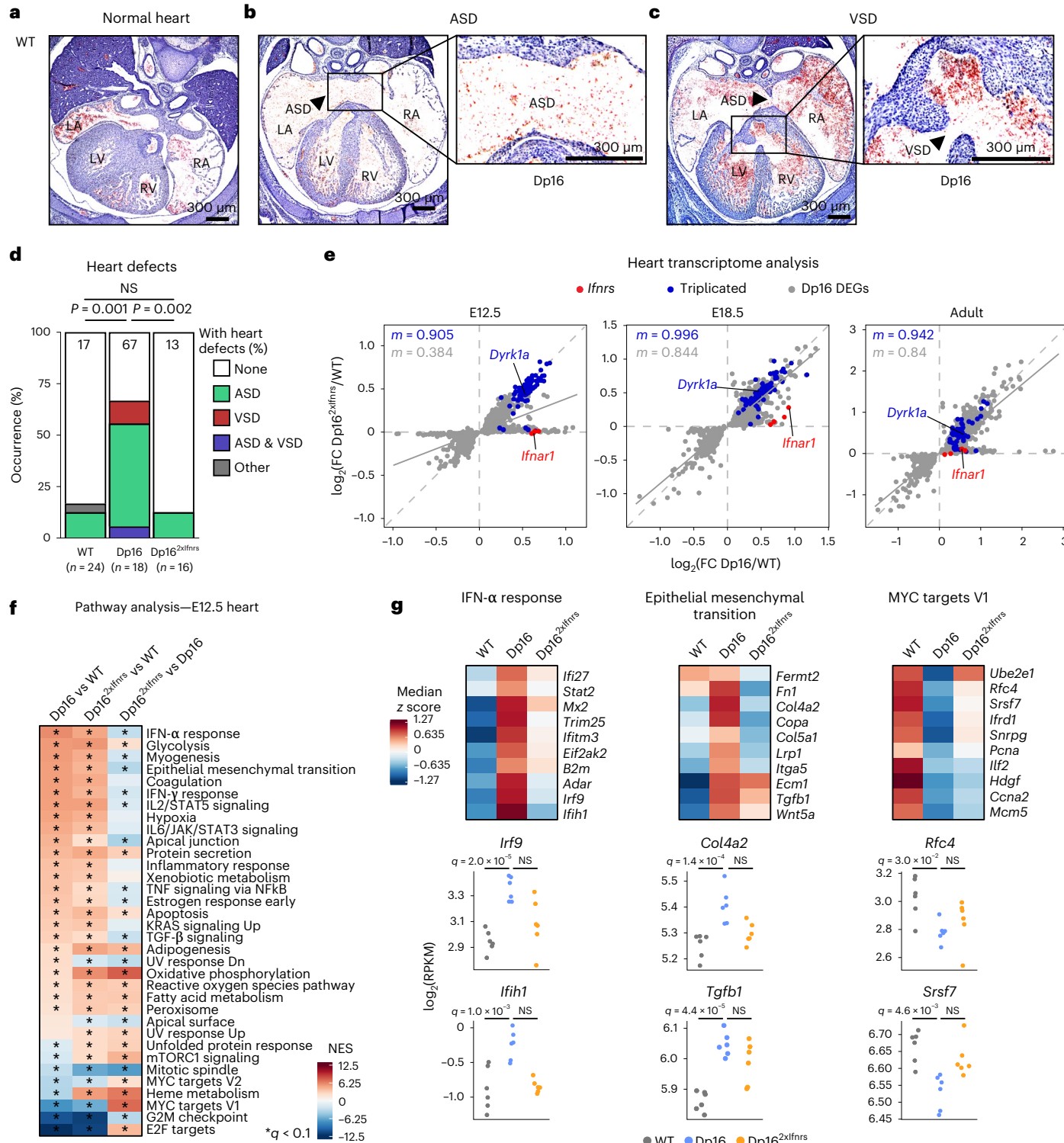

**Fig. 4 | Triplication of the *Ifnr* locus is necessary for increased incidence of heart malformations in a mouse model of DS. a–c**, Representative images of hematoxylin and eosin stained serial sections through entire mouse hearts at embryonic day (E)15.5, showing (**a**) normal septation of the four heart chambers in a WT embryo, (**b**) an ASD in a Dp16 embryo and (**c**) a VSD in a Dp16 embryo. A total of 58 formalin-fixed paraffin-embedded embryos were processed and analyzed across four independent batch experiments. **d**, Heart malformation frequencies at E15.5. Other−outflow tract anomaly. *P* values were calculated for differences in CHD occurrence for pairwise comparisons between genotypes using two-sided Fisher's exact test. Number of animals−WT (*n* = 24, 12 male and 12 female), Dp16 (*n* = 18; 9 male, 7 female and 2 undetermined sex) and Dp16^2xIfnrs (*n* = 16, 4 male and 12 female). **e**, Scatter plots comparing mRNA fold-changes for Dp16 DEGs in heart tissue from

mice at the indicated ages for Dp16/WT and Dp16^2xIfnrs/WT, with *Ifnrs* highlighted in red, Dp16 triplicated genes in blue, nontriplicated Dp16 DEGs in gray and slope (*m*) colored accordingly; solid gray lines represent linear fits for the nontriplicated Dp16 DEGs. Number of animals−E12.5 hearts WT (*n* = 6, 2 male and 4 female), Dp16 (*n* = 6, 3 male and 3 female) and Dp16^2xIfnrs (*n* = 6, 5 male and 1 female); E18.5 hearts WT (*n* = 6, 4 male and 2 female), Dp16 (*n* = 6, 3 male and 3 female) and adult hearts WT (*n* = 5, 2 male and 3 female), Dp16 (*n* = 6, 3 male and 3 female) and Dp16^2xIfnrs (*n* = 5, 3 male and 2 female). **f**, Heatmap of GSEA for transcriptome changes in E12.5 hearts, sorted by NES for Dp16 versus WT (sample sizes described in **e**); asterisks indicate *q* < 0.1 from GSEA. **g**, Heatmaps (top) representing median RPKM expression *z* scores per genotype and sina plots (bottom) for example genes from the indicated gene sets; *q* values determined by DESeq2. R, right; L, left; A, atrium; V, ventricle.

analysis identified common and unique gene signatures dysregulated in Dp16 that are partially attenuated in Dp16[2xIfnrs] at all three time points (Fig. 4f and Extended Data Fig. 4c). Consistently with this, Dp16 heart tissues show increased IFN-α and IFN-γ signaling concurrent with the elevation of diverse inflammatory pathways, elevated expression of genes involved in epithelial-mesenchymal transition (EMT) and decreased expression of genes associated with cell proliferation (for example, MYC targets and E2F targets; Fig. 4f and Extended Data Fig. 4c–e). In Dp16[2xIfnrs], some of these gene expression changes are attenuated, with variation across time points. At E12.5, a critical time point for heart septation, Dp16[2xIfnrs] show lesser dysregulation of ISGs, EMT genes and MYC target genes (Fig. 4f,g). Thus, the decreased incidence of CHD observed in Dp16[2xIfnrs] is accompanied by the partial rescue of the global transcriptome changes, wherein only some of the gene expression changes observed are due to *Ifnr* triplication.

Altogether, these results indicate that triplication of the *Ifnr* locus elicits a signaling cascade in the developing heart involving elevated JAK/STAT signaling, dysregulation of EMT processes and decreased cell proliferation, which may explain its contribution to the appearance of CHDs.

## The *Ifnr* locus affects development and cognition

Children with DS and Dp16 neonates exhibit variable delays in achieving developmental milestones[3,20]. Relative to WT controls, Dp16 neonates show a reduced chance of success in achieving the surface righting reflex as well as ear twitch and auditory startle sensitivities on any given day, but no differences in eye-opening, with Dp16 females showing the most pronounced differences (Fig. 5a and Extended Data Fig. 5a,b). Notably, Dp16[2xIfnrs] show the rescue of all three Dp16 developmental delays (Fig. 5a and Extended Data Fig. 5a,b). Neonate length and weight were substantially lower in both Dp16 and Dp16[2xIfnrs], indicating that the rescue of developmental delays was not accompanied by improved growth (Extended Data Fig. 5c).

We evaluated cognitive deficits in adult mice using contextual fear conditioning (CFC) and Morris water maze (MWM)[25,26]. During conditioning in the CFC test for associative learning and memory, mice were presented with two foot shocks. Upon the second shock, Dp16 displayed substantially decreased freezing responses (Fig. 5b). When reintroduced to the shock context on day 2, both WT and Dp16 froze at higher baseline rates relative to the beginning of day 1, yet WT mice froze at a substantially higher rate than Dp16. Throughout the experiment, both on day 1 and day 2, Dp16[2xIfnrs] displayed a significant rescue of these phenotypes (Fig. 5b).

Upon examination of spatial learning and memory via MWM[26], adult mice of all genotypes were equally capable of learning to escape the maze during the acquisition learning phase (Extended Data Fig. 5d–f). However, Dp16 but not Dp16[2xIfnrs] males swam substantially closer to the periphery when introduced to the maze (Extended Data Fig. 5g). Although this behavior is associated with the hindrance of

learning[27], such thigmotaxis (that is, tendency to stay toward the edge of a new environment) was moderate in Dp16 males, and they still learned to escape the maze as quickly as the other genotypes during the acquisition phase (Extended Data Fig. 5f). Immediately upon change of platform location in the reversal phase, both Dp16 and Dp16[2xIfnrs] presented with deficits in memory extinction (Extended Data Fig. 5f, block 7). However, only Dp16 males exhibited impaired relearning of platform location (Extended Data Fig. 5f, blocks 8–9). All cohorts showed improved performance over time in the reversal phase (Extended Data Fig. 5f). These subtle yet significant differences by genotype are in line with previous studies using MWM to investigate Dp16 deficits in memory extinction and relearning[16,18–20]. To test for differences in allocentric memory, we evaluated swim path efficiency, which showed no difference in acquisition learning by genotype, revealing instead a significant deficit in memory extinction and relearning in Dp16, more pronounced in males, with significant rescue in Dp16[2xIfnrs] (Fig. 5c,d). Furthermore, Dp16 but not Dp16[2xIfnrs] demonstrated reduced target quadrant occupancy during the reversal probe trial relative to controls (Fig. 5e). Lastly, impaired Dp16 motor coordination measured by the rotarod performance test[28] was not rescued in Dp16[2xIfnrs] (Extended Data Fig. 5h).

Transcriptome analysis of adult brain tissue confirmed overexpression of triplicated genes in Dp16 and Dp16[2xIfnrs], along with dysregulation of hundreds of DEGs that were substantially attenuated in Dp16[2xIfnrs] (Fig. 2e,f). Pathway analysis identified multiple gene signatures important for brain function that are strongly dysregulated in Dp16 but less so in Dp16[2xIfnrs] (Fig. 5f,g and Extended Data Fig. 5i). Salient examples include genes involved in synaptogenesis, SNAP receptor (SNARE) signaling and dopamine signaling. As observed in other tissues, the effects of correcting *Ifnr* locus dosage on gene expression changes are partial and selective for specific signaling pathways, once again revealing that clear phenotypic differences can be observed without full correction of underlying transcriptome changes.

Altogether, these results indicate that *Ifnr* gene dosage affects developmental milestones and key domains of cognitive function, including associative learning and memory as well as spatial memory.

## The *Ifnr* locus contributes to craniofacial anomalies

Craniofacial abnormalities are a hallmark of DS, including brachycephaly, maxillary deficiency and smaller cranial base[29]. Given that Dp16 mice reproduce many aspects of the distinct craniofacial morphology of DS[17], we evaluated the impact of *Ifnr* locus dosage on Dp16 skull size and shape. We employed Euclidean distance matrix analysis (EDMA) using 30 craniofacial and mandibular landmarks[30] collected from micro-computed tomography (μCT) scans (Fig. 6a,b, Extended Data Fig. 6a,b and Supplementary Table 11). Analysis of pairwise distances between landmarks normalized to overall skull size revealed that 58% (149/259) of all interlandmark distances differed between Dp16 and WT controls, consistent with prior studies[17] (Fig. 6b and Extended Data

**Fig. 5 | Triplication of the *Ifnr* locus promotes developmental delays and cognitive deficits in a mouse model of DS. a**, Odds ratio plots for developmental milestone achievement in neonates as assessed by mixed effects Cox regression for the indicated pairwise comparisons between Dp16, Dp16[2xIfnrs] and WT animals, with adjustment for the covariates sex (fixed) and litter (random). Square points represent 'success' ratios with size proportional to $-\log_{10}(q)$ and error bars corresponding to 95% CIs; red indicates $q < 0.1$ after Benjamini–Hochberg correction; vertical dashed lines indicate odds ratio of 1. Numbers of animals assessed for each milestone are shown in the table at right. **b**, Freezing behavior during CFC of adult WT ($n = 33$, 13 male and 20 female), Dp16 ($n = 17$, 8 male and 9 female) and Dp16[2xIfnrs] ($n = 23$, 11 male and 12 female). Data are represented as means ± s.e.m., with significance determined by two-way repeated measures ANOVA and Tukey's HSD test; asterisks indicate $P < 0.05$ and are colored by comparison—Dp16 versus WT (blue), Dp16[2xIfnrs] versus WT (orange) and Dp16 versus Dp16[2xIfnrs] (pink). See Source Data and Extended Data

Fig. 5 for exact *P* values. **c**, Swim path efficiency of mice navigating to the escape platform in an MWM for male and female (M and F), WT ($n = 39$, 19 male and 20 female), Dp16 ($n = 28$, 13 male and 15 female) and Dp16[2xIfnrs] ($n = 25$, 12 male and 13 female). Statistics as in **b. d**, Representative swim trials for males from block 9 in the MWM with platform location denoted for the acquisition (gray circle) and reversal (black circle) phases. **e**, Mouse target quadrant occupancy during reversal probe trial of MWM; data are represented as means ± s.e.m., with significance determined by one-way ANOVA and Dunnett's correction. **f**, Heatmap displaying *P* values from IPA of transcriptome changes in brains from Dp16 versus WT animals, ranked by decreasing significance (right-tailed Fisher's exact test, WT ($n = 6$, 2 male and 4 female), Dp16 ($n = 5$, 2 male and 3 female) and Dp16[2xIfnrs] ($n = 7$, 4 male and 3 female). **g**, Heatmaps (top) representing median RPKM expression *z* scores per genotype and sina plots (bottom) for the example genes from the indicated pathways; *q* values determined by DESeq2, with significance set at $q < 0.1$.

Fig. 6b). Remarkably, 23% (34/149) of these differences were rescued in Dp16[2xlfnrs], including 79% (11/14) of Dp16 mandibular phenotypes (Fig. 6b and Extended Data Fig. 6b). The remaining interlandmark differences that persisted in Dp16[2xlfnrs] were less drastic than in Dp16 (Extended Data Fig. 6c and Supplementary Table 11). Examples include a shortening of

the basisphenoid (BS) bone (distance between landmarks 24 and 27), increase in the sphenofrontal suture width (a proxy for intertemple width; distance between landmarks 21 and 22) and alterations in many mandibular interlandmark distances (for example, landmarks 18–20), all of which are observed in Dp16 but attenuated in Dp16[2xlfnrs] (Fig. 6c).

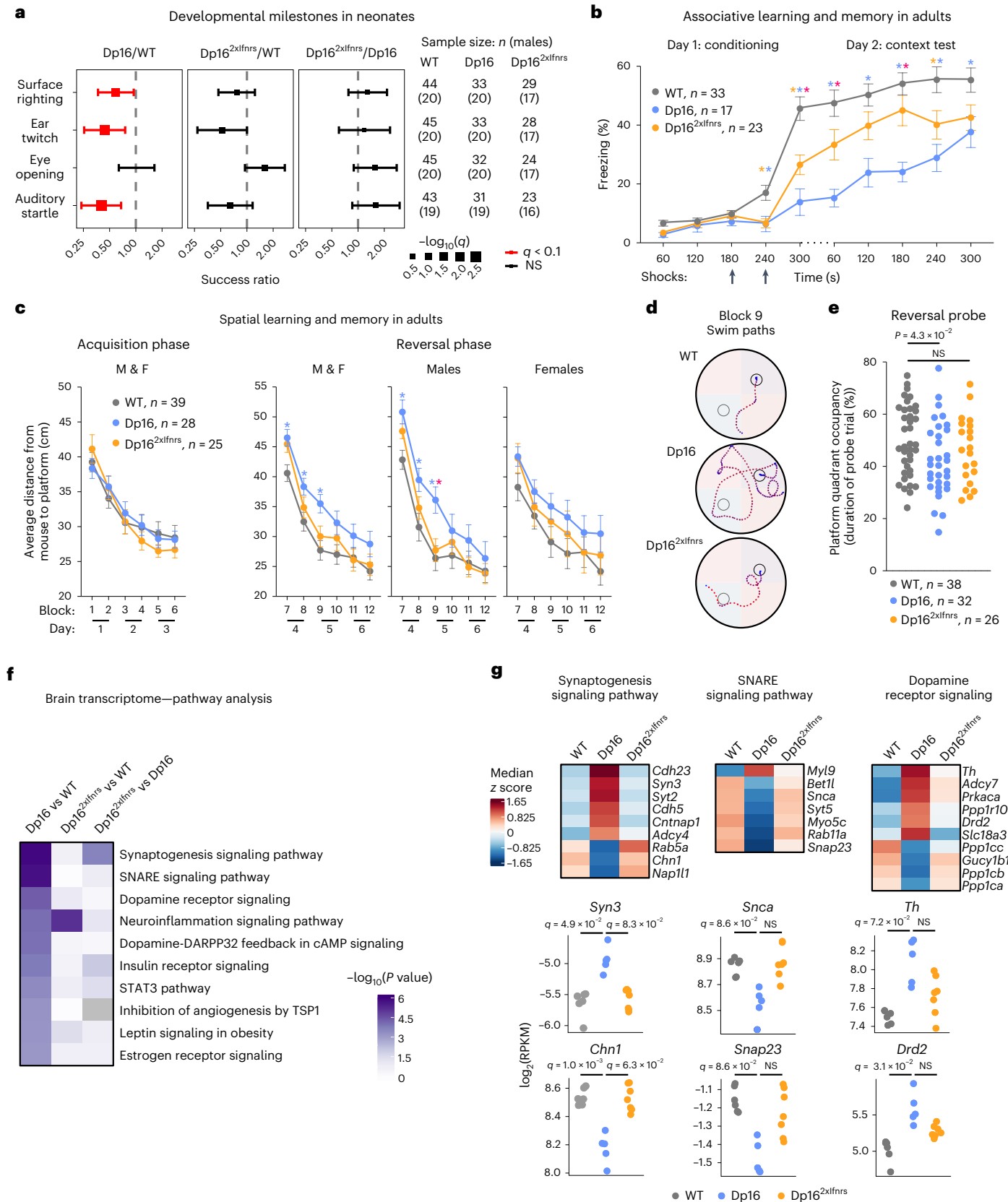

Many of the differences sensitive to *Ifnr* locus dosage clustered at the cranial base (that is, landmarks 23–30, Fig. 6d). Inspection of the cranial base revealed a loss of intersphenoidal synchondrosis (ISS, the cartilaginous joint between two parts of the sphenoid bone) in Dp16, which was attenuated in Dp16²ˣˡᶠⁿʳˢ (Fig. 6e,f). This trait is likely due to premature fusion of the presphenoid (PS) and BS bones and resembles the early mineralization of the spheno-occipital synchondrosis (SOS; the cartilaginous joint between the sphenoid and basioccipital bones) observed in individuals with DS[31]. Early mineralization of the anterior cranial base restricts midfacial outgrowth and is often associated with the altered shape of the calvarium, the upper domelike portion of the skull[32]. Notably, Dp16 display substantially shortened BS length and midface length (landmarks 24–27 and 1–24, respectively), but these phenotypes are ameliorated in Dp16²ˣˡᶠⁿʳˢ (Extended Data Fig. 6b). Furthermore, we observed a significant inverse correlation between BS length and midface length versus ISS fusion (Fig. 6g,h).

To gain further insight, we completed transcriptome analysis of neural crest-derived facial mesenchyme at E10.5, an embryonic tissue relevant to craniofacial development. As for other tissues, triplicated genes were clearly overexpressed in Dp16 and Dp16²ˣˡᶠⁿʳˢ (Extended Data Fig. 6d). Pathway analysis revealed key changes in Dp16 attenuated in Dp16²ˣˡᶠⁿʳˢ, including the induction of multiple genes involved in oxidative phosphorylation and downregulation of gene signatures associated with cell proliferation (G2M, mitotic spindle and E2F targets; Fig. 6i,j, Extended Data Fig. 6e and Supplementary Table 10). These results indicate that cell proliferation in this embryonic tissue is negatively impacted by an extra copy of the *Ifnr* locus. As for other traits sensitive to *Ifnr* locus dosage, Dp16²ˣˡᶠⁿʳˢ mice show amelioration of craniofacial phenotypes even with mild dampening of dysregulated gene expression.

Altogether, these results indicate that triplication of *Ifnrs* contributes to major craniofacial features distinctive of DS in mice, suggesting a role for hyperactive IFN signaling in dysregulated skeletal morphogenesis.

## Discussion

Despite many efforts, the mechanisms by which trisomy 21 causes the developmental and clinical hallmarks of DS remain poorly understood[3]. Multiple genes could contribute to a specific phenotype, and the aneuploidy itself could exert effects independent of gene content[3,12]. Clearly, elucidation of gene–phenotype relationships in DS would accelerate therapeutic strategies to serve this population. Within this framework, deciphering the mechanisms by which trisomy 21 causes lifelong IFN hyperactivity could enable immunomodulatory strategies to improve health outcomes in DS.

Using transcriptome analysis of a large human cohort study, we observed that overexpression of few select HSA21 genes correlates with signatures of IFN hyperactivity and inflammation, including the four

IFNRs. Although hyperactive IFN signaling has been noted in cells with trisomy 21 since the 1970s, its contribution to systemic phenotypes of DS in vivo has not been defined[9]. In the Ts16 mouse strain carrying triplication of essentially all MMU16 genes, including many genes not orthologous to HSA21, reduction of IFN signaling improved some aspects of Ts16 fetal development[33]. However, because these mice die shortly after birth, examination of postnatal phenotypes was unfeasible[33]. We, therefore, reduced *Ifnr* locus dosage in the Dp16 model[16,34]. This approach revealed that triplication of the *Ifnr* locus contributes to a lethal antiviral response, heart malformations, developmental delays, cognitive deficits and craniofacial abnormalities in mice. These results expand the evidence for harmful effects of aberrant IFN signaling in development[13], while supporting the notion that DS can be understood in part as an interferonopathy[6,14,35]. Nevertheless, it is also possible that some of the effects observed are due to other events affected by the triplication of this 192 kb genomic locus, including potential contributions from noncoding RNAs and *cis*-regulatory elements.

Our results may help explain the high rate of morbidity and mortality from respiratory infections observed in DS, as well as the increased rate of autoimmune disorders[4,5,36–38]. Trisomy 21 is a top risk factor for severe COVID-19, leading to increased rates of hospitalization and mortality[38,39]. IFN signaling exerts both protective and harmful effects on COVID-19 pathology[40–49]. Reduced *Ifnar1* copy number prevents lung pathology in mouse models of SARS-CoV-1/2 infections[50,51] and Type I/III IFNs disrupt lung barrier function during viral infections[52,53]. IFN hyperactivity has been consistently associated with autoimmunity, as both pharmacological IFN treatment and genetic variants leading to heightened IFN signaling increase the risk of developing autoimmune conditions[54,55]. Notably, correction of *Ifnr* locus dosage does not fully rescue transcriptome signatures of inflammation and immune dysregulation in Dp16 mice, indicating the presence of additional mechanisms contributing to immune dysregulation in DS. For example, cellular stress from extra DNA and resulting activation of damage-associated molecular patterns could activate IFN signaling by elevating IFN ligands or alternative mechanisms[12,56].

Our findings define a role for the *Ifnr* locus during heart development. In mice, several regions orthologous to HSA21 were shown to contribute to heart malformations, only some of which include the *Ifnr* gene cluster, supporting the notion of a polygenic basis for this phenotype (Extended Data Fig. 7)[3,16,23,24,57–59]. Although our results do not demonstrate that *Ifnr* triplication is sufficient to cause CHD, they indicate that the *Ifnr* locus contributes to this trait, likely potentiating the effects of other necessary genes. Notably, polymorphisms in *IFNGR2* and *IL10RB* have been associated with the risk of CHD in DS[60]. We show here that triplication of the *Ifnr* locus contributes to increased JAK/STAT signaling and dysregulation of major signaling pathways in the developing heart, including cell cycle control, EMT and mitochondrial metabolism. Although these effects are more pronounced during

**Fig. 6 | Triplication of the *Ifnr* locus exacerbates craniofacial anomalies in a mouse model of DS. a**, Representative lateral views of μCT reconstructions of skulls from WT, Dp16 and Dp16²ˣˡᶠⁿʳˢ animals, aligned and scaled based on the same 3D linear measurement. **b**, Lateral (top) and dorsal (middle) views of the outer portion of the skull proper transparently overlaid on cranial base interior view from a WT mouse, with landmarks on the skull (top and middle) and mandible (bottom) in yellow, and interior landmarks on the cranial base in turquoise. Smaller and larger interlandmark distances (blue and green, respectively) in WT relative to Dp16 or Dp16²ˣˡᶠⁿʳˢ calculated by EDMA. Number of animals—WT ($n = 7$, 2 male and 5 female), Dp16 ($n = 7$, 4 male and 3 female) and Dp16²ˣˡᶠⁿʳˢ ($n = 6$, 4 male and 2 female), followed by bootstrapping 10,000 times. Turquoise—distances different in WT relative to Dp16 rescued in Dp16²ˣˡᶠⁿʳˢ. Red—distances with no difference in WT relative to Dp16 but different in Dp16²ˣˡᶠⁿʳˢ. **c**, Example interlandmark distances before bootstrapping. Number of animals as in **b**. *P* values determined by one-way ANOVA with Tukey's HSD test, with significance set at $P < 0.05$; horizontal dashes indicate means. **d**, Form

difference ratios of mean population estimates for distances on the skull after bootstrapping, with colors assigned to any distance that includes a cranial base landmark. **e**, Images of cranial base interior views for WT and Dp16 displaying normal or completely fused ISS. **f**, Frequency of complete ISS fusion (black) compared between cohorts by pairwise two-sided Fisher's exact test with significance set at $P < 0.05$. Number of animals—WT ($n = 20$, 12 male and 8 female), Dp16 ($n = 11$, 6 male and 5 female) and Dp16²ˣˡᶠⁿʳˢ ($n = 14$, 7 male and 7 female). **g,h**, Scatter plots comparing ISS fusion scores to BS length (**g**) or midface length (**h**) for each skull where total $n = 20$ (numbers per genotype and sex as in **b**), showing Spearman $\rho$ and *P* values (permutation test). Black lines represent linear fits with 95% CIs shaded in gray. **i**, Heatmap displaying median RPKM expression of *z* scores for example genes dysregulated in E10.5 facial mesenchyme. Number of animals—WT ($n = 3$, 2 male and 1 female), Dp16 ($n = 3$, 2 male and 1 female) and Dp16²ˣˡᶠⁿʳˢ ($n = 3$, 2 male and 1 female). **j**, Sina plots displaying expression levels for example genes; *q* values determined by DESeq2 with Benjamini–Hochberg correction, with significance set at $q < 0.1$. PS, presphenoid bone.

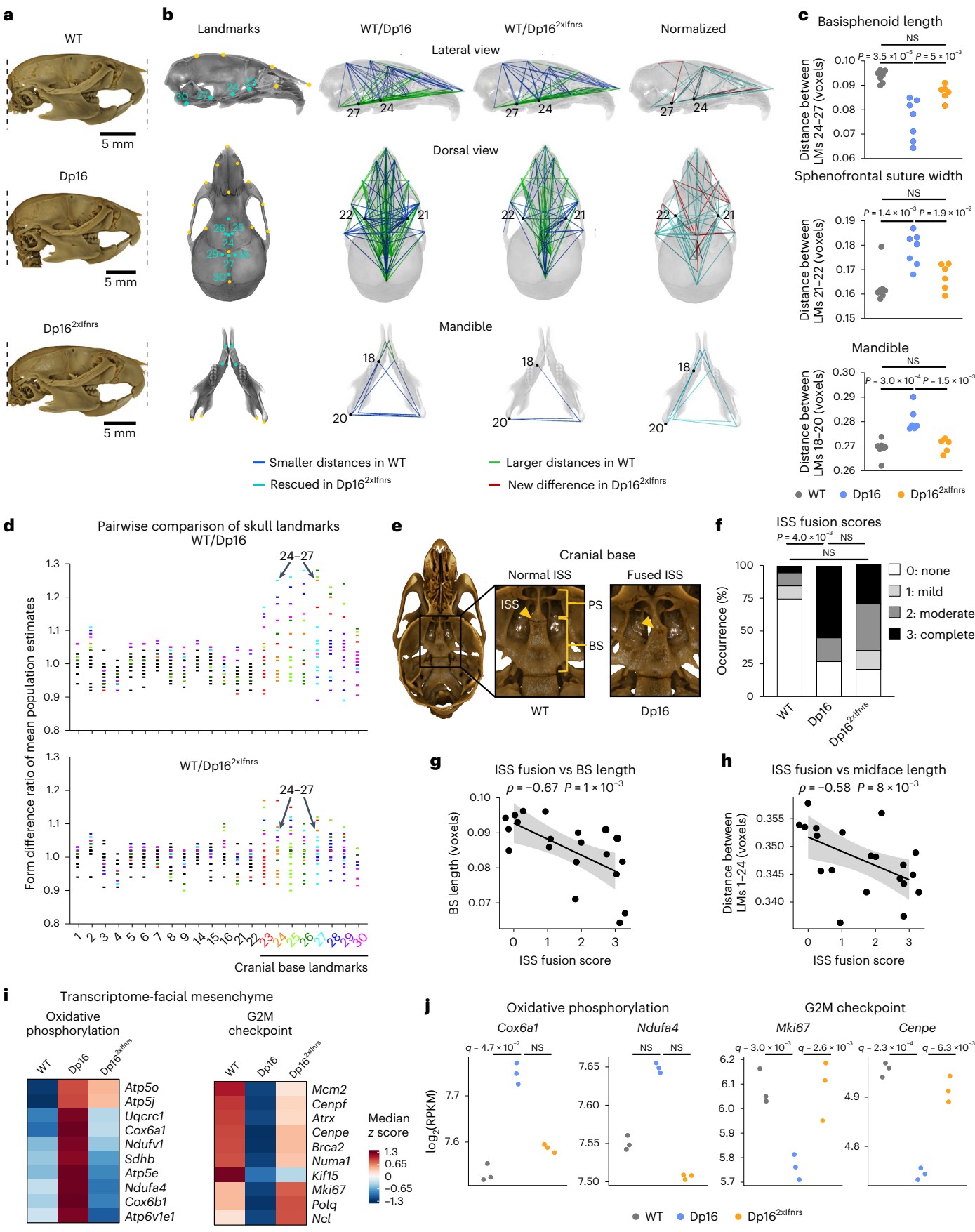

early development, some persist in adult heart tissue. Intriguingly, the developing heart in Dp16 displays clear signs of inflammation in the absence of known immune stimuli. Various IFNs are made throughout fetal development, such as Type III IFNs constitutively produced by syncytiotrophoblasts[61,62]. Hyperactive IFN signaling has been indirectly implicated in abnormal prenatal heart development during maternal lupus, viral infections and some monogenic interferonopathies, albeit with variable penetrance[61,63,64]. Interferonopathies caused by mutations in *ADAR*, *IFIH1* and *DDX58* have been associated with childhood-onset cardiac valvular disease[64–66]. However, not all disorders of elevated JAK/STAT signaling are associated with cardiovascular abnormalities, as illustrated by STAT1 gain-of-function mutations[67]. Therefore, additional research is needed to illuminate the mechanisms by which the *Ifnr* locus contributes to CHD.

Our results showing that triplication of the *Ifnr* locus affects the development of the proper skull, cranial base and mandible provides much-needed insight about the etiology of this phenotype, with potential implications for understanding abnormal fetal development in other settings. Maternal viral infections known to alter the inflammatory milieu for a developing fetus can disrupt bone development and cause cranial calcifications and microcephaly, also common in monogenic interferonopathies[68].

Lastly, the observed effects of *Ifnr* triplication on early development and cognitive function lend support to the notion of hyperactive IFN signaling as a driver of brain pathology after congenital infections and in interferonopathies[13,61]. Our transcriptome analysis revealed multiple pathways sensitive to *Ifnr* locus dosage, including synaptogenesis, SNARE signaling and dopamine signaling, all of which could contribute to cognitive phenotypes. Notably, the sex-specific differences in Dp16 execution of the MWM observed in our study are in line with previous studies[16,18–20], but the relevance of this sexual dimorphism in people with DS warrants further investigation[69–71]. Correction of *Ifnr* locus copy number did not fully rescue the cognitive impairments in Dp16, nor the global gene expression changes, indicating that other triplicated genes could also have a role. For example, *Dyrk1a*, a gene with documented roles in brain development and function in DS and other genetic disorders[72], is overexpressed in Dp16[2xIfnrs] tissues, including the brain, where it could exert additional effects independent of *Ifnr* dosage.

This study has several limitations. First, the contribution of the *Ifnr* locus was defined in a single mouse model of DS, and caution should be exercised when extrapolating these results to the human condition. Second, there are many examples of experimental interventions that reversed phenotypes in mouse models of DS[73–77], but none of these has been translated yet into an approved therapy. The notion that prenatal interventions targeting IFN signaling could ameliorate CHD or cognitive impairments would need support from additional clinical research. Third, the phenotypic differences observed upon correction of *Ifnr* locus dosage are accompanied by partial attenuation of global gene expression changes, and it is unclear which of the pathways affected by *Ifnr* triplication contribute to the rescued phenotypes. Furthermore, these results indicate that many effects of the trisomy are independent of *IFNR* dosage, including full induction of inflammatory pathways.

Despite these important caveats, our results encourage future research to define the value of immunomodulatory agents in DS. IFN signaling can be attenuated with agents approved for the treatment of diverse autoinflammatory conditions, most prominently JAK inhibitors[21,78]. JAK inhibition blocks the immune hypersensitivity phenotype observed in Dp16 mice[21], and a clinical trial for JAK inhibition in DS is underway (NCT04246372). In young Dp16 animals, acetaminophen treatment decreased microglia activation and improved cognition[73]. In alternative mouse models also harboring triplication of *Ifnrs* and displaying hyperactive IFN signaling[20], developmental delays were rescued by prenatal treatment with the anti-inflammatory compound apigenin[74]. Nevertheless, other therapeutic approaches without clear immunomodulatory effects are also strongly supported by research in

mouse models of DS (reviewed in refs. 76,77). In sum, our results point to a role for cytokine signaling in key phenotypes of DS, supporting additional research on the potential benefits of immunomodulating agents in this population.

## Online content

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

## Methods

### Clinical study design

The main goals of the study were to determine associations across inflammatory markers in people with DS in comparison to euploid controls (D21) using RNA sequencing (RNA-seq) and multiplexed immunoassays. Research participants were enrolled in the Crnic Institute Human Trisome Project Biobank (HTP) under a study protocol approved by the Colorado Multiple Institutional Review Board (COMIRB, 15-2170). Information on sample size by karyotype, sex and age is presented in Supplementary Table 1. Participants received US$100 compensation per blood draw. Procedures were performed in accordance with COMIRB guidelines and regulations. Written informed consent was obtained from participants who were cognitively able or their guardians. This study has been conducted in accordance with the Declaration of Helsinki.

### Whole-blood transcriptome analysis

Peripheral blood was collected in PAXgene RNA Tubes (QIAGEN), and total RNA was purified using the PAXgene Blood RNA Kit (QIAGEN). Globin RNA depletion, poly-$A^+$ RNA enrichment and strand-specific library preparation were carried out using the GLOBINclear Human Kit (Thermo Fisher Scientific), NEBNext Poly(A) mRNA Magnetic Isolation Module (New England Biolabs) and NEBNext Ultra II Directional RNA Library Prep Kit for Illumina (New England Biolabs). Paired-end 150 bp sequencing was carried out on an Illumina NovaSeq 6000 instrument. Reads were demultiplexed and converted to FASTQ format using bcl2fastq v2.20.0.422. Data quality was assessed using FASTQC v0.11.5 and FastQ Screen v0.11.0. Filtering of low-quality reads was performed using bbduk from BBTools v37.99 (ref. [79]) and fastq-mcf from ea-utils v1.05. Alignment to the human reference genome (assembly GRCh38) was carried out using HISAT2 v2.1.0 (ref. [80]). Alignments were sorted and filtered for mapping quality (MAPQ > 10) using SAMtools v1.5 (ref. [81]). Gene-level count data were quantified using HTSeq-count v0.6.1 (ref. [82]). Differential gene expression in DS versus euploid controls was evaluated using DESeq2 v1.28.1 with age, sex and sample source, as covariates in R v4.0.1 using $q < 0.1$ [false discovery rate (FDR) < 10%] as the threshold as recommended for DESeq2 (ref. [83]).

### Analysis of human inflammatory markers in plasma

Peripheral blood was collected into BD Vacutainer K2 EDTA tubes (Beckton Dickinson) and separated into plasma, buffy coat and red blood cells (RBCs). CRP and IL6 were measured in two technical replicates for each plasma sample using V-PLEX Human Biomarker 54-Plex Kit ((Meso Scale Discovery (MSD)) on a MESO QuickPlex SQ 120 instrument. Concentration values (pg ml$^{-1}$) were calculated against a standard curve using provided calibrators and imported to R v4.0.1. For each analyte, missing values were replaced with either the minimum (if below-fit curve range) or maximum (if above-fit curve range) calculated concentration and means of replicates used in downstream analysis. Extreme outliers were classified as measurements more than three times the interquartile range below or above the first and third quartiles, respectively, and excluded from the analysis. Differential abundance was defined using mixed effects linear regression as implemented in the *lmer()* function from the lmerTest R package v3.1-2 with log$_2$-transformed concentration as the outcome/dependent variable, trisomy 21 status as the predictor/independent variable, age and sex as fixed covariates and sample source as a random effect. Multiple hypothesis correction was performed with the BH method using FDR10% ($q < 0.1$).

### Spearman correlation analysis

Spearman $\rho$ values and $P$ values were calculated using the *rcorr* function from the Hmisc package v4.4-0 with BH correction. XY scatter plots with points colored by density were generated using ggplot2 v3.3.1 (ref. [84]). Ranked lists of Spearman values were analyzed by GSEA[85] as described below.

### GSEA and ingenuity pathway analysis (IPA)

GSEA[85] was carried out in R using the fgsea package v1.14.0 (ref. [86]), using Hallmark gene sets[87] and either a ranked list of log$_2$-transformed fold-changes (for transcriptome studies) or a ranked list of Spearman $\rho$ values as the ranking metric. GSEA was used to test for enrichment of specific gene sets within a ranked list to define whether specific signaling pathways were enriched among upregulated or downregulated mRNAs (in transcriptome studies) or among positively or negatively correlated features (in Spearman correlation analyses). We used IPA (release winter 2022), which employs right-tailed Fisher's exact tests to define the enrichment of a particular gene set within a gene list, to investigate pathways enriched among DEGs in the adult brain transcriptome study.

### Heterozygote deletion of the *Ifnr* locus on MMU16

One copy of the 192 kb genomic segment containing the four *Ifnr* genes was deleted using CRIPSR–Cas9 (ref. [34]). CRISPR–Cas9 target sites were identified using http://crispr.mit.edu/. Two guide RNAs (gRNAs) were synthesized per target site flanking the *Ifnr* gene cluster on MMU16 using the MEGAshortscript T7 Transcription Kit (Life Technologies) and MEGAclear Transcription Clean-Up Kit (Life Technologies) (Supplementary Table 4). C57BL/6NTac zygotes (Taconic) were microinjected with 25 ng μl$^{-1}$ Cas9 mRNA (Sigma) and four total gRNAs at 7 ng μl$^{-1}$ each, then implanted into pseudopregnant females. Heterozygote WT$^{1xIfnr}$ (*Ifnr$^{-/+}$*) mutant mice were made in collaboration with Dr. Jennifer Matsuda and James Gross of the Genetics Core Facility at National Jewish Health, CO.

### Mice genotyping

DNA was genotyped by PCR[34]. Genomic DNA was prepared from 1 mm to 2 mm of toe, tail or ear tissue using the HotSHOT method[88] then run through PCR according to Supplementary Table 4 or outsourced for automated genotyping by reverse transcription–PCR with specific probes designed for each gene (Transnetyx). Potential founders (F0s) were genotyped by PCR to identify those that appeared to lack the entire *Ifnr* cluster without additional chromosomal rearrangements.

### Sanger sequencing of the deleted region

Potential F0s lacking one copy of the *Ifnr* gene cluster (WT$^{1xIfnrs}$) were bred to WT mice on the same C57BL/6NTac (Taconic) substrain background to generate heterozygous F1 progeny (WT$^{1xIfnrs}$). PCR products spanning the deleted region were subjected to Sanger sequencing using a 3730xl DNA Analyzer (Thermo Fisher Scientific) to identify transmission of a single modified allele (oligonucleotides in Supplementary Table 4). Sequence-verified F1 WT$^{1xIfnrs}$ mice with identical deletion events were selected to maintain the line started by a single F0 WT$^{1xIfnrs}$ male.

### WGS of mutant mice

WGS was used to confirm clean chromosomal rearrangements of the proven male F0 C57BL/6NTac (Taconic) WT$^{1xIfnrs}$ by copy number variant (CNV) analysis[89]. Peripheral blood was collected in heparin sodium tubes (Sigma-Adrich) from the submandibular vein of the F0 male WT$^{1xIfnrs}$ and two male C57BL/6NTac WT controls; all mice were 6–8 months old at the time of blood draw. RBCs were then lysed by ammonium chloride-potassium[90] and DNA was isolated using AllPrep DNA/RNA/Protein Mini Kit (QIAGEN). Libraries were generated from 1 mg DNA using NEBNext DNA Library Prep Kit (New England Biolabs). DNA was fragmented to ~350 bp by shearing, end-polished, A-tailed and ligated with the NEBNext adapter for Illumina sequencing, and further PCR-enriched by P5 and indexed P7 oligonucleotides (Supplementary Table 4). PCR products were purified and sequenced on an Illumina NovaSeq 6000 instrument. Analysis of library complexity and sequence quality was performed using FastQC v0.11.5. Trimming of low-quality bases, short reads and adapter sequences

with the fastqc-mcf tool; removal of mycoplasma, mitochondria and rRNA contaminant sequences with FASTQ Screen v0.11.0; read alignment to the GRCm38 [mm10] reference using BWA v0.7.15; filtering of high-quality mapped reads with SAMtools v1.5 and final quality performed using RSeQC v4.0.0. Using the package CNV-seq v0.2-8, we confirmed the presence of *Ifnr* deletion in the F0 male WT[1xIfnrs]. Given a significance level ($p'$) of 0.01 and a CNV detection threshold ratio ($r'$) of 0.06, the theoretical minimum window size was determined by the default method[89]. The window size used for the detection of CNVs was 1.5× (default) the theoretical minimum window size. Using these parameters, we confirmed the presence of the deletion with no other coherent CNVs of similar size.

### Animal husbandry

Experiments were approved by the Institutional Animal Care and Use Committee at the University of Colorado Anschutz Medical Campus under protocol 00111 and performed in accordance with National Institutes of Health (NIH) guidelines. One candidate F1 male progeny of the validated F0 WT[1xIfnrs] was backcrossed to WT C57BL/6J (The Jackson Laboratory) for at least three generations before female WT[1xIfnrs] were intercrossed with Dp16 males, also of the C57BL/6J background. Dp16 mice[16] were originally purchased from The Jackson Laboratory and also gifted by Dr. Faycal Guedj and Dr. Diana Bianchi (NIH), then intermixed and maintained on the C57BL/6J background. WT[1xIfnrs] mice were viable and fertile, with no obvious phenotypes. After intercrossing female WT[1xIfnrs] with male Dp16, mice were confirmed to be at least 87.5% C57BL/6J via Transnetyx automated PCR services. To power the comparison arms representing each genotype in this study, multiple litters were combined, where each litter contributed randomly to the sum of each cohort, thus minimizing the impact of any potential shift in genetic background. Mice were housed separately by sex in groups of 1–5 mice per cage under a 14 h light/10 h dark cycle with controlled temperature and 35% humidity, and ad libitum access to food (6% fat diet) and water. Unless otherwise noted, for all subsequent experiments, all mice were at least 87.5% C57BL/6J with the remaining background inferred as C57BL/6NTac (Taconic).

### Spectral flow cytometry analysis of IFNR surface expression

Peripheral blood was collected from 4- to 6-month-old mice from the submandibular vein into lithium heparin tubes (Sarstedt) and stained[21]. Twenty-five microliters of fresh whole blood was pre-incubated with anti-mouse CD16/CD32 (TruStain FcX; BioLegend, clone 93) at 1:100 for 10 min at room temperature then spiked with a concentrated stain of all surface markers for 30 min at 4 °C. Staining with fluorochrome-conjugated antibodies (complete antibody information in Supplementary Table 4) were diluted in BD Horizon Brilliant Stain Buffer Plus (BD Biosciences).

After staining, whole blood underwent two immediate 2 min and 5 min incubations in 200 µl of ammonium chloride-potassium (ACK) lysis buffer[90]. Cells were then washed twice in flow cytometry wash buffer (1× PBS, 2% FBS, 10 mM HEPES pH 7.5 and 0.1% sodium azide) and then fixed for 10 min at room temperature in paraformaldehyde (Ted Pella) diluted to 4% in 1× PBS. Fixative was washed-out in flow cytometry wash buffer, and then cells were analyzed using a five-laser Cytek Aurora spectral flow cytometer. Flow cytometry data were analyzed with FlowJo v10 (Becton Dickinson). Number of animals per group were as follows: IFNAR1 WT[1xIfnrs] ($n = 5$, 3 male and 2 female), WT ($n = 8$, 2 male and 6 female), Dp16 ($n = 7$, 5 male and 2 female) and Dp16[2xIfnrs] ($n = 6$, 4 male and 2 female); IFNGR2 WT[1xIfnrs] ($n = 5$, 3 male and 2 female), WT ($n = 16$, 10 male and 6 female), Dp16 ($n = 7$, 5 male and 2 female) and Dp16[2xIfnrs] ($n = 8$, 6 male and 2 female); and IL10RB WT[1xIfnrs] ($n = 14$, 3 male and 11 female), WT ($n = 17$, 7 male and 10 female), Dp16 ($n = 15$, 7 male and 8 female) and Dp16[2xIfnrs] ($n = 15$, 8 male and 7 female).

### Spectral flow cytometry to assess pSTAT1

Peripheral blood was collected and stained as described above for IFNR stains[21]. Twenty-five microliters of blood were subjected to RBC lysis and then stimulated for 30 min at 37 °C with 10,000 units per ml of recombinant IFN-α2A (R&D Systems) or 100 unit ml[−1] of recombinant IFN-γ (R&D Systems). Antibodies conjugated to methanol-stable fluorophores targeting epitopes that are not stable through fixation were also included in the stimulation media in the presence of FcR block (that is, SiglecF, Ly6C, CD115, CD8 and CD11b). Following stimulation, cells were washed in FACS buffer, subjected to BD Lyse-Fix buffer (BD Biosciences), then to permeabilization buffer III (BD Biosciences). Cells were then stained with fluorophore-conjugated antibodies specific for the following epitopes that were not destroyed with fixation: CD45, CD3, CD4, B220, NK1.1, Ly6G, IA/IE and phospho-STAT1 (Tyr701). See Supplementary Table 4 for antibody information. Fixative was washed-out in FACS buffer then cells were analyzed using a Cytek Aurora instrument. Data were analyzed with FlowJo v10 (Becton Dickinson). Number of animals per group were as follows: unstimulated WT ($n = 25$, 16 male and 9 female), Dp16 ($n = 15$, 7 male and 8 female) and Dp16[2xIfnrs] ($n = 8$, 8 male); +IFN-α WT ($n = 23$, 14 male and 9 female), Dp16 ($n = 12$, 3 male and 9 female) and Dp16[2xIfnrs] ($n = 9$, 9 male); and +IFN-γ WT ($n = 25$, 16 male and 9 female), Dp16 ($n = 12$, 12 male) and Dp16[2xIfnrs] ($n = 7$, 7 male).

### Enzyme-linked immunoassay (ELISA) for IFNAR2

Peripheral blood was collected from the submandibular vein of 3- to 9-month-old mice into tubes containing serum gel with clotting activator (Sarstedt). Serum was diluted 1:1,000 and analyzed by ELISA with Mouse IFN-alpha/beta R2 ELISA Kit (RayBiotech). Plates were analyzed on a Synergy H4 Hybrid Multi-Mode Microplate Reader (BioTek). Number of animals (undocumented sex) per group were as follows: WT[1xIfnrs] ($n = 28$), WT ($n = 30$), Dp16 ($n = 13$) and Dp16[2xIfnrs] ($n = 26$).

### Poly(I:C) treatment

Poly(I:C) (InvivoGen) of 10 mg kg[−1] was administered intraperitoneally at 2-d intervals for up to 16 d. Animals were killed one day after the final dose (day 17) or on losing ≥15% of body weight. Number of animals per group were as follows: WT sham ($n = 9$, 2 male, 7 female), WT poly(I:C) ($n = 13$, 6 male, 7 female), Dp16 poly(I:C) ($n = 9$, 6 male, 3 female) and Dp16[2xIfnrs] poly(I:C) ($n = 13$, 6 male, 7 female).

### Flow cytometry to measure cytokine concentrations

Peripheral blood was collected from the submandibular vein of 5- to 9-month-old mice into lithium heparin tubes (Sarstedt) on day 3 at 18 h after the second poly(I:C) exposure of the chronic exposure timeline described above. Plasma cytokine levels were measured using the LEGENDplex Mouse Anti-Virus Response Panel (BioLegend) on an Accuri C6 flow cytometer and analyzed using LEGENDplex v2021. All samples were analyzed in duplicate, and the average was used for statistical analysis. Number of animals per group were as follows: WT sham ($n = 6$, 2 male and 4 female), WT poly(I:C) ($n = 7$, 3 male and 4 female), Dp16 poly(I:C) ($n = 6$, 5 male and 1 female) and Dp16[2xIfnrs] poly(I:C) ($n = 6$, 3 male and 3 female).

### Embryo tissue collection

Male Dp16 mice were crossed overnight with 8- to 12-week-old synced female WT[1xIfnrs]. Dams were checked daily for vaginal plugs; the first morning of visual confirmation was denoted embryonic day (E)0.5. At E12.5, E15.5 upon four-chamber heart formation[91], and E18.5, embryos were collected after $CO_2$ asphyxiation and cervical dislocation and allowed to exsanguinate on ice in 1× PBS. At E12.5 and E18.5, hearts were manually dissected from fresh embryos and then flash-frozen at −80 °C.

To enrich for neural crest cell-derived facial mesenchyme after dam killing as described above, maxillary and mandibular processes were isolated from the developing face of embryos at E10.5 (ref. 92).

Using a dissecting stereomicroscope, the myometrium, decidua, chorion and amnion were pulled away from the embryo slowly and sequentially to avoid disruption of cranium form through sudden pressure change. Maxillary and mandibular processes were manually dissected from the face. Mesenchymal tissue was isolated following the removal of the ectoderm via digestion in 2% trypsin in PBS on ice (BioWorld). Facial process mesenchyme was then stored in 594 μl of lysis buffer RLT Plus (QIAGEN) and 6 μl of 2-mercaptoethanol (Sigma-Aldrich) at −80 °C for RNA extraction.

### Embryonic heart histology
On collection, embryos were fixed at 4 °C overnight in 2% paraformaldehyde (Ted Pella) diluted in 1× PBS and then stored in 70% ethanol before embedding in paraffin. Embedded embryos were sectioned transversely at 7 μm thickness using a LEICA RM 2155 Rotary Microtome. Serial sections were collected and then hematoxylin and eosin Y (H&E) staining was performed, followed by imaging with a Keyence BZ-X710 All-in-One Fluorescence Microscope. The investigators who sectioned embryos and performed H&E analysis were blind to embryo genotype. Number of animals per group were as follows: WT ($n$ = 24, 12 male and 12 female), Dp16 ($n$ = 18; 9 male, 7 female and 2 undocumented sex) and Dp16[2xlfnrs] ($n$ = 16, 4 male and 12 female).

### Western blot of embryo hearts
Hearts were collected from fresh embryos at E15.5 and then frozen in liquid nitrogen. Frozen tissues were lysed in ice-cold lysis buffer (150 mM NaCl, 50 mM Tris–HCl pH 7.4, 1 mM EDTA, 1% Triton), with Complete mini tablet (Roche), 1 mM phenylmethylsulphonyl fluoride and 1× Halt Phosphatase Inhibitor Cocktail (Thermo Fisher Scientific). Ten micrograms of protein lysates were resolved on a 10% polyacrylamide gel and transferred to a PVDF membrane. Supplementary Table 4 for antibody information. Number of animals (undocumented sex) per group were as follows: WT[1xlfnrs] ($n$ = 4), WT ($n$ = 3), Dp16 ($n$ = 3) and Dp16[2xlfnrs] ($n$ = 3).

### Developmental milestones
Neonatal achievement of developmental milestones[18,93–96] was analyzed from days (D)3–21 post birth at 800–1,100 h while experimenters were blind to genotype. Pups were placed in a holding cage with bedding maintained at 37 °C by heating pad and identified by marking footpads or tattoos. Length and weight were measured daily. Pups were assessed in a pseudorandom order by blinded investigators in the following order: (1) surface righting 2 d in a row (D3–10); (2) first day both eyes open (D7–21); (3) first day both ears twitch (D7–21) and (4) the first day of auditory startle (D11–21). Once criteria were reached, testing was stopped for that mouse.

To test for differences in the chance of success in achieving each milestone, results were treated as time-to-event data and analyzed using a mixed-effect Cox regression approach using the survival v3.2-7 (ref. 97), coxme v2.2-16 (ref. 98), emmeans v1.5.1 (ref. 99) and broom v0.7.9 (ref. 100) packages in R v4.0.1. Models for each milestone were generated using the *coxme()* function from the coxme package with a time-to-event survival object as the outcome variable, genotype as the variable of interest, and with adjustment for sex and sex × genotype interaction as fixed effects and litter as a random effect. Hazard ratios (referred to herein as 'success ratios'), 95% confidence intervals (CIs) and $P$ values for all pairwise genotype comparisons, either combined or stratified by sex, were obtained from model objects using the *emmeans()* and *contrast()* functions from the emmeans package, and $P$ values adjusted with the BH method. Number of animals per group were as follows: surface righting WT ($n$ = 44, 20 male and 24 female), Dp16 ($n$ = 33, 20 male and 13 female) and Dp16[2xlfnrs] ($n$ = 29, 17 male and 12 female); ear twitch WT ($n$ = 45, 20 male and 25 female), Dp16 ($n$ = 33, 20 male and 13 female) and Dp16[2xlfnrs] ($n$ = 28, 17 male and 9 female); eye-opening WT ($n$ = 45, 20 male and 25 female), Dp16 ($n$ = 32, 20 male

and 12 female) and Dp16[2xlfnrs] ($n$ = 24, 17 male and 7 female); auditory startle WT ($n$ = 43, 19 male and 24 female), Dp16 ($n$ = 31, 19 male and 12 female) and Dp16[2xlfnrs] ($n$ = 23, 16 male and 7 female).

### Cognitive tests
Four to five months old mice were handled 2 min per day for 2–5 d leading up to the first test. When mice were run through multiple assays, they were run through rotarod first, MWM and then CFC. For all assays, mice were allowed to acclimate to the experimental room in their home cages for 30 min before testing began each day at 1200–1600 h. All experiments were performed by experimenters blinded to genotype, but Dp16 mice are somewhat smaller and have craniofacial differences, which may be noticeable to some personnel. The rotarod test was used to measure motor coordination[20,28]. Each mouse was given two practice sessions at 16 revolutions per minute (RPM), followed by two test sessions at 16, 24 and 32 RPM. Each session ended after 120 s or when the mouse fell. Latency to fall was tracked using Rotarod v1.4.1 (MED Associates) and averaged across both test sessions for each speed. The MWM test[20,101,102] was done in a pool 120 cm in diameter filled with opaque water in which an escape platform was hidden at 30 cm away from the center of the area. Mice started with one unrecorded swim trial where they were directed to the platform by a visual cue on the platform. Mice were then released into the MWM at pseudorandomized locations of the pool edge for a total of four swim trials per MWM block and two blocks per day with the final reported experimental value per block reflecting the average of the four swim trials. Each swim trial concluded when the mouse found the hidden platform or after 1 min, whichever came first. Immediately following block 6, mice underwent a 1 min probe trial where the platform was removed to conclude the acquisition phase of swim trials 1–24. After the first probe, we immediately employed a reversal phase for another 6 blocks of swim trials 25–48 that immediately precluded a second 1 min Probe Trial. Swim data were collected using the video tracking system Ethovision v8.5 (Noldus). Nesting was applied to swim paths in Ethovision before analysis where the center point (mouse) must be between start and stop threshold velocities of 2 cm s$^{-1}$ and 1.75 cm s$^{-1}$ to avoid giving weight of initial interaction of animal to arena and tracking of experimenters' hand during mouse drop into the maze. The CFC[20] test of associative learning and memory was performed in CFC boxes (30.5 × 24.1 × 29.2 cm) consisting of a light, recording device and a metal grid on the floor (Med Associates). During the 5 min training phase, mice explored the enclosure freely before receiving two 0.5 mA foot shocks (2 s) at 180 s and 240 s. Testing phase occurred 24 h after training and was identical to training without foot shocks. During both phases, freezing behavior was measured with FreezeScan v2.00 (Clever Sys). Number of animals per group were as follows: rotarod WT ($n$ = 16, 8 male and 8 female), Dp16 ($n$ = 14, 7 male and 7 female) and Dp16[2xlfnrs] ($n$ = 17, 8 male and 9 female); MWM WT ($n$ = 39, 19 male and 20 female), Dp16 ($n$ = 28, 13 male and 15 female) and Dp16[2xlfnrs] ($n$ = 25, 12 male and 13 female); CFC WT ($n$ = 33, 13 male and 20 female), Dp16 ($n$ = 17, 8 male and 9 female) and Dp16[2xlfnrs] ($n$ = 23, 11 male and 12 female).

### Skull morphometric analysis
Mice were decapitated at 7–8 weeks of age, and then whole heads were imaged using a SkyScan 1275 (Microphotonics). Scanning was performed at 17.6-μm resolution using the following parameters: 55 kV, 180 mA, 0.5 mm Al filter; 0.3° rotation step over 180° and 3-frame averaging. All raw scan data were reconstructed to multiplanar slice data using NRecon v1.7.4.6 (Bruker). Reconstructed data were then rendered in 3D with consistent thresholding parameters using Drishti v2.6.5 (ref. 103) for gross visual assessment of the craniofacial skeleton. Representative-rendered images were captured and processed using Photoshop 24.2.0 (Adobe). Morphometric analysis of craniofacial landmarks was then used to compare skull form between genotypes[30]. Coordinates for 30 homologous landmarks were independently collected

from each 3D-rendered skull with Drishti v2.6.5 by two investigators blinded to genotype. Landmark datasets from two investigators were averaged and then normalized by their respective root centroid size (RCS) values to remove overall skull size as a variable and are reported in voxel units (Supplementary Table 11). The WinEDMA v2021 package was used to conduct EDMA, which analyses morphological differences between two groups of specimens by assessing the change in ratio values between respective landmark pairs[104]. Following Lele and Richtsmeier[105], the 90% CIs were calculated by bootstrapping the shape difference matrix 10,000 times. The FORM[105] procedure was employed to find interlandmark distances that differed between populations, as well as landmarks influencing or driving those differences. Interlandmark distances were deemed different between populations if the CIs did not cross 1. Number of animals per group were as follows: WT ($n = 7$, 2 male and 5 female), Dp16 ($n = 7$, 4 male and 3 female) and Dp16$^{2xlfnrs}$ ($n = 6$, 4 male and 2 female).

## Qualitative assessment of the skull

Except for calvaria rounding, which was done on mice 7–10 weeks of age, all qualitative assessments of skulls from decapitated mice were done at 7–8 weeks of age by an expert in craniofacial morphology blinded to genotypes. As mineralization of SOS typically begins around postnatal D28 to ultimately mediate fusion of the BS and basioccipital bones by 12 weeks of age[30], unusually pronounced ossification of the intersphenoid synchondrosis (ISS) was formally assessed. An ISS severity score was assigned using the following criteria: 0 = normal unfused appearance; 1 ≤ one-third of the ISS width bridged by ossification; 2 = between one-third and two-thirds of the ISS bridged by ossification; 3 = the presphenoid and BS bones of the anterior cranial base were completely fused, obliterating the ISS. Number of animals per group were as follows: WT ($n = 20$, 12 male and 8 female), Dp16 ($n = 11$, 6 male and 5 female) and Dp16$^{2xlfnrs}$ ($n = 14$, 7 male and 7 female).

## RNA-seq of mouse tissues

RNA was purified using TRIzol reagent (Invitrogen) following the manufacturer's instructions. Whole hearts from E12.5 and E18.5 mice were homogenized in Lysing Matrix D tubes (MP Biomedicals) with 500 μl of TRIzol reagent (Invitrogen) for 30 s using a Mini-Beadbeater-24 (BioSpec Products). For E10.5 facial mesenchyme, a water bath at 37 °C was used for quick freeze/thaw, and total RNA was isolated using the QIAshreder (QIAGEN) and AllPrep DNA/RNA/Protein Mini Kit (QIAGEN). For adult tissues, mice were killed by $CO_2$ asphyxiation and cervical dislocation then immediately perfused with 1× PBS using a Perfusion Two Automated Perfusion Instrument (Leica). Whole brain samples consist of the entire right brain hemisphere from 6- to 9-month-old mice. Mesenteric lymph nodes and heart were similarly removed from perfused 4- to 5-month-old mice. The 3–5 mesenteric lymph nodes per animal were flash-frozen together while brain and adult heart samples were first placed in Lysing Matrix D tubes (MP Biomedicals) containing 594 μl of lysis buffer RLT Plus (QIAGEN) and 6 μl of 2-mercaptoethanol (Sigma-Aldrich) before storage at −80 °C. Upon quick freeze/thaw at 37 °C, mesenteric lymph nodes were placed in Lysing Matrix D tubes also containing RLT with 2-ME. Adult tissues were then homogenized for 30 s using a Mini-Beadbeater-24 (BioSpec Products) and then frozen at −80 °C. Total RNA was isolated using the AllPrep Kit (QIAGEN) following the manufacturer's instructions. Library preparation was carried out using a Universal Plus mRNA Kit Poly(A) (Tecan). Paired-end, 150 bp sequencing was carried out on an Illumina NovaSeq 6000. Subsequent analysis was carried out as for human whole-blood RNA-seq, except alignment and gene-level count summarization used a mouse GRCm38 reference genome index and Gencode M24 annotation GTF. RNA-seq data yield was a minimum of ~30 million raw reads. Differential gene expression was evaluated using DESeq2 v1.28.1 with sex and batch as covariates, and $q < 0.1$ as the threshold for DEGs. GSEA[85] was carried out in R v4.0.1 as described above. Number of animals per group were as

follows: E10.5 facial mesenchyme WT ($n = 3$, 2 male and 1 female), Dp16 ($n = 3$, 2 male and 1 female) and Dp16$^{2xlfnrs}$ ($n = 3$, 2 male and 1 female); E12.5 hearts WT ($n = 6$, 2 male and 4 female), Dp16 ($n = 6$, 3 male and 3 female) and Dp16$^{2xlfnrs}$ ($n = 6$, 5 male and 1 female); E18.5 hearts WT ($n = 6$, 4 male and 2 female), Dp16 ($n = 6$, 3 male and 3 female) and Dp16$^{2xlfnrs}$ ($n = 6$, 3 male, 3 female); adult mesenteric lymph nodes WT ($n = 5$, 2 male, 3 female), Dp16 ($n = 6$, 3 male and 3 female) and Dp16$^{2xlfnrs}$ ($n = 6$, 3 male and 3 female); adult brains WT ($n = 6$, 2 male and 4 female), Dp16 ($n = 5$, 2 male and 3 female) and Dp16$^{2xlfnrs}$ ($n = 7$, 4 male and 3 female); and adult hearts WT ($n = 5$, 2 male and 3 female), Dp16 ($n = 6$, 3 male and 3 female) and Dp16$^{2xlfnrs}$ ($n = 5$, 3 male and 2 female).

## Statistical analysis and result visualization

The sample size was determined a priori based on effect sizes of previous studies[6,10,11,106,107] or by post hoc analyses to ensure >80% power was achieved to reduce type II error (for example, heart histology, developmental milestone achievement and cognition in adults). All statistical analyses were conducted in GraphPad Prism v8.0.1 or R v4.0.1 (refs. 108,109) and are listed with sample sizes in the corresponding figure legends. Preprocessing, statistical analysis and initial plot generation for all datasets were carried out using R v4.0.1. Method schematics were generated using Biorender.com. All figures were finalized in Adobe Illustrator v24.1.

## Reporting summary

Further information on research design is available in the Nature Portfolio Reporting Summary linked to this article.

## Data availability

The minimum dataset required to interpret, verify and extend the research in this article is made available in the accompanying Source Data files and through public repositories. Mouse WGS data are deposited in the National Center for Biotechnology Information Sequence Read Archive under BioProject ID PRJNA776534. Human RNA-seq data generated by the Crnic Institute Human Trisome Project are deposited in the Gene Expression Omnibus (GEO) with accession number GSE190125 and are also available through the INCLUDE Data Hub (https://portal.includedcc.org/). Human demographics and clinical source metadata are also available through the INCLUDE Data Hub. Murine RNA-seq data were deposited in GEO with the following accession numbers: GSE218883: adult mouse heart tissue; GSE218885: adult mouse brain tissue; GSE218887: embryonic mouse facial mesenchyme tissue; GSE218888: embryonic E12.5 mouse heart tissue; GSE218889: embryonic E18.5 mouse heart tissue; GSE218890: adult mouse mesenteric lymph nodes. All other source data are provided in the Source Data files with this manuscript. Reference datasets employed in this study were mouse reference genome assembly GRCm38 (mm10) with Gencode vM24 basic annotation (https://www.gencodegenes.org/mouse/release_M24.html), and human reference genome assembly GRCh38 (hg38) with Gencode v33 basic annotation. Images have been deposited in the Figshare platform under entries https://doi.org/10.6084/m9.figshare.22317835 (ref. 110; heart histology) and https://doi.org/10.6084/m9.figshare.22317922 (ref. 111; skull morphology). Flow cytometry source data is deposited in Figshare under entry https://doi.org/10.6084/m9.figshare.22320661 (ref. 112). Source data are provided with this paper.

## Code availability

No new software was developed during this study. All data analysis was carried out using existing software as described in the Online Methods for each specific experiment. Software employed in this study includes FASTQC v0.11.5, bcl2fastq v2.20.0.422, FastQ Screen v0.11.0, BBTools v37.99, ea-utils v1.05, HISAT2 v2.1.0, SAMtools v1.5, HTSeq-count v0.6.1, BWA v0.7.15, RSeQC v4.0.0, R 4.0.1, RStudio 2022.02.0, Bioconductor 3.11, the R packages DESeq2 v1.28.1, lmerTest v3.1-2, Hmisc v4.4-0,

ggplot2 v3.3.1, fgsea v1.14.0, survival v3.2-7, coxme v2.2-16, emmeans v1.5.1, and broom v0.7.9, CNV-seq v0.2-8, IPA (winter 2022 release), FlowJo v10, LEGENDplex v2021, Rotarod v1.4.1, Ethovision v8.5, FreezeScan v2.00, NRecon v1.7.4.6, Drishti v2.6.5, Photoshop 24.2.0, WinEDMA v2021, GraphPad Prism v8.0.1, Adobe Illustrator v24.1, Microsoft Word v16.70, Microsoft Excel v16.71 and Endnote v20.5.

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

## Acknowledgements

This work was supported primarily by NIH grants R01AI145988 (to K.D.S.) and R01AI150305 from the NIH INCLUDE Project (to J.M.E.). Additional funding was provided by NIH grants R01HL133230 (to K.S.), T32CA190216 (to K.A.W.), 2T32AR007411 (to K.A.W.), R01DE027689 (to K.A.F.), R01DE030864 (to K.A.F.), K02DE028572 (to K.A.F.), 5UL1TR002535, P30CA046934, the Linda Crnic Institute for Down Syndrome, the Global Down Syndrome Foundation, the Anna and John J. Sie Foundation, the Stowers Family Endowed Chair in Dental and Mineralized Tissue Research (to T.C.C.), the University of Colorado Department of Medicine Outstanding Early Career Scholar Program, the GI & Liver Innate Immune Program, the Human Immunology and Immunotherapy Initiative, the Gates Frontiers Fund, the University of Colorado School of Medicine and the Boettcher Foundation and Fast Grants. We are grateful to all research participants in the Human Trisome Project. We thank L. Bush for administrative support, M. Ahmed for technical advice, H. Potter for equipment, the Genetics Core Facility at National Jewish Health, the University of Colorado

Diabetes Research Center Cell and Tissue Analysis Core supported by NIH grant P30DK116073, T. Bruno at the University of Pittsburgh and I. D'Orso at the University of Texas Southwestern for logistical support, and all vivarium personnel.

## Author contributions

K.A.W. contributed to conceptualization, methodology, validation, investigation, resources, data curation, formal analysis, visualization, funding acquisition and manuscript writing and review. R.M. contributed to conceptualization, methodology, validation, investigation, resources and manuscript writing and review. J.B. contributed to investigation, resources and manuscript review. C.C. contributed to methodology, investigation, formal analysis and manuscript review. M.D.G. contributed to methodology, software, data curation, formal analysis, visualization and manuscript writing and review. K.D.T. contributed to methodology and manuscript review. N.P.E. and K.T.K. contributed to data curation, formal analysis, visualization and manuscript review. Z.A., P.A., H.D. and L.N.D. contributed to investigation and manuscript review. M.L. and N.B. contributed to methodology, investigation, and manuscript review. K.A.S. and D.T. contributed to investigation, resources, and manuscript review. K.P.S. and R.E.G. contributed to methodology, resources and manuscript review. S.K. contributed to formal analysis, visualization and manuscript review. R.D.A. and L.L.C. contributed to investigation, data curation and manuscript review. B.E.E. contributed to investigation, data curation and manuscript review. A.L.R. contributed to investigation, data curation, formal analysis, supervision and manuscript review. H.R.L. and E.C.B. contributed to investigation and manuscript review. K.A.F. contributed to methodology, investigation, funding acquisition and manuscript writing and review. D.J.O. contributed to investigation, manuscript writing and review. J.L.M. contributed to methodology, validation, resources, manuscript writing and review. K.S., T.C.C. and K.D.S., contributed to conceptualization, methodology, data curation, visualization, funding acquisition, supervision and manuscript writing and review. J.M.E. contributed to conceptualization, formal analysis, visualization, funding acquisition, supervision and manuscript writing and review.

## Competing interests

J.M.E. has provided consulting services for Eli Lilly Co. and Gilead Sciences Inc. and serves in the advisory board of Perha Pharmaceuticals. The remaining authors declare no competing interests.

## Additional information

**Extended data** is available for this paper at https://doi.org/10.1038/s41588-023-01399-7.

**Correspondence and requests for materials** should be addressed to Kelly D. Sullivan or Joaquin M. Espinosa.

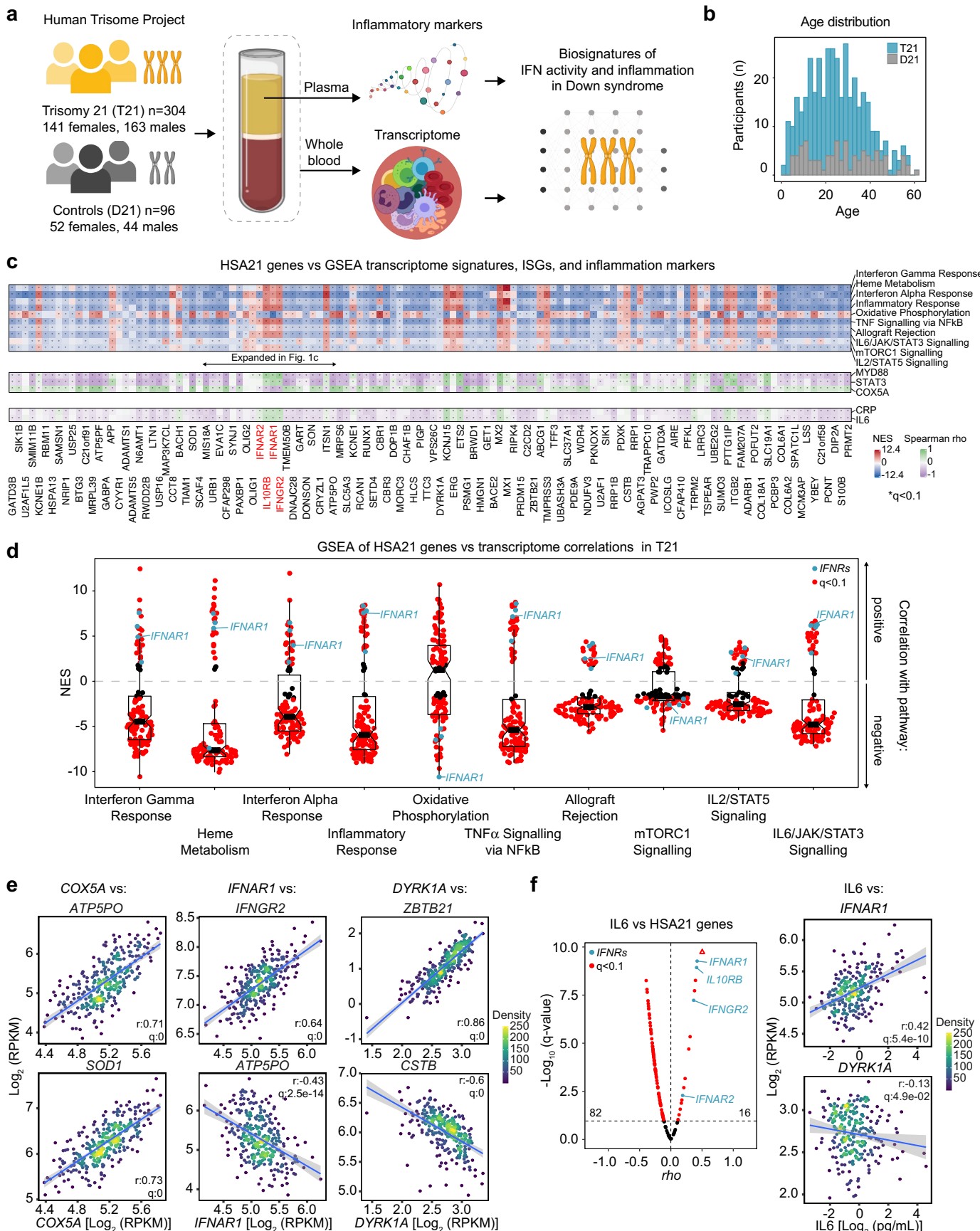

**Extended Data Fig. 1 | See next page for caption.**

**Extended Data Fig. 1 | Overexpression of *IFNRs* associates with inflammation in people with trisomy 21. a**, Schematic of biospecimen source and processing for datasets underlying Fig. 1 and Extended Data Fig. 1. **b**, Age distribution of participants by karyotype. Trisomy 21: T21, Down syndrome (n = 304, 163 male and 141 female). D21: euploid controls (n = 96, 44 male and 52 female). **c, Top:** Heatmap displaying the results of Gene Set Enrichment Analysis (GSEA) of ranked rho values from Spearman correlation analysis of mRNAs encoded on HSA21 versus all other mRNAs in the whole blood transcriptome of individuals with T21 (n = 304). NES: normalized enrichment score. Significance defined by GSEA as q < 0.1 after Benjamini-Hochberg correction. **Middle:** Spearman correlations between mRNAs encoded on HSA21 and the indicated differentially expressed genes encoded elsewhere in the genome among individuals with T21 (n = 304). **Bottom:** Spearman correlations between mRNAs encoded on HSA21 and the plasma levels of CRP and IL6 proteins in individuals with T21 (n = 249, 137 male and 112 female). q < 0.1 from Spearman with permutation test

and Benjamini-Hochberg correction for middle and bottom. **d**, Distribution of NES values of GSEA run on Spearman $\rho$ value matrices to assess correlations between expression of mRNAs encoded on HSA21 and top 10 gene sets elevated in the transcriptome of individuals with T21 (n = 304). Data are presented as sina plots where boxes represent interquartile ranges and medians, and notches approximate 95% confidence intervals. **e**, Correlations between indicated mRNAs among individuals with T21 (n = 304). **f**, Volcano plot of $\rho$ and q-values from Spearman correlations of IL6 protein abundance in plasma versus expression of HSA21 genes in the transcriptome of individuals with T21 (left) and scatter plots showing Spearman correlations between the indicated mRNAs and IL6 (right) (n = 249). In **e**, **f**, statistical significance defined as q < 0.1 by Spearman correlations with permutation test and Benjamini-Hochberg correction. h. In all scatter plots, blue lines represent linear regression fits with 95% confidence intervals in gray. In the volcano plot in **f**, the triangle indicates q = 0.

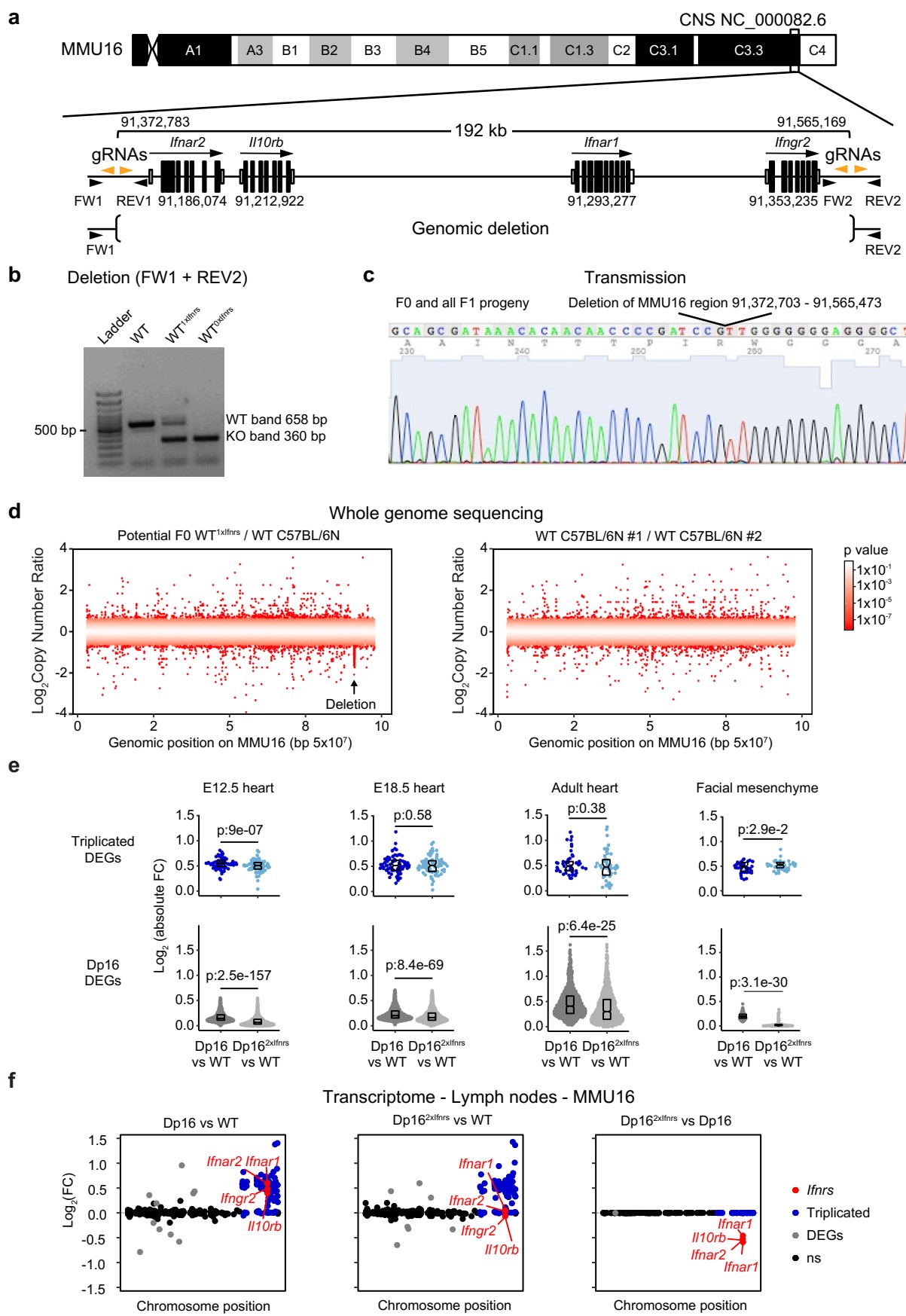

**Extended Data Fig. 2 | See next page for caption.**

**Extended Data Fig. 2 | Mouse model to determine if triplication of the *Ifnr* locus on mouse chromosome 16 is necessary for Down syndrome phenotypes. a**, Schematic of the mouse interferon receptor (*Ifnr*) gene locus and CRIPSR/Cas9 guide RNAs (gRNAs, orange arrowheads) used for deletion of 192 kb genomic locus on chromosome 16 (MMU16). Black arrows indicate forward (FW) and reverse (REV) primers. Location of genes based on GRCm38 reference genome is indicated on an ideogram of MMU16 cytogenetic regions colored according to Giemsa banding. **b**, Representative PCR from DNA of wild-type (WT) mouse, and mice heterozygous or homozygous for the expected knock-out (KO) deletion (WT$^{1xIfnrs}$ or WT$^{0xIfnrs}$, respectively). Gel image represents example of genotyping PCR used to characterize >50 pups derived from modified zygotes. This approach screened 5113 different descendants to date from a single heterozygous founder (F0). **c**, Representative Sanger sequencing of the single modified allele transmitted from a F0 WT$^{1xIfnrs}$ male to the first generation of progeny (F1) after inter-crossing with a WT female. **d**, Whole genome sequencing followed by copy number variant analysis of the F0 WT$^{1xIfnrs}$ with site of deletion on MMU16 denoted by arrow (left) that is absent

when two non-related C57BL/6 N WT mice are compared (right). Significance was determined by CNV-seq (*p < 0.1). **e**, Transcriptome analysis of hearts at embryonic day (E)12.5 - WT (n = 6, 2 male, 4 female), Dp16 (n = 6, 3 male, 3 female), Dp16$^{2xIfnrs}$ (n = 6, 5 male, 1 female), E18.5 - WT (n = 6, 4 male, 2 female), Dp16 (n = 6, 3 male, 3 female), Dp16$^{2xIfnrs}$ (n = 6, 3 male, 3 female), adult - WT (n = 5, 2 male, 3 female), Dp16 (n = 6, 3 male, 3 female), Dp16$^{2xIfnrs}$ (n = 5, 3 male, 2 female), and facial mesenchyme at E10.5 - WT (n = 3, 2 male, 1 female), Dp16 (n = 3, 2 male, 1 female), Dp16$^{2xIfnrs}$ (n = 3, 2 male, 1 female). Sina plots of mRNAs encoded on MMU16 triplicated in Dp16 excluding the four *Ifnrs* (top), and other differentially expressed genes (DEGs) across the genome for Dp16 versus WT (bottom). p-values calculated by two-sided paired Wilcoxon rank test. Data are presented as modified sina plots where boxes represent interquartile ranges and medians, and notches approximate 95% confidence intervals. **f**, Manhattan plots of mRNAs encoded on MMU16 differentially expressed by genotype in the mesenteric lymph nodes and colored as indicated. *q < 0.1 determined by DESeq2 with Benjamini-Hochberg correction.

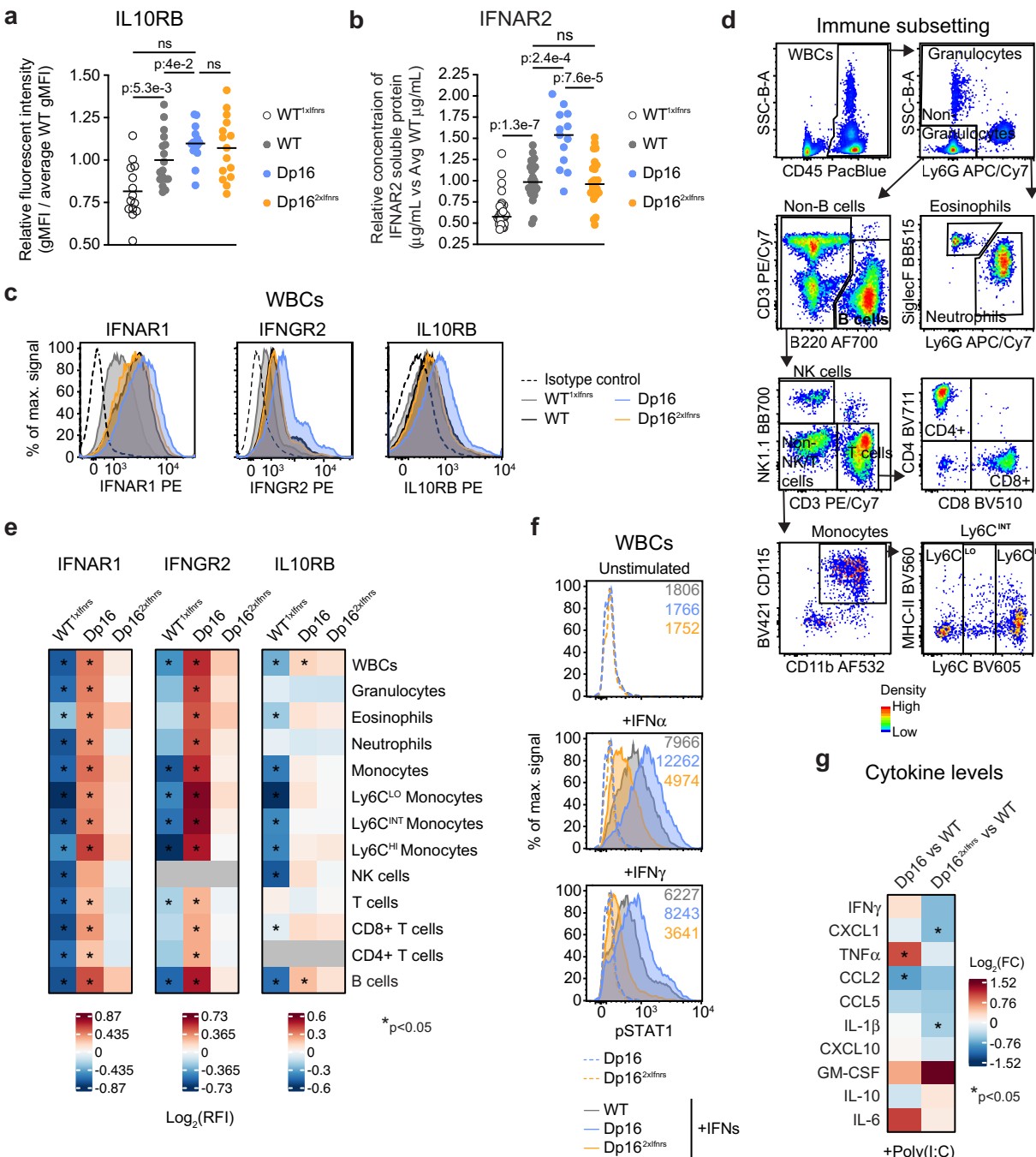

**Extended Data Fig. 3 | Triplication of the *Ifnr* locus drives increased IFNR protein expression and an aberrant antiviral response in a mouse model of Down syndrome. a**, Relative geometric mean fluorescent intensities (gMFIs) for IL10RB protein on CD45 + white blood cells (WBCs) from heterozygous *Ifnr* knockout mice (WT[1xIfnrs], n = 14, 3 male, 11 female), wild-type (WT, n = 17, 7 male, 10 female), Dp16 (n = 15, 7 male, 8 female), and Dp16[2xIfnrs] (n = 15, 8 male, 7 female), by flow cytometry. **b**, Soluble IFNAR2 protein by ELISA in the plasma of each biological replicate relative to the average WT value per experiment. The number of animals per group are the following: n = 28 WT[1xIfnrs], n = 30 WT; n = 13 Dp16, and n = 26 Dp16[2xIfnrs] (undocumented sex ratio). **c**, Example histogram for IFNR proteins on WBCs by flow cytometry. **d**, Pseudo-colored plot of gating strategy for immune subsets in whole blood by flow cytometry. Areas of high relative population density shown in red and orange, mid density in yellow, and low density in green and blue. **e**, IFNR protein expression on immune subsets isolated

as in (d) from CD45 + WBCs. Heatmap indicates relative fluorescent intensities (RFIs) for gMFIs of the indicated genotype over the WT average IFNR gMFI per experiment. Gray denotes IFNRs not detected above isotype background. The number of animals is the same as for panel a. **f**, Representative histograms with gMFIs indicated for phosphorylated STAT1 (pSTAT1) by flow cytometry of WBCs at baseline or after stimulation with IFNα (10,000 U/mL) or IFNγ (100 U/mL) for 30 minutes. **g**, Cytokine levels in the plasma of mice treated with poly(I:C). Heatmap indicates log₂ fold-change (FC) of cytokine protein in plasma of the indicated cohort +poly(I:C) relative to the WT poly(I:C) cohort after 3 days of the poly(I:C) regimen. WT poly(I:C) (n = 7, 3 male, 4 female), Dp16 poly(I:C) (n = 6, 5 male, 1 female), and Dp16[2xIfnrs] poly(I:C) (n = 6, 3 male, 3 female). **In a-b**, each dot represents an independent biological animal replicate with the mean indicated with a dash. For a-b, e, and g, significance (*p < 0.05) and exact p-values were determined by a two-sided Mann Whitney test.

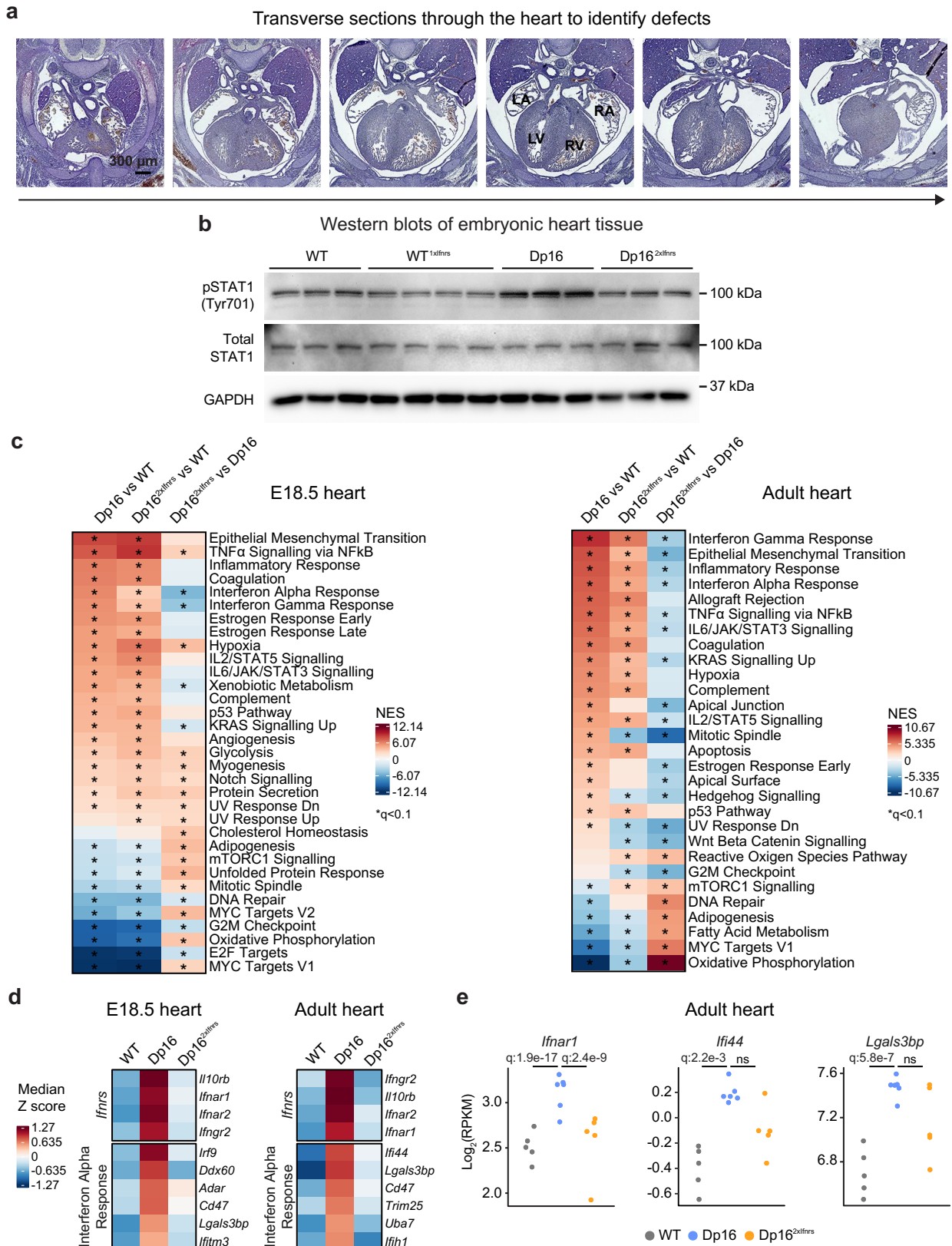

**a** Transverse sections through the heart to identify defects

**b** Western blots of embryonic heart tissue

**c** E18.5 heart    Adult heart

**d** E18.5 heart    Adult heart

**e** Adult heart

**Extended Data Fig. 4 | See next page for caption.**

**Extended Data Fig. 4 | Correction of *Ifnr* locus copy number prevents embryonic heart malformations in a mouse model of Down syndrome.**
**a**, Representative images of haematoxylin and eosin stained serial sections through entire murine hearts at embryonic day (E)15.5. Serial sections were cut through the entire region of the developing heart with transverse directionality indicated by arrow. Images represent a total of 58 formalin fixed paraffin embedded embryos that were processed and analyzed across four independent experiment batches; R: right, L: left, A: atrium, V: ventricle. WT (n = 24, 12 male, 12 female), Dp16 (n = 18, 9 male, 7 female, 2 undocumented sex), and Dp16$^{2xIfnrs}$ (n = 16, 4 male, 12 female). **b**, Western blot analysis of total and phosphorylated STAT1 at tyrosine 701 (pSTAT1) protein from developing hearts at E15.5 of wild-type (WT), heterozygous *Ifnr* knockout mice (WT$^{1xIfnrs}$), Dp16, and Dp16$^{2xIfnrs}$

animals, where n = 3/4/3/3 per group (undocumented sex ratio), respectively.
**c**, Heatmaps displaying Normalized Enrichment Scores (NES) from Gene Set Enrichment Analysis (GSEA) of transcriptome changes in E18.5 and adult heart tissue, sorted by NES for Dp16 versus WT. E18.5 - WT (n = 6, 4 male, 2 female), Dp16 (n = 6, 3 male, 3 female), Dp16$^{2xIfnrs}$ (n = 6, 3 male, 3 female), Adult - WT (n = 5, 2 male, 3 female), Dp16 (n = 6, 3 male, 3 female), Dp16$^{2xIfnrs}$ (n = 5, 3 male, 2 female); asterisks indicate q < 0.1 defined by GSEA after Benjamini-Hochberg correction.
**d**, Heatmaps displaying median RPKM expression Z-scores per genotype for example genes from gene sets dysregulated in E12.5 and adult heart tissue.
**e**, Sina plots displaying expression levels in adult heart tissue for example genes. q-values determined by DESeq2 with significance set at q < 0.1 after Benjamini-Hochberg correction. In d-e, sample sizes are as described in c.

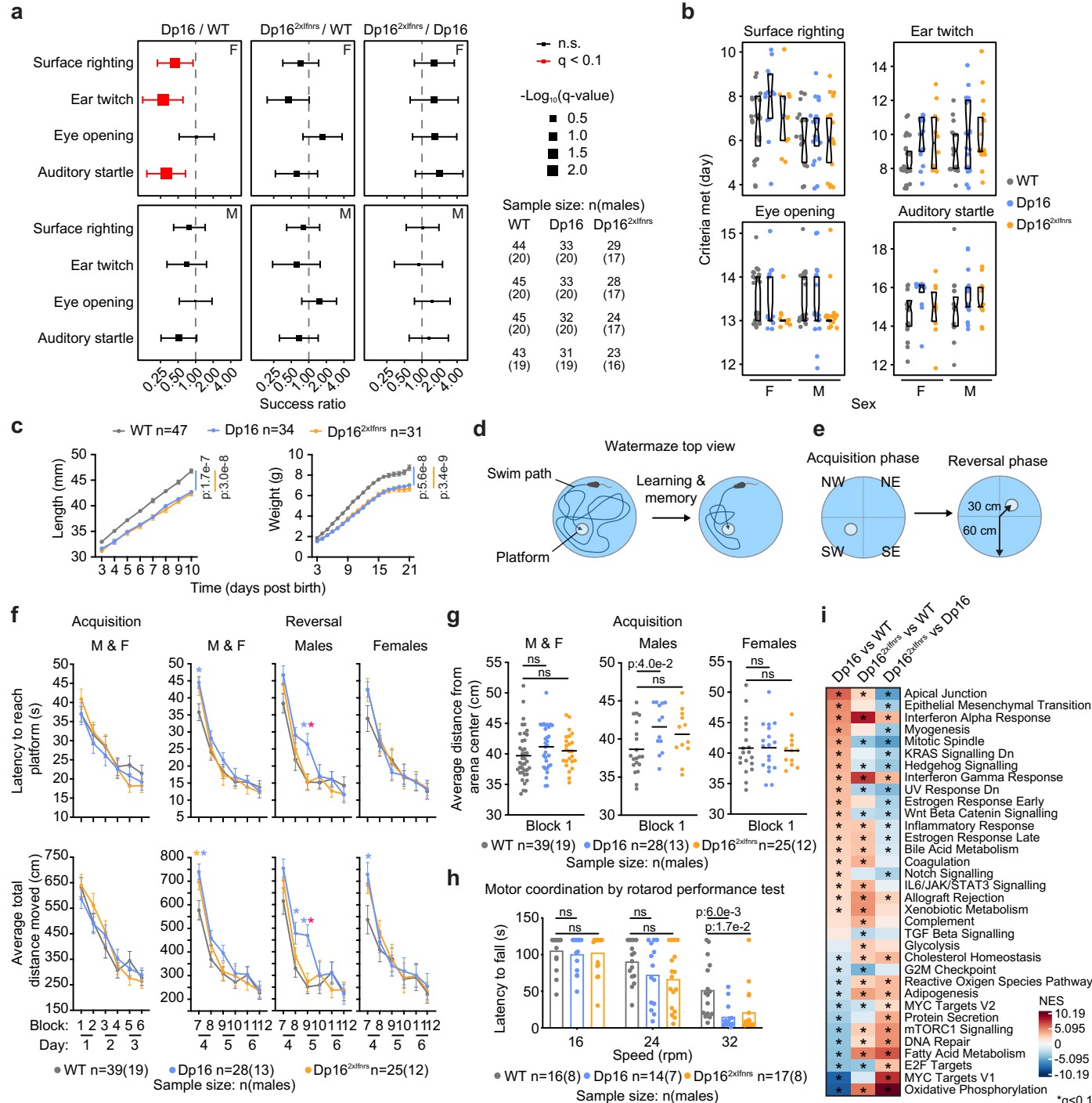

**Extended Data Fig. 5 | Cognition and behavior are improved with corrected gene dosage of the *Ifnr* locus in a mouse model of Down syndrome. a**, Odds ratio plots for developmental milestone achievement in neonates as assessed by mixed effects Cox regression for the indicated pairwise comparisons between Dp16, Dp16²ˣᴵᶠⁿʳˢ, and wild-type (WT) animals, separated by sex (Female-F/Male-M), with adjustment for litter (random). Square points represent 'success' ratios with size proportional to -log₁₀(q), error bars indicate 95% confidence intervals, red indicates q < 0.1 after Benjamini-Hochberg correction, dashed lines indicate odds ratio=1. **b**, Sina plots showing distributions for day of developmental milestone achievement, with boxes indicating interquartile ranges and medians, and notches approximating 95% confidence intervals; sample sizes shown in panel a. **c**, Growth curves for neonates assessed in a-b. Data are represented as means ± SEM. p-values determined by repeated measures two-way ANOVA. **d-e**, Schematics of Morris water maze (MWM). Mice navigate to a hidden platform more efficiently over time (d). Swim sessions are divided into two phases where the platform is in opposite quadrants labeled by intercardinal

directions (e). **f**, Duration (top) and total distance of path traveled (bottom) until platform escape during MWM for all animals or separated by sex; data are represented as means ± SEM, with significance determined by two-way repeated measures ANOVA with Tukey's HSD test; asterisks indicate p < 0.05 and are colored by comparison: Dp16 versus WT (blue), Dp16²ˣᴵᶠⁿʳˢ versus WT (orange), Dp16 versus Dp16²ˣᴵᶠⁿʳˢ (red). See Source Data Extended Data Fig. 5 for all p-values. **g**, Distributions of average distance of mice from MWM center; horizontal dashes represent means. WT (n = 39, 19 male, 20 female), Dp16 (n = 28, 13 male, 15 female), and Dp16²ˣᴵᶠⁿʳˢ (n = 25, 12 male, 13 female). **h**, Time until fall from a rotating rod. WT (n = 16, 8 male, 8 female), Dp16 (n = 14, 7 male, 7 female), and Dp16²ˣᴵᶠⁿʳˢ (n = 17, 8 male, 9 female). For g-h, p-values were determined by one-way ANOVA with Tukey's HSD test. **i**, Heatmap displaying Normalized Enrichment Scores (NES) from Gene Set Enrichment Analysis (GSEA) of transcriptome changes in adult brains, sorted by Dp16 versus WT (n = 6, 2 male, 4 female), Dp16 (n = 5, 2 male, 3 female), and Dp16²ˣᴵᶠⁿʳˢ (n = 7, 4 male, 3 female); asterisks indicate q < 0.1 by GSEA after Benjamini-Hochberg correction.

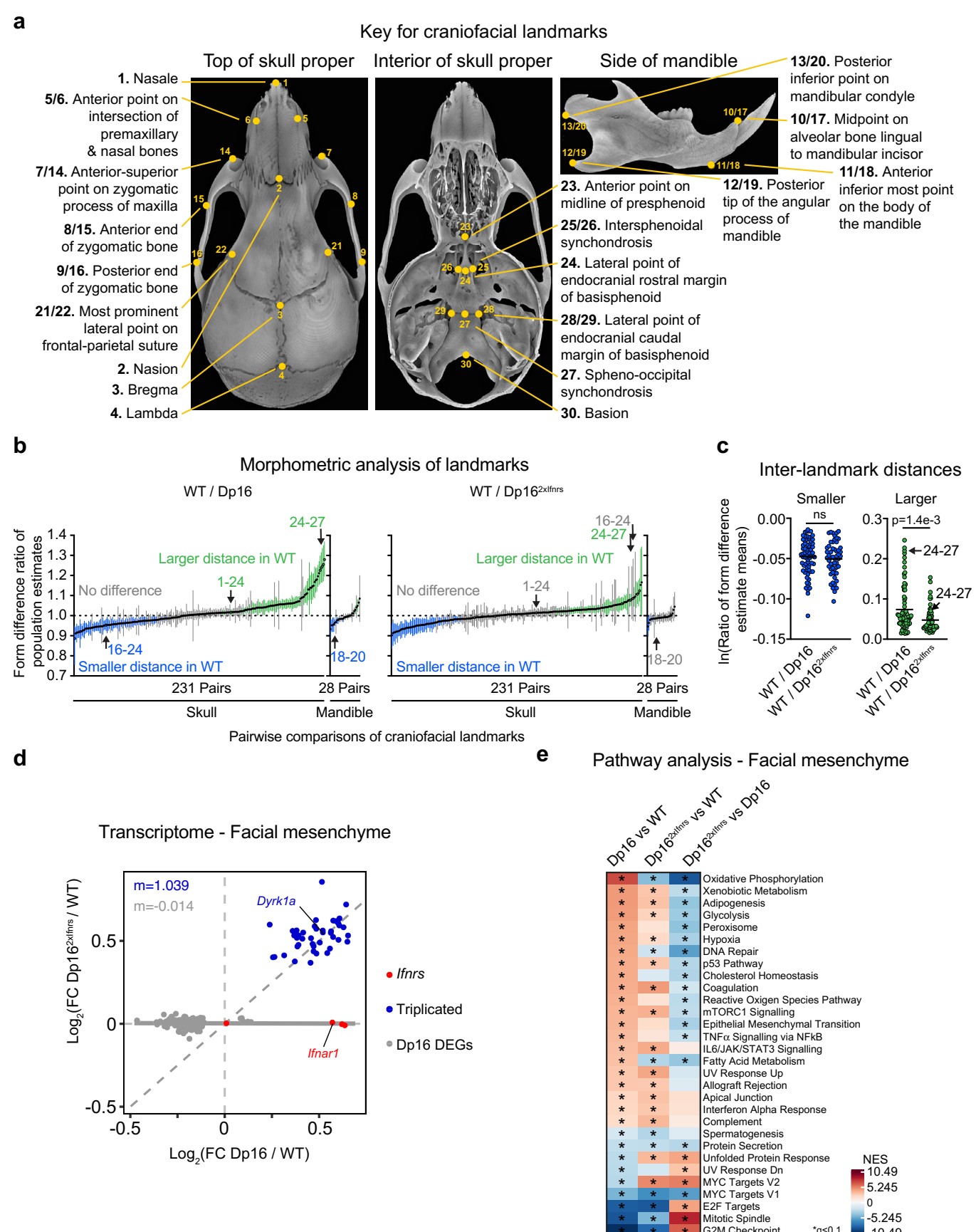

**a** Key for craniofacial landmarks

**1.** Nasale
**5/6.** Anterior point on intersection of premaxillary & nasal bones
**7/14.** Anterior-superior point on zygomatic process of maxilla
**8/15.** Anterior end of zygomatic bone
**9/16.** Posterior end of zygomatic bone
**21/22.** Most prominent lateral point on frontal-parietal suture
**2.** Nasion
**3.** Bregma
**4.** Lambda

Top of skull proper
Interior of skull proper
Side of mandible

**23.** Anterior point on midline of presphenoid
**25/26.** Intersphenoidal synchondrosis
**24.** Lateral point of endocranial rostral margin of basisphenoid
**28/29.** Lateral point of endocranial caudal margin of basisphenoid
**27.** Spheno-occipital synchondrosis
**30.** Basion

**13/20.** Posterior inferior point on mandibular condyle
**10/17.** Midpoint on alveolar bone lingual to mandibular incisor
**11/18.** Anterior inferior most point on the body of the mandible
**12/19.** Posterior tip of the angular process of mandible

**b** Morphometric analysis of landmarks

**c** Inter-landmark distances

**d** Transcriptome - Facial mesenchyme

**e** Pathway analysis - Facial mesenchyme

**Extended Data Fig. 6 | See next page for caption.**

**Extended Data Fig. 6 | Triplication of the *Ifnr* locus exacerbates craniofacial morphometric differences in a mouse model of Down syndrome.**
**a**, Representative micro-computed tomography (μCT) reconstructions of a wild-type (WT) mouse skull. Landmarks are indicated on dorsal views of the outer skull proper (left) and interior cranial base (center) as well as on a lateral view of the hemi-mandible (right). **b**, Form difference ratio of inter-landmark distances for the skull proper and mandible by Euclidean Distance Matrix Analysis for cohorts of Dp16 and Dp16$^{2xIfnrs}$ mice compared to WT controls, followed by bootstrapping 10,000x. The precise number of animals per group prior to bootstrapping are WT (n = 7, 2 male, 5 female), Dp16 (n = 7, 4 male, 3 female), and Dp16$^{2xIfnrs}$ (n = 6, 4 male, 2 female). Data are represented as mean population estimates (black dots) with 95% confidence intervals (lines) colored according to differences versus WT where green represents larger inter-landmark distance, blue represents smaller inter-landmark distance, and gray represents confidence intervals that intersect 1 (that is, no difference). **c**, Distributions of form difference ratios (natural log (ln)-transformed) for all mean population estimates of inter-landmark distances

on the skull proper and mandible shown in panel (b) that were smaller (blue) or larger (green) in WT versus Dp16 or versus Dp16$^{2xIfnrs}$ (smaller, blue, or larger, green). WT (n = 7, 2 male, 5 female), Dp16 (n = 7, 4 male, 3 female), and Dp16$^{2xIfnrs}$ (n = 6, 4 male, 2 female). p-values determined by two-sided Mann-Whitney test with significance set at p < 0.05. **d**, Scatter plots comparing mRNA fold-changes for Dp16 differentially expressed genes (DEGs) in facial mesenchyme tissue from E10.5 mice for Dp16/WT and Dp16$^{2xIfnrs}$/WT, with *Ifnrs* highlighted in red, Dp16 triplicated genes in blue, non-triplicated Dp16 DEGs in gray, and slope (m) colored accordingly; solid gray lines represent linear fits for the non-triplicated Dp16 DEGs. WT (n = 3, 2 male, 1 female), Dp16 (n = 3, 2 male, 1 female), Dp16$^{2xIfnrs}$ (n = 3, 2 male, 1 female). **e**, Heatmap displaying Normalized Enrichment Scores (NES) from Gene Set Enrichment Analysis (GSEA) of transcriptome changes in E10.5 facial mesenchyme for indicated comparisons, sorted by the for Dp16 versus WT comparison; asterisks indicate q < 0.1 by GSEA after Benjamini-Hochberg correction.

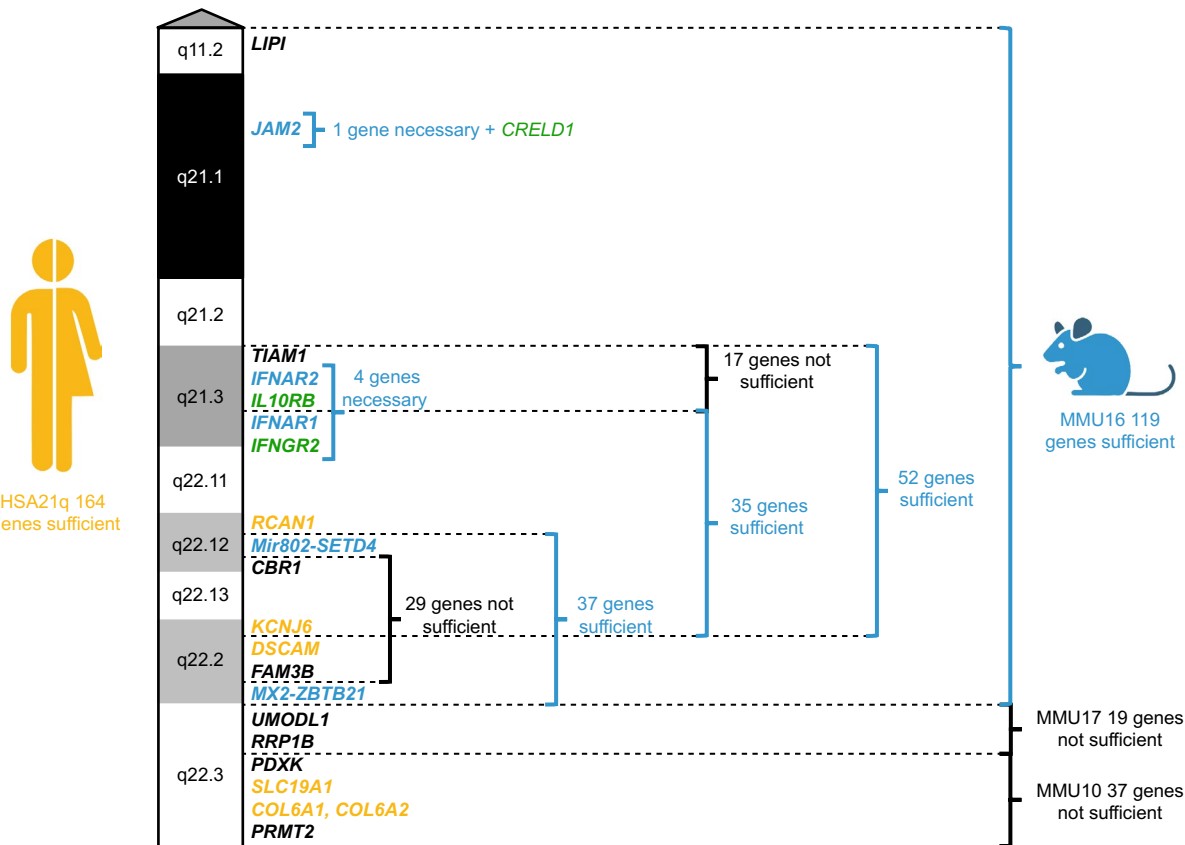

**Extended Data Fig. 7 | Overview of experimental evidence regarding the genetic basis of heart defects in mouse models of Down syndrome.** Diagram depicts genetic variants on human chromosome 21 (HSA21) that may contribute to risk of congenital heart defects (CHDs) in humans with trisomy 21 (T21, yellow), genes with functional evidence whose triplication is necessary or sufficient to increase incidence of CHDs in mouse models of Down syndrome (DS, blue), and genes with supporting evidence in both humans and mouse models of DS (green)[3,16,23,24,57–59]. Relative cytogenetic locations and number of bolded protein-coding genes are indicated along ideogram of the q arm of HSA21 colored according to Giemsa banding. Sources supporting this summary overview is provided in Source Data Extended Data Fig. 7.

# Reporting Summary

## Statistics

For all statistical analyses, confirm that the following items are present in the figure legend, table legend, main text, or Methods section.

| n/a | Confirmed | |
|---|---|---|
| ☐ | ☒ | The exact sample size (*n*) for each experimental group/condition, given as a discrete number and unit of measurement |
| ☐ | ☒ | A statement on whether measurements were taken from distinct samples or whether the same sample was measured repeatedly |
| ☐ | ☒ | The statistical test(s) used AND whether they are one- or two-sided *Only common tests should be described solely by name; describe more complex techniques in the Methods section.* |
| ☐ | ☒ | A description of all covariates tested |
| ☐ | ☒ | A description of any assumptions or corrections, such as tests of normality and adjustment for multiple comparisons |
| ☐ | ☒ | A full description of the statistical parameters including central tendency (e.g. means) or other basic estimates (e.g. regression coefficient) AND variation (e.g. standard deviation) or associated estimates of uncertainty (e.g. confidence intervals) |
| ☐ | ☒ | For null hypothesis testing, the test statistic (e.g. *F*, *t*, *r*) with confidence intervals, effect sizes, degrees of freedom and *P* value noted *Give P values as exact values whenever suitable.* |
| ☒ | ☐ | For Bayesian analysis, information on the choice of priors and Markov chain Monte Carlo settings |
| ☒ | ☐ | For hierarchical and complex designs, identification of the appropriate level for tests and full reporting of outcomes |
| ☐ | ☒ | Estimates of effect sizes (e.g. Cohen's *d*, Pearson's *r*), indicating how they were calculated |

*Our web collection on statistics for biologists contains articles on many of the points above.*

## Software and code

Policy information about availability of computer code

| | |
|---|---|
| Data collection | RNA-seq and murine whole genome sequencing data were collected with an Illumina NovaSeq 6000 instrument. Inflammatory marker data were collected with MESO QuickPlex SQ 120 instrument. Sanger sequencing was done using a 3730xl DNA Analyzer (ThermoFisher Scientific). Flow cytometry data were collected with a Cytek® Aurora spectral flow-cytometer and a Accuri C6 flow-cytometer. ELISA data were collected on a Synergy H4 Hybrid Multi-Mode Microplate Reader (BioTek). Heart histology data was collected with a Keyence BZ-X710 All-in-One Fluorescence Microscope. Rotarod data were collected with Rotarod v1.4.1 software (MED Associates, Inc.). Morris water maze data was collected using the video tracking system Ethovision v8.5 (Noldus). Context fear conditioning data were collected with FreezeScan v2.00 (Clever Sys. Inc.). Micro-computed tomography scan data were reconstructed to multiplanar slice data using NRecon v1.7.4.6 (Bruker). Coordinates for craniofacial landmarks were defined with Drishti v2.6.5. |
| Data analysis | No new software was developed during this study. All data analysis was carried out using existing software as described in the Online Methods for each specific experiment. Software employed in this study includes FASTQC v0.11.5, bcl2fastq v2.20.0.422, FastQ Screen v0.11.0, BBTools v37.99, ea-utils v1.05, HISAT2 v2.1.0, SAMtools v1.5, HTSeq-count v0.6.1, BWA v0.7.15, RSeQC v4.0.0, R 4.0.1, RStudio 2022.02.0, Bioconductor 3.11, the R packages DESeq2 v1.28.1, lmerTest v3.1-2, Hmisc v4.4-0, ggplot2 v3.3.1, fgsea v1.14.0, survival v3.2-7, coxme v2.2-16, emmeans v1.5.1, and broom v0.7.9, CNV-seq v0.2-8, IPA (winter 2022 release), FlowJo v10, LEGENDplex v2021, Rotarod v1.4.1, Ethovision v8.5, FreezeScan v2.00, NRecon v1.7.4.6, Drishti v2.6.5, Photoshop 24.2.0, WinEDMA v2021, GraphPad Prism v8.0.1, Adobe Illustrator v24.1, Microsoft Word v16.70, Microsoft Excel v16.71, and Endnote v20.5. |

For manuscripts utilizing custom algorithms or software that are central to the research but not yet described in published literature, software must be made available to editors and reviewers. We strongly encourage code deposition in a community repository (e.g. GitHub). See the Nature Portfolio guidelines for submitting code & software for further information.

# Data

Policy information about availability of data

All manuscripts must include a data availability statement. This statement should provide the following information, where applicable:

- Accession codes, unique identifiers, or web links for publicly available datasets
- A description of any restrictions on data availability
- For clinical datasets or third party data, please ensure that the statement adheres to our policy

The minimum dataset required to interpret, verify, and extend the research in this article is made available in the accompanying Source Data files and through public repositories. Mouse WGS data are deposited in the National Center for Biotechnology Information Sequence Read Archive (NCBI-SRA) under BioProject ID PRJNA776534. Human RNA-seq data generated by the Crnic Institute Human Trisome Project are deposited in the Gene Expression Omnibus (GEO) with accession number GSE190125 and also available through the INCLUDE Data Hub (https://portal.includedcc.org/). Human demographics and clinical source metadata is also available through the INCLUDE Data Hub. Murine RNA-seq data was deposited in GEO with the following accession numbers: GSE218883: adult mouse heart tissue; GSE218885: adult mouse brain tissue; GSE218887: embryonic mouse facial mesenchyme tissue; GSE218888: embryonic E12.5 mouse heart tissue; GSE218889: embryonic E18.5 mouse heart tissue; GSE218890: adult mouse mesenteric lymph nodes. All other source data are provided in the Source Data files with this manuscript. Reference datasets employed in this study were mouse reference genome assembly GRCm38 (mm10) with Gencode vM24 basic annotation (https://www.gencodegenes.org/mouse/release_M24.html), and human reference genome assembly GRCh38 (hg38) with Gencode v33 basic annotation. Images have been deposited in the Figshare platform under entries 10.6084/m9.figshare.22317835 (heart histology) and 10.6084/m9.figshare.22317922 (skull morphology). Flow cytometry source data is deposited in Figshare under entry 10.6084/m9.figshare.22320661.

# Human research participants

Policy information about studies involving human research participants and Sex and Gender in Research.

| | |
|---|---|
| Reporting on sex and gender | This study reports data from 400 research participants, 304 with Down syndrome (trisomy 21) (141 female and 163 males) and 96 euploid controls (52 females and 44 males). |
| Population characteristics | This study reports whole blood transcriptome analysis and measurements of the inflammatory markers CRP and IL6 in plasma samples from research participants enrolled in the Crnic Institute Human Trisome Project (HTP). The cohort employed for this study includes 304 participants with Down syndrome (trisomy 21) and 94 euploid controls. In addition to karyotype (trisomy 21 (T21) versus euploid controls, D21), and sex (male, female) the other key co-variate used in the various analyses was age. The median age and interquartile range values are 27.6 years (14.4 - 38.9) for euploid controls and 23.2 years (15.2 - 32.2) for participants with trisomy 21. |
| Recruitment | Recruitment into the Crnic Institute Human Trisome Project (HTP) took place at the University of Colorado Anschutz Medical Campus in Aurora, Colorado, USA, as well as multiple conferences in the USA. The study was promoted through scientific presentations at community conferences, IRB-approved flyers, the Human Trisome Project website (www.trisome.org), websites from affiliated organizations (e.g. the Global Down Syndrome Foundation), the DS-Connect registry, and social media. Participants received US$100 compensation per blood draw. Procedures were performed in accordance with IRB guidelines and regulations. Given the focus on recruitment of individuals with Down syndrome, a self-selection bias is unlikely, as recruitment is based on karyotype, which is confirmed from review of electronic health records. |
| Ethics oversight | Research participants were enrolled to the Crnic Institute Human Trisome Project Biobank (HTP) under a study protocol approved by the Colorado Multiple Institutional Review Board (IRB; COMIRB #15-2170). Procedures were performed in accordance with COMIRB guidelines and regulations. Written informed consent was obtained from participants who were cognitively able or by guardians of each participant. The study was conducted in accordance with the Declaration of Helsinki. |

Note that full information on the approval of the study protocol must also be provided in the manuscript.

# Field-specific reporting

Please select the one below that is the best fit for your research. If you are not sure, read the appropriate sections before making your selection.

☒ Life sciences ☐ Behavioural & social sciences ☐ Ecological, evolutionary & environmental sciences

For a reference copy of the document with all sections, see nature.com/documents/nr-reporting-summary-flat.pdf

# Life sciences study design

All studies must disclose on these points even when the disclosure is negative.

| | |
|---|---|
| Sample size | Sample size was determined a priori based on effect sizes of previous studies or by post hoc analyses to ensure >80% power was achieved to reduce type II error (e.g., heart histology, developmental milestone achievement, and cognition in adult mice). Key references we employed to determine the sample size of the human research studies are:<br>Sullivan, K. D. et al. Trisomy 21 consistently activates the interferon response. Elife 5 (2016). https://doi.org/10.7554/eLife.16220<br>Powers, R. K. et al. Trisomy 21 activates the kynurenine pathway via increased dosage of interferon receptors. Nat Commun 10, 4766 (2019). |

https://doi.org/10.1038/s41467-019-12739-9

Waugh, K. A. et al. Mass Cytometry Reveals Global Immune Remodeling with Multi-lineage Hypersensitivity to Type I Interferon in Down Syndrome. Cell reports 29, 1893-1908 e1894 (2019). https://doi.org/10.1016/j.celrep.2019.10.038

Araya, P. et al. Trisomy 21 dysregulates T cell lineages toward an autoimmunity-prone state associated with interferon hyperactivity. Proc Natl Acad Sci U S A 116, 24231-24241 (2019). https://doi.org/10.1073/pnas.1908129116

Sullivan, K. D. et al. Trisomy 21 causes changes in the circulating proteome indicative of chronic autoinflammation. Scientific reports 7, 14818 (2017). https://doi.org/10.1038/s41598-017-13858-3

**Data exclusions**

In high throughput measurements of CRP and IL6, extreme technical outliers were classified per-karyotype and per-analyte as measurements more than three times the interquartile range below or above the first and third quartiles, respectively, and excluded from further analysis. In measurements of cytokines by flow cytometry in murine samples, values are only shown for cytokines detected above background. All other data are included for all experiments.

**Replication**

All results presented are derived from multiple independent biological replicates, and in the case of mouse studies, also across multiple independent experiments. The numbers of human research participants, animals, and replicates is indicated for each experiment in the corresponding in the Methods and figure legends.

**Randomization**

For developmental milestones in neonates and cognitive assessment of adults, animals were assessed in a pseudorandom order. For all other murine experiments, animals were evaluated by litter as they became available, and randomized by taking key covariates into consideration including genotype, age, and sex.

**Blinding**

The investigators who sectioned embryos and performed histology analysis were blinded to embryo genotype to detect heart malformations. The investigators who assessed developmental milestones and cognitive phenotypes were blind to genotype. Craniofacial morphology was assessed by two investigators blinded to genotype. For all other experiments, data generation was done blinded to karyotype (e.g. for human studies of transcriptome and inflammatory markers) or mouse genotype (e.g., transcriptome analyses, flow cytometry) and the key variable was not unmasked until data analysis to define the effect of the variable.

# Reporting for specific materials, systems and methods

We require information from authors about some types of materials, experimental systems and methods used in many studies. Here, indicate whether each material, system or method listed is relevant to your study. If you are not sure if a list item applies to your research, read the appropriate section before selecting a response.

## Materials & experimental systems

| n/a | Involved in the study |
|-----|----------------------|
| ☐ | ☒ Antibodies |
| ☒ | ☐ Eukaryotic cell lines |
| ☒ | ☐ Palaeontology and archaeology |
| ☐ | ☒ Animals and other organisms |
| ☐ | ☒ Clinical data |
| ☒ | ☐ Dual use research of concern |

## Methods

| n/a | Involved in the study |
|-----|----------------------|
| ☒ | ☐ ChIP-seq |
| ☐ | ☒ Flow cytometry |
| ☒ | ☐ MRI-based neuroimaging |

## Antibodies

**Antibodies used**

Antibodies for flow cytometry:

Mouse CD4-BV711 (BioLegend #100550, clone RM4-5, RRID:AB_2562607, dilution 1:600); Mouse IA/IE-BV650 (BioLegend #107641, clone M5/114.15.2, RRID:AB_2565975, dilution 1:4800); Mouse Ly6C-BV605 (BioLegend #128036, clone HK1.4, RRID:AB_2562353, dilution 1:600); Mouse CD8-BV510, (BioLegend #100751, clone 53-6.7, RRID:AB_2561389, dilution 1:100); Mouse CD115-BV421, (BioLegend #135513, AFS98, RRID:AB_2562667, dilution 1:100); Mouse CD45-Pacific Blue (BioLegend #109820, clone 104, RRID:AB_492872, dilution 1:400); Mouse SiglecF-BB515 (BD Biosciences #564514, clone E50-2440, RRID:AB_2738833, dilution 1:100); Mouse CD11b-AF532 (Invitrogen #58-0112-82, clone M1/70, RRID:AB_2811905, dilution,1:800); Mouse NK1.1-BB700 (BD Biosciences #566502, clone PK136; RRID:AB_2744491, dilution 1:100); Mouse CD3-PE/Cy7 (BioLegend #100220, clone 17A2, RRID:AB_1732057, dilution 1:100); Mouse Ly6G-BV605 (BioLegend #127623, clone 1A8, RRID:AB_10645331, dilution 1:600); Mouse B220-AF700 (Invitrogen #56-0452-82, RA3-6B2, RRID:AB_891458, dilution 1:400); Mouse IFNAR1-PE (BioLegend #127312, MAR1-5A3, RRID:AB_2248800, dilution 1:50); Mouse IgG1k- PE (Invitrogen #MA1-10407, clone MOPC-21, RRID:AB_2536775, dilution 1:50); Mouse IFNGR2-PE (BioLegend #113603, clone MOB-47, RRID:AB_313560, dilution 1:10); Armenian hamster IgG-PE (BioLegend #400907, clone HTK888, RRID:AB_326593, dilution 1:10); Mouse IL10RB-PE, (Miltenyi #130-114-497, clone REA856, RRID:AB_2726668, dilution 1:10); REA control human IgG1-PE (Miltenyi #130-113-462, clone REA293, RRID:AB_2751113, dilution 1:10): phospho-STAT1 (Tyr701)-PE, (BD Biosciences #562069, clone 4a, RRID:AB_399855, dilution 1:10)

Antibodies for Western blots:

STAT1 (Cell Signaling Technology #9172, RRID:AB_2198300, dilution 1:1000); phospho-STAT1 (Tyr701) (Cell Signaling Technology #9167S, clone 58C6, RRID:AB_561284, dilution 1:1000); GAPDH (Thermo #AM4300, clone 6C5, RRID:AB_2536381, dilution 1:5000); Goat anti-mouse (Southern Biotech #1031-05, RRID:AB_2794307, dilution 1:2000); Goat anti-rabbit (Thermo #65-6120,

RRID:AB_2533967,dilution 1:2000)

Validation

All flow cytometry antibodies were titrated by authors with appropriate positive and negative internal cell population controls as well as isotype and fluorescent minus one (FMO) negative controls when geometric mean fluorescent intensities are reported. Additional validation information from the manufacturers is as follows:

For mouse CD4 - BioLegend Cat #100550, Clone RM4-5, RRID:AB_2562607; Mouse IA/IE - BioLegend Cat #107641, Clone M5/114.15.2, RRID:AB_2565975; Mouse Ly6C - BioLegend Cat #128036, Clone HK1.4, RRID:AB_2562353; Mouse CD8 - BioLegend Cat #10751, Clone 53-6.7, RRID:AB_2561389; Mouse CD115 - BioLegend Cat #135513, Clone AFS98, RRID:AB_2562667; Mouse CD45 - BioLegend Cat #109820, Clone 104, RRID:AB_492872; Mouse CD3 - BioLegend Cat #100220, Clone 17A2, RRID:AB_1732057; Mouse Ly6G - BioLegend Cat #127623, Clone 1A8, RRID:AB_10645331; Mouse IFNAR1 - BioLegend Cat #127312, Clone MAR1-5A3, RRID:AB_2248800; Mouse IFNGR2 - BioLegend Cat #113603, Clone OB-47, RRID:AB_313560; and armenian hamster IgG - BioLegend Cat #400907, Clone HTK888, each lot of these antibodies are quality control tested by the manufacturer by immunofluorescent staining with flow cytometry analysis.

For mouse SiglecF, BD Biosciences Cat #564514, Clone E50-2440, RRID:AB_2738833; Mouse NK1.1, BD Biosciences Cat #566502, Clone PK136, RRID:AB_2744491; phospho-STAT1 (Tyr701), and BD Biosciences Cat #562069, Clone 4a, RRID:AB_399855, these antibodies were validated by immunofluorescent staining with flow cytometry analysis versus isotype controls.

For mouse CD11b, Invitrogen Cat #58-0112-82, Clone M1/70, RRID:AB_2811905; Mouse B220, Invitrogen Cat #56-04522-82, Clone RA3-6B2, RRID:AB_891458; Mouse IgG1k, and Invitrogen Cat #MA1-10407, Clone MOPC-21, RRID:AB_2536775, these antibodies were validated by immunofluorescent staining with flow cytometry analysis versus isotype controls.

For, mouse IL10Rb, Miltenyi Cat #130-114-497, Clone REA856, RRID:AB_2726668; and REA control human IgG1, Miltenyi Cat #130-113-462, Clone REA293, RRID:AB_2751113, these antibodies were validated by immunofluorescent staining with flow cytometry analysis versus isotype controls.

All antibodies used for Western blots were validated by manufacturers as follows: STAT1, Cell Signaling Technology Cat# 9172, RRID:AB_2198300, was validated using STAT1 knockout cell lines; phospho-STAT1 (Tyr701), Cell Signaling Technology Cat# 9167, Clone, 58C6, RRID:AB_2198300, was validated using HeLa cells treated with IFN; GAPDH, Thermo Cat# AM4300, Clone 6C5, RRID:AB_2536381, was validated with Lysates from human, mouse, and rat lysates.

References supporting the validation statements: Harsha Krovi S, et al. 2020. Nat Commun. 4.790277778; Ferrere G, et al. 2021. JCI Insight; Miranda K, et al. 2022. iScience. 25:104994; Luo J, et al. 2022. J Nanobiotechnology. 20:228; Schloss MJ, et al. 2022. Nat Immunol. 23:605; Sandu I, et al. 2020. Nat Commun. 11:4454; Qi Z, et al. 2022. Nat Commun. 13:182; Zaman R, et al. 2021. Immunity; Banks DA, et al. 2019. J Immunol. 202:2348; Su Y, et al. 2022. J Hematol Oncol; Zenke S, et al. 2022. Nat Commun; Angata T, et al. 2001. J Biol Chem. 276(48); Arase N, et al. 1997. J Exp Med. 186(12); Perez OD, et al. 2005. Curr Protoc Cytom.; Verheijden S, et al. 2015. Glia. 63(9); Cheng N, et al. 2008. PLoS One. 3(4); Kang S, et al. 2016. PLoS One. 11(9); Spencer S.D. et al., 1998. J Exp Med. 187(4); Mehl JL, et al. 2022. iScience; Wu W-Y, et al. 2022. Molecules; Awad PN, et al. 2018. Cereb Cortex. 28(11).

# Animals and other research organisms

Policy information about studies involving animals; ARRIVE guidelines recommended for reporting animal research, and Sex and Gender in Research

Laboratory animals

The Dp(16Lipi-Zbtb21)1Yey/J mouse model of Down syndrome was originally purchased from The Jackson Laboratory (Cat# JAX:013530, RRID:IMSR_JAX013530) as well as gifted from Dr. Diana Bianchi's lab (National Institutes of Health, NIH) than intermixed and maintained on the C57BL/6J background (The Jackson Laboratory). C57BL/6NTac zygotes (Taconic) were used to generate the mutant mice. Experiments were done animals harvested at embryonic day (E)10.5 (facial mesenchyme ), E.12.5 (embryonic heart transcriptome), E15.5 (embryonic heart histology), E.18.5 (embryonic heart transcriptome), day 3-21 post-birth (developmental milestone and size/growth measurements), and ~4-6 months of age (cognitive testing, poly(I:C) treatment, transcriptome studies for mesenteric lymph nodes, adult heart), 5-9 months of age (flow cytometry), 3-9 months of age (ELISA), ~6-9 months of age (brain transcriptome), and 7-10 weeks of age (craniofacial morphology). Mice were housed separately by sex in groups of 1-5 mice/cage under a 14:10 light:dark cycle with controlled temperature and 35% humidity had ad libitum access to food (6% fat diet) and water.

Wild animals

None

Reporting on sex

We carefully report on sex and how data were analysed to analyze and/or account for potential differences by sex.

Field-collected samples

None

Ethics oversight

All animal experiments were approved by the Institutional Animal Care and Use Committee (IACUC) at the University of Colorado Anschutz Medical Campus, under Protocol #00111 and performed in accordance with the NIH guidelines for the care and use of animals in research.

Note that full information on the approval of the study protocol must also be provided in the manuscript.

# Clinical data

Policy information about clinical studies

All manuscripts should comply with the ICMJE guidelines for publication of clinical research and a completed CONSORT checklist must be included with all submissions.

| Clinical trial registration | Not applicable. |
|---|---|
| Study protocol | Research participants were enrolled to the Crnic Institute Human Trisome Project Biobank (HTP) under a study protocol approved by the Colorado Multiple Institutional Review Board (IRB; COMIRB #15-2170). |
| Data collection | Samples were collected through the Crnic Institute Human Trisome Project (HTP) at either the University of Colorado Anschutz Medical Campus in Aurora, Colorado, USA, or at multiple conferences in the USA. |
| Outcomes | The main goals of the clinical study were to determine baseline changes in circulating inflammatory markers of people with Down syndrome (DS, trisomy 21; T21) 1) in comparison to typical (D21) controls and 2) among individuals with T21. Please see each analysis section for details on RNA-seq and cytokine protein analyses. |

# Flow Cytometry

## Plots

Confirm that:

☒ The axis labels state the marker and fluorochrome used (e.g. CD4-FITC).

☒ The axis scales are clearly visible. Include numbers along axes only for bottom left plot of group (a 'group' is an analysis of identical markers).

☒ All plots are contour plots with outliers or pseudocolor plots.

☒ A numerical value for number of cells or percentage (with statistics) is provided.

## Methodology

| Sample preparation | Peripheral blood was collected from the submandibular vein of mice 17-25 weeks of age into tubes of lithium heparin (Sarstedt, Cat#41.1393.105) then stained as previously described with minor alterations. |
|---|---|
| Instrument | Cells were analyzed using a five laser Cytek® Aurora spectral flow-cytometer. |
| Software | Flow cytometry data was similarly analyzed with FlowJo Software (Becton, Dickinson & Company). |
| Cell population abundance | Cells were not sorted. Only the geometric mean fluorescent intensity is compared between the immune cell populations enriched by flow-cytometry according to Extended Data Fig. 3d instead of cell population abundance. |
| Gating strategy | Please see Extended Data Fig. 3d for the gating strategy. Briefly, white blood cells were gated by CD45+; Eosinophils and Neutrophils by scatter, Ly6G and SiglecF; B cells by B220; T cell subsets by CD3, CD4, and CD8; NK cells by NK1.1; Monocytes by CD115 and CD11b; then monocyte subsets by Ly6C. |

☒ Tick this box to confirm that a figure exemplifying the gating strategy is provided in the Supplementary Information.

