## [Peer Review File · Nature Genetics]

Peer Review Information

Manuscript Title: Interferon receptor locus contributes to hallmarks of Down syndrome in a mouse model

Corresponding author name(s): Dr Joaquin Espinosa Professor Kelly Sullivan

Reviewer Comments & Decisions:

Decision Letter, initial version:

7th Jun 2022

Dear Professor Espinosa,

I apologise for the delay in returning this decision to you. Thank you for your patience.

Your Article, "Interferon receptor gene dosage determines diverse hallmarks of Down syndrome" has now been seen by 4 referees (please note that Reviewers #3 and #4 reviewed together and uploaded the same report). You will see from their comments below that while they find your work of interest, some important points are raised. We are interested in the possibility of publishing your study in Nature Genetics, but would like to consider your response to these concerns in the form of a revised manuscript before we make a final decision on publication.

To guide the scope of the revisions, the editors discuss the referee reports in detail within the team, including with the chief editor, with a view to identifying key priorities that should be addressed in revision and sometimes overruling referee requests that are deemed beyond the scope of the current study. You'll see that the Reviewers have commented on the potential value of the mouse model you have generated, and many of their comments are designed to derive more mechanistic characterisation of said model. We agree that this would significantly boost the impact of the work so please pay careful consideration to these remarks. We would also ask that you address all the other comments, ideally experimentally where possible, or textually where necessary.

Please do not hesitate to get in touch if you would like to discuss these issues further.

We therefore invite you to revise your manuscript taking into account all reviewer and editor comments. Please highlight all changes in the manuscript text file. At this stage we will need you to upload a copy of the manuscript in MS Word .docx or similar editable format.

We are committed to providing a fair and constructive peer-review process. Do not hesitate to contact us if there are specific requests from the reviewers that you believe are technically impossible or

unlikely to yield a meaningful outcome.

*2) If you have not done so already please begin to revise your manuscript so that it conforms to our Article format instructions, available [here](http://www.nature.com/ng/authors/article_types/index.html). Refer also to any guidelines provided in this letter.

[redacted]

We are pleased to offer you a flexible timeframe for your revisions, providing that the novelty of the work is not compromised by other published studies in the meantime.

Sincerely,

Safia Danovi
Editor
Nature Genetics

Referee expertise:

Referee #1: immunogenomics incl. DS

Referee #2: gene dosage

Referee #3: DS genetics

Reviewer #4: reviewed with Reviewer #3

Reviewers' Comments:

Reviewer #1:

Remarks to the Author:

Waugh and colleagues evaluated the impact of interferon receptor gene dosage on various phenotypes of individuals with DS through use of a whole blood transcriptome analysis and a mouse model with segmental duplication. The authors showed the rescue of some Down syndrome phenotypes with normalization of IFNAR1, IFNAR2, IFNGR2, IL10RB region. While hyperresponsiveness to PolyI:C is perhaps not unexpected given its potent induction of IFN-I, the rescue observed in the hearts of mouse embryos has a very nice novelty. This study thus could be an important in defining the role of aberrant IFN signaling in T21. This could imply IFN as important in organ and cognition development.

Major Points

1. The findings reflecting the rescue of number of the heart defects are of the biggest potential impact. Authors cite clinical overlap of individuals with DS to those Type I IFNopathies (patients with AGS and others), however patients with Type I IFNopathies do not present with developmental heart signs or symptoms. Additionally, patients with STAT1 GOF which affects all four receptors triplicated in DS, Type I (both IFNAR and IFNLR) and type II IFNs (via IFNGR) also do not have developmental heart issues. Thus at least in other conditions activity of these four 4 genes is not sufficient for heart development problems to arise. Furthermore other 37 genes (not containing the 4 IFNR genes) are sufficient to cause increase in CHD as pointed out by the authors (In extended data Fig 7). But science is surprising, and if phenotype rescue is taken at face value, authors should perform significantly more characterization. Only a single time point and one RNAseq experiment (Figure 3E-J) is all data presented to make the case of 4 gene influence on the heart development.
2. I want to extent gratitude for the mouse model that was created in this paper. It will serve many of us in the future. However, the authors state that they knockout a 192 kb genomic segment on MMU16 encoding all four Ifnrs. For the point to be made that its only the function of the 4 genes, these 4

genes only need to be knocked out, and not 192 kb genomic segment. Perhaps tempering language that this segment is sufficient, and that maybe it IFN genes and maybe other factors is a better qualifying language to be used. In this 192kb segment, there are many noncoding RNAs (lncRNAs and miRNAs), which can all be important for development. Alternatively authors can attempt pharmacological rescue of the model via inhibition of 4 gene functions with anti IFN receptor antibodies, although this might be difficult in the developing embryo.

3. For Figure 3. G-J, the statistics between the WT, Dp16 and Dp162XIFNR mice is at best dynamic, and at times is only trending and not agreeing. Probably could be rectified by having more than a single experiment.

4. The authors mention a few differences in mice between males and females. For example, only Dp16 males had impairment of relearning in the MWM. Given that Down syndrome affects males and females at an equal frequency, can the authors provide a reason to why there are sex differences in mice?

5. Craniofacial abnormalities and the validation of craniofacial landmarks by two individuals who were blinded to the different genotypes is a great approach. Phenotype rescue when removing the 192 kb genomic segment is clear, but what is on this segment that is causing it is unclear.

6. The differences in cognitive deficits appear to be extremely mild. Associative learning appears more affected, while there is very little if no rescue in spatial learning. Any speculation as to why? Again, greater differences are noted in males than females in reversal phase (Figure 4C). Any speculation as to why?

Reviewer #2:

Remarks to the Author:

The study by Waugh et al. investigates the mechanism of interferon pathway hyperactivity in Down syndrome (TS21). The same lab has shown earlier that TS21 cells are hypersensitive to Interferon stimulation. The authors now expand on this firstly by performing a comprehensive transcriptome and plasma marker analysis on whole blood of TS21 patients with matched controls. Integrated data analysis reveals a strong association of elevated expression of Interferon Receptors (notably located on chromosome 21) with the DS signature and related activated pathways. These effects appear more directly connected to Interferon genes compared to other triplicated genes in DS. The authors then present a new mouse model of TS21, where they have deleted the Interferon-gene cluster and thereby rescue the elevated dosage caused by TS21. This mouse model is carefully characterized with regards to several phenotypic manifestations of DS patients. Restoration of IFNR gene dosage rescues from a malfunctioning immune system. It rescues embryonic heart defects. It ameliorates some of the cognitive features and craniofacial defects observed in TS21 and the DP16 model. Hence, several of the key physiological features of DS seem to be directly related to the INF-receptor dosage.

I really enjoyed reading this manuscript. The writing is excellent and I felt the interpretation of the data is balanced. The experiments, analyses and mouse model data are carefully performed/analyzed/characterized (e.g. a whole genome sequencing of the mouse model is provided and usually several approaches are presented before reaching a conclusion). The supplementary

information and raw data (GEO) are complete and informative. The human dataset on its own will be a valuable resource for the field. Overall, I think the results are of high relevance, since they link a “pro-inflammatory” phenotype caused by increased gene dosage of the IFNR to multiple phenotypic (systemic) alterations (which perhaps one could simplistically assume are caused by the increased gene dosage of several individual genes on chr21 mattering for a given phenotype/organ but not another).

I only have one major point, which I think the authors (given their strong transcriptome analysis in Figure 1 and Figure 3) can easily address: I missed some transcriptome characterizations of the mouse model beyond the heart (e.g. the brain relating to the cognitive experiments, perhaps the neural crest for the facial features or some immune cell type related to experiments presented in Figure 2). I think this would nicely make a point in that several tissues, organs and phenotypes – despite their very different developmental trajectories/origins – may be affected by the increased gene dosage in DP16 and accordingly the rescue in DP16-2xIFNR. This may also help to understand why certain features are completely restored, while others (e.g. cognitive) are only ameliorated, but do not fully rebalanced to WT.

Such datasets would also clarify the expression of DP16-2xIFNR in the different tissues compared to WT (two alleles on the two parental alleles). It is possible that because the two doses of the receptor are expressed in a slightly different way than in WT (excellent scheme in Extended Data Fig. 2e), the levels might not be exactly balanced (although the qPCR in Fig. 2b and the FACS in 2c would suggest this is the case). I would suggest that the authors provide a compiled qPCR and FACS, if possible, for all representative organs / cell-types studied (white blood cells, heart, brain, craniofacial (e.g. neural crest?)).

Minor Points:

The authors chose a q-value of 0.1 for their differential expression analysis. This seems fairly high in my view, as typically 0.05 or even 0.01 is chosen. Could the authors perhaps comment in methods of why this threshold was chosen and how many DE genes would be recovered at FDR of 0.05?

The methods paragraph for the Gene Set Enrichment Analysis and Spearman Correlation is relatively short and refers to packages and previously published information. Could the authors expand this in order to better explain the analysis for a non-expert?

The authors should double check the references for Figures in their text. E.g. on page 5 (“... rho values by GSEA (see Methods, Extended Data Fig. 1b)”) I think this refers to Extended Data Fig. 1c and following.

Could the authors provide a supplementary list which other 120 genes are triplicated in the Dp16 mouse model?

Do the authors have an explanation why the RFI of STAT1 in the stimulated condition for DP16xIFNR is lower than in WT (Extended Data Fig. 3f)?

Page 9 “... versus Dp16 transcriptomes revealed significant normalization of the gene signatures dysregulated in Dp16”. I am not sure what the authors mean with “normalization”, most likely because this a technical term referred to RNA-seq normalization. Perhaps the authors can rephrase this slightly?

Fig 3f: Can the authors, instead of the pathways, provide a simple heatmap of the 73 DEGs on the triplicated segment and how they change in the three conditions / genotypes analyzed? Is expression of other genes on DP16 (e.g. DYRK1A) restored by rebalancing IFNR to 2x? I would also like to see a heatmap of all 120 triplicated genes and their expression in the three conditions. Can they also provide a heatmap of the remaining DEGs (not on the triplicated segment)? Furthermore, can the volcano plot (3e) be provided for all comparisons (DP16/WT, DP162x/WT, DP162x/DP16) and the overlaps between this dataset (e.g. upset plot) be shown? I understand that some of these informations can be retrieved from the Supplementary Lists, but I feel these comparisons are key to the findings and should be provided as a Main Figure.

Since this a standard RNA-seq performed from mice, I would recommend choosing a FDR of 0.05.

Fig. 4a – Hard to distinguish the size of the square symbol that indicates the $-\log_{10}q$

Fig. 4e – is the comparison between DP16 and DP16-2xIFNR statistically significant?

The paragraph explaining the results of the craniofacial anomalies is a bit hard to understand for a non-specialist. Perhaps this paragraph can be rewritten for a more general audience, while the details can be provided in a supplementary note?

Earlier work has provided crucial roles for DYRK1A dosage in Trisomy21, especially with regards to the neurological and cognitive features of the disease. Perhaps in the nervous system, IFNR together with DYRK1A dosage are the key drivers for the DS phenotype? I noticed that in the heart RNA-seq, Dyrk1a expression was restored by deleting the third copy of Interferon receptors. Maybe this is not the case in the brain? I would encourage the authors to expand the results and discussion in that regard. p.15 / L7 – "... provides much needed mechanistic insight about the ..." – I would remove the word "mechanistic" here, because at the mechanistic / molecular level the link between IFNR and the craniofacial features still remains unclear.

Extended Data Fig 1a: The authors measured two inflammatory markers from plasma (CRP and IL-6). The scheme (and also one passage in the text stating "proteome") gives the impression many markers were tested. I would suggest adjusting the scheme and just write CRP/IL-6.

Extended Data Fig 1c: Too small font / labels.

Reviewer #3:

Remarks to the Author:

I read this manuscript with much interest because it addresses an important question in Down syndrome research, with biological and translational implications. Several prior studies, including from this group, have reported that individuals with Down syndrome have elevated inflammation and immune dysregulation, co-occurring with dysregulation of interferon-related genes. Since many things can influence these complex bodily systems, one could not assume that these effects are due to the presence of four interferon receptor genes on Chr21. Nonetheless, this is a compelling hypothesis, which Waugh et al. have now tested in a comprehensive study in a mouse model of Down syndrome. The heart of the study is the deletion of a cluster of four IFN receptor genes from a previously studied mouse model, Dp16, which carries a segmental duplication of ~120 genes orthologous to human

Chr21. While the study begins with pathway and cytokine analysis in human blood samples, the novel and impressive main part of the study is the engineering and analysis of trisomic mice corrected for triplication of the IFN receptor genes. The authors deleted a ~192kb segment containing the four interferon receptor genes in a WT mouse, then interbred these with the trisomic DP16 mouse, thereby correcting dosage for just those genes. They then assess and report that correcting IFN receptor gene dosage essentially prevents several major phenotypes of the DS mouse, some of which are remarkable and surprising.

The implications of these findings for humans remains to be directly established, but is plausible, and the findings in this DS mouse model alone are of high impact and significance. Overall this study appears to be comprehensive, well-done, and well-presented, although I have some significant comments or questions that should be addressed.

1. The first section analyzing the human blood transcriptome and cytokines is a reasonable addition; it is not novel nor strongly compelling, but provides context likely intended to strengthen the connection to human DS. The data serves to reinforce that inflammatory signatures are prevalent in human DS, although I didn't find the argument so compelling that the inflammatory pathways impacted correlate with the IFNR expression rather than other Chr21 genes. (ITSN1 seemed perhaps the strongest correlation.) It is also a bit counter-intuitive how one could discriminate this point clearly, given that expression of most Chr21 genes would, a priori, be elevated in concert due to trisomy. The authors emphasize the stronger correlation of IFNR mRNAs with inflammatory pathways in contrast to DYRK1A, a Chr21 gene which many studies report are connected to numerous phenotypes of DS. (It wasn't clear to me how the use of whole blood RNAseq here would complicate such comparisons.) It seems notable that seven of the top ten GSEA gene sets involve IFN signaling and inflammation, although it would be good for authors to indicate how many gene sets in blood could be considered related to IFNs and inflammation. Also, does triplication of six IFN-response genes (including MX1 and MX2) itself make it more likely that these pathways are near the top of the list? Or were the chr21 genes removed from the pathway analysis?

2. What is most impressive is the creation of and results from the mouse model with normalized copy number for Interferon receptor gene cluster, which they report prevents several major DS-related hallmarks in Dp16 mice. In this context, it is important that the comparison of WT, Dp16, Dp162xIFNRs mice, for all these assays, was done between mice/embryos for which there is essentially no difference in genetic backgrounds. In reading the methods, I surmise that this is the case, and that the comparisons were done between littermates with different genotypes, not pups for different litters....and not with differences in genetic backgrounds. This is important to the whole study, and assuming my reading is correct, the comparison of mice from the same litter and background should be briefly highlighted in the text. (Should there be any caveats to this, or complicating factors, this should be acknowledged and important to address in revision.)

3. In order to make the mice, the authors first deleted one copy of the IFNR gene cluster from WT mice....thereby creating mice monosomic for the IFNR genes....but there is no mention of anything about these monosomic mice. Their diagram indicates monosomic mice would represent 25% of F1 pups....so were these mice noticeably impacted in any way? This is interesting in general, but also has some relevance to the dosage sensitivity of IFNR genes....so there should be at least some brief summary about this. Do the authors think that monosomy for IFNR genes is more benign than trisomy for these genes?

4. The evidence that the IFNR correction prevents the heart defects is particularly striking.....even in embryos in which I would assume there are not immune insults? I was pleased to see that the analysis of embryos was done blind to their genotypes. However, a significant question comes up in Figure 3F. In transcriptomic analysis of E15.5 heart tissues, why are the differences in DP16/WT comparisons almost the same as in Dp162xIFNRs/WT comparisons? The IFNR-corrected Dp16 mice transcriptomes should be more similar to WT if so much is driven by the IFNR genes, but these results suggest that very similar pathways are dysregulated by other mechanisms independent of trisomy for the IFNR cluster. I understand that the Dp16 versus Dp162xIFNRs comparison shows some difference, but it is surprising and important that this data shows that mice with normalized IFNR gene dosage still have dysregulation of the key pathways that include: IFN alpha response, IFN gamma response, inflammatory response, IL6 Jak Stat3 signaling, IL2 Jak Stat2 signaling. These pathways seem central to this paper, but this point is not discussed. Is it a discrepancy that warrants substantial softening of the impression that inflammation is driven by or mostly by the IFNR gene cluster, or how do the authors explain it?

5. The evidence for correction or mitigation of "cognitive" deficits and craniofacial changes are both surprising, with important implications. Analysis of memory and learning deficits in mice can be tricky or potentially subjective, so I was pleased to see this analysis was also done blind, perhaps worth a mention in the text. While not required here, it is generally the case that it is good to have independent analysis and corroboration from an independent lab....which I assume will come in the future. If any of these findings are difficult to interpret, it would be fine to acknowledge any limitations in Discussion.

6. It is understandable that the emphasis is placed on phenotypes that have been ameliorated, but the authors should comment somewhat more about any phenotypes not mitigated by reducing IFNRs to disomy....this of interest too. For example, Fig 2e does not show significant difference in weight loss between Dp16 and Dp162xIFNRs on exposure to viral mimetic. Similarly in Figure 3h-j, there isn't a significant difference between the Dp162xIFNRs, DP16 and WT mice. Is reducing IFNR dosage sufficient to maintain homeostasis in pathways regulating interferon alpha response, oxidative phosphorylation and Myc targets. Finally, it is noted that motor co-ordination in Dp162xIFNRs and DP16 mice do not show much rescue. Please comment on other readily apparent phenotypes if present in Dp16, such as effects on longevity or size.

Minor comments:

1. Figure 1c in legend is mislabeled as "b"
2. Extended data figure 1e: There seems to be a red triangle on the figure above IFNAR1 gene which hasn't been explained in the legend. What does it imply for?

Reviewer #4:

Remarks to the Author:

I read this manuscript with much interest because it addresses an important question in Down syndrome research, with biological and translational implications. Several prior studies, including from this group, have reported that individuals with Down syndrome have elevated inflammation and

immune dysregulation, co-occurring with dysregulation of interferon-related genes. Since many things can influence these complex bodily systems, one could not assume that these effects are due to the presence of four interferon receptor genes on Chr21. Nonetheless, this is a compelling hypothesis, which Waugh et al. have now tested in a comprehensive study in a mouse model of Down syndrome. The heart of the study is the deletion of a cluster of four IFN receptor genes from a previously studied mouse model, Dp16, which carries a segmental duplication of ~120 genes orthologous to human Chr21. While the study begins with pathway and cytokine analysis in human blood samples, the novel and impressive main part of the study is the engineering and analysis of trisomic mice corrected for triplication of the IFN receptor genes. The authors deleted a ~192kb segment containing the four interferon receptor genes in a WT mouse, then interbred these with the trisomic DP16 mouse, thereby correcting dosage for just those genes. They then assess and report that correcting IFN receptor gene dosage essentially prevents several major phenotypes of the DS mouse, some of which are remarkable and surprising.

The implications of these findings for humans remains to be directly established, but is plausible, and the findings in this DS mouse model alone are of high impact and significance. Overall this study appears to be comprehensive, well-done, and well-presented, although I have some significant comments or questions that should be addressed.

1. The first section analyzing the human blood transcriptome and cytokines is a reasonable addition; it is not novel nor strongly compelling, but provides context likely intended to strengthen the connection to human DS. The data serves to reinforce that inflammatory signatures are prevalent in human DS, although I didn't find the argument so compelling that the inflammatory pathways impacted correlate with the IFNR expression rather than other Chr21 genes. (ITSN1 seemed perhaps the strongest correlation.) It is also a bit counter-intuitive how one could discriminate this point clearly, given that expression of most Chr21 genes would, a priori, be elevated in concert due to trisomy. The authors emphasize the stronger correlation of IFNR mRNAs with inflammatory pathways in contrast to DYRK1A, a Chr21 gene which many studies report are connected to numerous phenotypes of DS. (It wasn't clear to me how the use of whole blood RNAseq here would complicate such comparisons.) It seems notable that seven of the top ten GSEA gene sets involve IFN signaling and inflammation, although it would be good for authors to indicate how many gene sets in blood could be considered related to IFNs and inflammation. Also, does triplication of six IFN-response genes (including MX1 and MX2) itself make it more likely that these pathways are near the top of the list? Or were the chr21 genes removed from the pathway analysis?

2. What is most impressive is the creation of and results from the mouse model with normalized copy number for Interferon receptor gene cluster, which they report prevents several major DS-related hallmarks in Dp16 mice. In this context, it is important that the comparison of WT, Dp16, Dp16xIFNRs mice, for all these assays, was done between mice/embryos for which there is essentially no difference in genetic backgrounds. In reading the methods, I surmise that this is the case, and that the comparisons were done between littermates with different genotypes, not pups for different litters....and not with differences in genetic backgrounds. This is important to the whole study, and assuming my reading is correct, the comparison of mice from the same litter and background should be briefly highlighted in the text. (Should there be any caveats to this, or complicating factors, this should be acknowledged and important to address in revision.)

3. In order to make the mice, the authors first deleted one copy of the IFNR gene cluster from WT mice.....thereby creating mice monosomic for the IFNR genes....but there is no mention of anything

about these monosomic mice. Their diagram indicates monosomic mice would represent 25% of F1 pups....so were these mice noticeably impacted in any way? This is interesting in general, but also has some relevance to the dosage sensitivity of IFNR genes....so there should be at least some brief summary about this. Do the authors think that monosomy for IFNR genes is more benign than trisomy for these genes?

4. The evidence that the IFNR correction prevents the heart defects is particularly striking.....even in embryos in which I would assume there are not immune insults? I was pleased to see that the analysis of embryos was done blind to their genotypes. However, a significant question comes up in Figure 3F. In transcriptomic analysis of E15.5 heart tissues, why are the differences in DP16/WT comparisons almost the same as in Dp162xIFNRs/WT comparisons? The IFNR-corrected Dp16 mice transcriptomes should be more similar to WT if so much is driven by the IFNR genes, but these results suggest that very similar pathways are dysregulated by other mechanisms independent of trisomy for the IFNR cluster. I understand that the Dp16 versus Dp162xIFNRs comparison shows some difference, but it is surprising and important that this data shows that mice with normalized IFNR gene dosage still have dysregulation of the key pathways that include: IFN alpha response, IFN gamma response, inflammatory response, IL6 Jak Stat3 signaling, IL2 Jak Stat2 signaling. These pathways seem central to this paper, but this point is not discussed. Is it a discrepancy that warrants substantial softening of the impression that inflammation is driven by or mostly by the IFNR gene cluster, or how do the authors explain it?

5. The evidence for correction or mitigation of "cognitive" deficits and craniofacial changes are both surprising, with important implications. Analysis of memory and learning deficits in mice can be tricky or potentially subjective, so I was pleased to see this analysis was also done blind, perhaps worth a mention in the text. While not required here, it is generally the case that it is good to have independent analysis and corroboration from an independent lab....which I assume will come in the future. If any of these findings are difficult to interpret, it would be fine to acknowledge any limitations in Discussion.

6. It is understandable that the emphasis is placed on phenotypes that have been ameliorated, but the authors should comment somewhat more about any phenotypes not mitigated by reducing IFNRs to disomy....this of interest too. For example, Fig 2e does not show significant difference in weight loss between Dp16 and Dp162xIFNRs on exposure to viral mimetic. Similarly in Figure 3h-j, there isn't a significant difference between the Dp162xIFNRs, DP16 and WT mice. Is reducing IFNR dosage sufficient to maintain homeostasis in pathways regulating interferon alpha response, oxidative phosphorylation and Myc targets. Finally, it is noted that motor co-ordination in Dp162xIFNRs and DP16 mice do not show much rescue. Please comment on other readily apparent phenotypes if present in Dp16, such as effects on longevity or size.

Minor comments:

1. Figure 1c in legend is mislabeled as "b"
2. Extended data figure 1e: There seems to be a red triangle on the figure above IFNAR1 gene which hasn't been explained in the legend. What does it imply for?

Author Rebuttal to Initial comments
--

Response to Referees – Manuscript NG-A60025-T by Waugh et al.**Reviewer #1.**Remarks to the Author

Waugh and colleagues evaluated the impact of interferon receptor gene dosage on various phenotypes of individuals with DS through use of a whole blood transcriptome analysis and a mouse model with segmental duplication. The authors showed the rescue of some Down syndrome phenotypes with normalization of IFNAR1, IFNAR2, IFNGR2, IL10RB region. While hyperresponsiveness to PolyI:C is perhaps not unexpected given its potent induction of IFN-I, the rescue observed in the hearts of mouse embryos has a very nice novelty. This study thus could be an important in defining the role of aberrant IFN signalling in T21. This could imply IFN as important in organ and cognition development.

Major Points

1. The findings reflecting the rescue of number of the heart defects are of the biggest potential impact. Authors cite clinical overlap of individuals with DS to those Type I IFNopathies (patients with AGS and others), however patients with Type I IFNopathies do not present with developmental heart signs or symptoms. Additionally, patients with STAT1 GOF which affects all four receptors triplicated in DS, Type I (both IFNAR and IFNLR) and type II IFNs (via IFNGR) also do not have developmental heart issues. Thus at least in other conditions activity of these four 4 genes is not sufficient for heart development problems to arise. Furthermore other 37 genes (not containing the 4 IFNR genes) are sufficient to cause increase in CHD as pointed out by the authors (In extended data Fig 7). But science is surprising, and if phenotype rescue is taken at face value, authors should perform significantly more characterization. Only a single time point and one RNAseq experiment (Figure 3E-J) is all data presented to make the case of 4 gene influence on the heart development.

Response: We are grateful for the overall positive assessment and constructive critiques, which we address in the revised manuscript with new experiments, new analyses, and important revisions to the text.

First, this comment by Reviewer #1 prompted us to revise the text of the manuscript to summarize the current state of the field in terms of the impacts of immune dysregulation on heart development and to better place our findings in the greater context of the literature. As Reviewer #1 notes, not all genetic conditions leading to elevated IFN signalling or downstream JAK/STAT signalling are associated with increased incidence of congenital heart defects. However, there is increasing evidence supporting a pathogenic role for elevated IFN signalling in heart development and function.

For example, increased risk of cardiovascular malformations has been associated with a subset of interferonopathy patients bearing mutations in *ADAR*, *IFIH1* (encoding MDA5), and *DDX58* (encoding

RIG-I)¹. More specifically, patients carrying *ADAR*, *IFIH1*, and *DDX58* mutations were found to develop calcifying cardiac valvular disease¹⁻³, and aortic calcification is a recognized feature of Singleton-Merten syndrome caused by *IFIH1* and *DDX58* mutations^{2,3}. As described by Crow et al¹, '*individuals with ADAR-related type I interferonopathy may develop childhood-onset multivalvular stenosis and incompetence which can progress insidiously to symptomatic, and ultimately fatal, cardiac failure*'. These findings have led leaders in the interferonopathy field to recommend regular surveillance echocardiograms for early detection of valvular disease¹. Given these findings, monogenic interferonopathies are now recognized as one of several hereditary disorders of cardiovascular calcification⁴.

Prior to the concept of interferonopathies, the term "pseudo-TORCH (toxoplasmosis, rubella, cytomegalovirus and herpes) syndrome" was applied to identify this group of 'serologically negative' disorders that mimic congenital TORCH infections⁵. Importantly, maternal rubella virus and cytomegalovirus infections are widely accepted risk factors for heart defects in developing fetuses⁶. For example, the cardiac abnormality most frequently found in rubella syndrome is a combination of branch pulmonary artery stenosis and patent ductus arteriosus, with isolated branch pulmonary artery stenosis twice as common as isolated patent ductus arteriosus⁶. Although rubella has been mostly eradicated in the United States, other countries are still actively working to determine additional developmental defects affecting exposed fetuses, such as craniofacial malformations (e.g., microcephaly) that also occur in subsets of individuals with interferonopathies⁷. Clearly, the field has just begun to understand the commonalities between diverse interferonopathies and maternal viral infections that could elicit elevated IFN and JAK/STAT signalling *in utero*, with additional research being needed to decipher the mechanisms underlying the ensuing developmental defects as well as the incomplete penetrance of these phenotypes.

Therefore, guided by the Reviewer's comment, we have modified the discussion to highlight that diverse monogenic interferonopathies, autoimmune conditions (i.e., lupus), and maternal infections have been associated with increased risk of various heart malformations, albeit to variable degrees^{1,6,8}, while also recognizing that not all disorders of elevated JAK/STAT signalling present cardiovascular abnormalities, such as in the case of STAT1 gain-of-function⁹.

In conclusion, the decreased prevalence of heart malformations observed upon decreased dosage of the *Ifnr* locus in a mouse model of Down syndrome is therefore novel but not necessarily misaligned with the current literature.

Second, we agree with Reviewer #1 that we have not demonstrated 'sufficiency' but rather 'requirement' of *Ifnr* locus triplication for increased incidence of congenital heart disease in the Dp16 preclinical model of Down syndrome. Our data are in line with previous publications of genetic potentiators of increased prevalence of heart defects in both humans and mouse models of Down syndrome that are also highlighted in extended data Fig. 7, such as *Jam2* (encoded on chromosome 21), which is as a potentiator of *Creld1* mutations (*Creld1* is encoded on human chromosome 3)¹⁰.

We agree with Reviewer #1 that it is important to acknowledge that such data in humans and mice are sometimes at odds with the ‘sufficiency’ experiments described in **Fig. 7**. However, we chose not to elaborate much more on this topic in the Discussion, as many of the available negative data are not sufficiently powered but still cited in **Fig. 7** as they have passed peer review.

For example, the comprehensive study by Lana-Elola *et al.*, 2016¹¹ achieves >80% power to determine that the proportion of Dp1Tyb with heart defects (61.5% of n=39 embryos) differ from wildtype embryos (26.9% of n=26 embryos). The Dp1Tyb are comparable to Dp16 in gene content. The Dp1Tyb were then broken into several other models that have smaller segmental replications of this total region to test which gene segments are sufficient for increased incidence of heart defects. Again, nearly 80% power was achieved to determine that the proportion of Dp3Tyb, or mice containing 37 genes that do not include the 4 *Ifnr* genes, bearing heart defects (44.0% of n=25 embryos) differ from wild type controls (11.5% of n=26 embryos). However, only 36% power was achieved to determine that the Dp2Tyb, or mice containing a segmental duplication that bore the 4 *Ifnrs* and other genes, did not differ in proportion of heart defects (34.6% of n=26 embryos) compared to wildtype controls (12.5% of n=34.6). Therefore, the field has only achieved ~36% power to test the null hypothesis that there is no difference in Dp2Tyb and wild type controls, which is well below the recommended lowest power to reduce the likelihood of Type II error (i.e., a false negative).

Guided by this Reviewer comment, we have modified the Discussion to make amply clear that our results do not demonstrate sufficiency, only requirement, of *Ifnr* triplication, while also placing our results into the context of current literature.

Third, we are grateful for the guidance to perform more experiments in this chapter of the manuscript, which prompted us to complete transcriptome analyses of cardiac tissue at two different embryonic timepoints, E12.5 and E18.5, as well as adult heart tissues, and to complete a more thorough analysis of these datasets. We also employed a larger sample size (n=5-6 mice per genotype) to identify more differentially expressed genes (DEGs).

As described in the revised manuscript, at all three time points, heart tissue from Dp16 mice show overexpression of most of the triplicated genes, as well as global dysregulation of key signalling pathways, with significant attenuation of fold changes in expression of the DEGs outside the triplicated region in Dp16^{2xIfnrs}, most prominently at E12.5 (new **Extended Data Fig. 2e**, new **Fig. 4e**). Pathway analysis identified both common and unique gene signatures dysregulated in Dp16 and attenuated in Dp16^{2xIfnrs} at all three time points (new **Fig. 4f**, new **Extended Data Fig. 4c**). Consistently, Dp16 heart tissues show increased interferon alpha and gamma signalling concurrent with elevation of diverse inflammatory pathways, such IL2/STAT5 signalling, IL6/JAK/STAT3 signalling, Inflammatory response, and TNF α signalling (new **Fig. 4f**, new **Extended Data Fig. 4c**). Other conserved signatures across time points include elevated expression of genes involved in Epithelial to Mesenchymal Transition (EMT) and decreased expression of genes associated with cell proliferation (e.g., MYC targets, E2F targets) (new **Fig. 4f**, new **Extended Data Fig. 4c**). In Dp16^{2xIfnrs} mice, many, but certainly not all, of these gene

expression changes are rescued, with some variation across time points. For example, at E12.5, a critical time point for septation of the heart, Dp16^{2xIfnrs} mice show attenuated dysregulation of ISGs (e.g., *Irf9*, *Ifih1*), EMT genes (e.g., *Col4a2*, *Tgfb1*), and MYC target genes (e.g., *Rcf4*, *Srsf7*) (new Fig. 4f-g). The Reviewer may appreciate that one of the genes induced in Dp16 but 'attenuated' in Dp16^{2xIfnrs} is *Ifih1*, encoding *Mda5*, which has been found mutated (gain-of-function) in Aicardi-Goutières Syndrome and Singleton-Merten Syndrome.

Altogether, these results indicate that triplication of *Ifnrs* elicits a signalling cascade in the developing heart involving elevated JAK/STAT signalling, dysregulation of EMT processes, along with decreased cell growth and proliferation, all of which could contribute to heart malformations. These new results are described in the revised manuscript and all data made available to facilitate additional research.

2. I want to extend gratitude for the mouse model that was created in this paper. It will serve many of us in the future. However, the authors state that they knockout a 192 kb genomic segment on MMU16 encoding all four *Ifnrs*. For the point to be made that it's only the function of the 4 genes, these 4 genes only need to be knocked out, and not 192 kb genomic segment. Perhaps tempering language that this segment is sufficient, and that maybe it IFN genes and maybe other factors is a better qualifying language to be used. In this 192kb segment, there are many noncoding RNAs (LnRNAs and miRNAs), which can all be important for development. Alternatively, authors can attempt pharmacological rescue of the model via inhibition of 4 gene functions with anti-IFN receptor antibodies, although this might be difficult in the developing embryo.

Response: We thank Reviewer #1 for this comment and look forward to sharing the mouse model we created with the community. The Reviewer may be pleased to know that in the revised manuscript we describe a total of **six** transcriptome analyses of different tissues relevant to the phenotypes studied: mesenteric lymph nodes, three time points for heart tissue, neural crest-derived facial mesenchyme, and brain tissue. These additional transcriptome studies represent a much more thorough characterization of the mouse model created. Consistently, the four *Ifnrs* stand out as the only detected mRNAs overexpressed in Dp16 **and** fully restored to normal expression levels in the Dp16^{2xIfnr} strain across all tissues tested. In each of these experiments, we demonstrate the clean effect of the genome editing strategy employed, which was carefully designed to avoid disrupting expression of flanking genes. With all that being said, we fully agree with the Reviewer that deletion of the 192kb segment may affect non-coding RNA species not detected by our transcriptome analysis or *cis* regulatory regions (e.g., enhancers). Accordingly, we modified the text to conclude that rescued phenotypes can now be traced back to gene dosage of the "*Ifnr* locus" or "*Ifnr* gene cluster" encoded on Mmu16, and have made sure to clarify that our experiments demonstrate requirement, but not sufficiency, of triplication of this locus for increased penetrance and/or severity of diverse phenotypes in the in Dp16 preclinical model of Down syndrome.

We considered the use of anti-IFN receptor antibodies as suggested by the Reviewer but were not convinced of the feasibility of this study for a few reasons: 1) Need of a cocktail of four different

antibodies with clear 'neutralizing' activity, 2) Unclear access of these antibodies to key tissues (e.g., embryonic heart, facial mesenchyme, brain), and 3) Unclear timeline for successful intervention. Nevertheless, we are very interested in finding ways to translate our results into clinical applications. We feel that data presented herein set the necessary foundation to begin to pursue both preventative and therapeutic targeting of IFN signalling pathways in Down syndrome. We now elaborate in the Discussion on the value of this genetic rescue, and how future pharmacologic studies can now begin to investigate appropriate timing, dosage, and methods of normalizing (vs. completely inhibiting) IFN signalling in trisomy 21.

3. For Figure 3. G-J, the statistics between the WT, Dp16 and Dp162XIFNR mice is at best dynamic, and at times is only trending and not agreeing. Probably could be rectified by having more than a single experiment.

Response: Thanks for this comment, which we have addressed with the additional transcriptome analyses described above. Notably, the revised manuscript describes transcriptome analysis of six different biospecimens across the three genotypes, as well as much expanded bioinformatics analyses, which led us to create an entire new figure (new **Fig. 2**) as well as many other panels in previous figures and new supplemental files.

As described throughout the revised manuscript, the transcriptome analyses consistently highlighted overexpression of the triplicated genes in the Dp16 strain and rescue of *Ifnr* overexpression in Dp16^{2xIfnrs}, while largely preserving overexpression of other triplicated genes (new **Fig. 2c-f**, new **Extended Data Fig. 2e-f**). For example, in the lymph nodes, the genomic deletion clearly rescued overexpression of *Ifnar1* and *Ifngr2* without affecting expression of the flanking gene *Tmem50b* or *Dyrk1a*, encoded elsewhere in the triplicated region (new **Fig. 2d**). Notably, these analyses identified hundreds of differentially expressed genes (DEGs) in Dp16 tissues across the genome (new **Fig. 2c**, new **Extended Data Fig. 2e**, new **Supplementary Tables 5-10**). Comparison of the fold changes for these DEGs in Dp16/WT versus Dp16^{2xIfnrs}/WT revealed an overall statistical significant attenuation of gene expression changes in Dp16^{2xIfnrs} in every tissue examined, albeit to varying degrees, as illustrated by lesser effects in the lymph nodes and stronger effects in the brain (new **Fig. 2d**). As described throughout each chapter of the revised manuscript, this dampening of gene expression changes in Dp16^{2xIfnrs} differentially affects specific signalling pathways in each tissue. Altogether, the new results demonstrate that triplication of the *Ifnr* locus not only drives *Ifnr* overexpression, but also contributes to dysregulated gene expression programs throughout the genome.

Of importance to the congenital heart defect chapter and relevant to the Reviewer's comment, we focused our description of the results on the pathways that are consistently dysregulated at every time point (e.g., interferon signalling, EMT, cell proliferation), which clearly adds confidence in the validity of these discoveries.

4. The authors mention a few differences in mice between males and females. For example, only Dp16 males had impairment of relearning in the MWM. Given that Down syndrome affects males and females at an equal frequency, can the authors provide a reason to why there are sex differences in mice?

Response: We thank the reviewer for the interest in the sexual dimorphism observed for some cognitive phenotypes. Notably, although little is known about cognitive differences between males and females with Down syndrome, there is precedent in the literature for sexual dimorphism in cognitive function in mouse models of Down syndrome¹²⁻¹⁵.

A few studies have explored how gender impacts the severity and prevalence of various phenotypes of interest to Down syndrome research, such as visuo-spatial learning and memory^{16, 17}, which is the cognitive domain assessed in our manuscript via the Morris Water Maze (MWM) in mice. For example, among people with Down syndrome, females were found to be less impaired than males in various cognitive assays (**where n=18 females and n=10 males with trisomy 21**)^{18, 19}. However, although the sex-specific differences in Dp16 execution of the MWM observed in our study are in line with previous MWM studies of Dp16¹²⁻¹⁵, clinical relevance of these data is still an active area of investigation in need of additional well-powered studies in people with Down syndrome¹⁷. Therefore, although T21 equally afflicts males and females, there is increasing appreciation that severity of cognitive deficits may vary by gender and specific cognitive domain, but the underlying mechanisms are unknown.

Prompted by the Reviewer's comment, we mention this incipient body of literature in the revised Discussion.

5. Craniofacial abnormalities and the validation of craniofacial landmarks by two individuals who were blinded to the different genotypes is a great approach. Phenotype rescue when removing the 192 kb genomic segment is clear, but what is on this segment that is causing it is unclear.

Response: Thanks for this comment. In the revised manuscript we include a new transcriptome analysis of neural crest-derived facial mesenchyme, an embryonic tissue giving rise to the upper and lower jaw (i.e., mandible). Importantly, our results show that mandibular development is very sensitive to *Ifnr* locus gene dosage. As observed in other tissues, triplicated genes were clearly overexpressed in Dp16 and Dp16^{2xIfnrs} facial mesenchyme but expression of the *Ifnrs* is corrected in Dp16^{2xIfnrs} (new **Extended Data Fig. 6d**). Notably, pathway analysis revealed key gene sets dysregulated in Dp16 but attenuated in Dp16^{2xIfnrs} (new **Fig. 6i-j**, new **Extended Data Fig. 6e**, new **Supplementary Table 10**). For example, the elevation of multiple genes involved in oxidative phosphorylation observed in Dp16 was attenuated in Dp16^{2xIfnrs} (e.g., the cytochrome oxidase subunits *Cox5a1* and *Ndufa4*, new **Fig. 6i-j**). Likewise, Dp16, but less so Dp16^{2xIfnrs}, showed downregulation of gene signatures associated with cell proliferation (e.g., G2M, mitotic spindle, E2F), as illustrated by the proliferation marker *Mki67* and the centromeric protein *Cenpe* (new **Fig. 6i-j**, new **Extended Data Fig. 6e**). Altogether, these results indicate that cell proliferation in this embryonic tissue is negatively impacted by an extra copy of the *Ifnr* locus.

These new results are described in the revised manuscript, while being mindful in our language of the Reviewer's concern about the nature of the genomic deletion created.

6. The differences in cognitive deficits appear to be extremely mild. Associative learning appears more affected, while there is very little if no rescue in spatial learning. Any speculation as to why? Again, greater differences are noted in males than females in reversal phase (Figure 4C). Any speculation as to why?

Response: We agree with the observation that associative learning (measured by the context fear conditioning, CFC test) is more clearly impacted than spatial learning (measured by the Morris Water Maze, MWM test). One possibility is that each type of learning involves different neuronal networks, even perhaps different neurotransmitter systems, which in turn could be differentially impacted by the various genotypes and sexes. For example, the CFC test, which measures a freezing response to an electric shock, may be less reliant on physical endurance, swimming performance, or three-dimensional visual cue recognition required to find the hidden platform in the MWM test. Notably, our results showing differential phenotypes of cognitive deficit severity measured by MWM and CFC for Dp16 as well as sex-specific differences are in line with previous work using this model¹²⁻¹⁵.

Of relevance to this comment, the Reviewer may appreciate that the revised manuscript describes transcriptome analysis of adult brain tissue. This transcriptome analysis confirmed overexpression of triplicated genes in Dp16 and Dp16^{2xifnrs}, along with dysregulation of hundreds of DEGs that was clearly attenuated in Dp16^{2xifnrs} (new Fig. 2e-f). Pathway analysis identified multiple gene signatures important for brain function that are dysregulated in Dp16 but attenuated in Dp16^{2xifnrs}. Salient examples include genes involved in synaptogenesis, such as chimerin 1 (*Chn1*), synapsin 3 (*Syn3*), adenylate cyclase 4 (*Adcy4*), the contactin associated protein 1 (*Cntnap1*), and multiple cadherins (e.g., *Cdh5*, *Cdh23*); SNARE signalling, which is critical for neuronal function, such as synuclein alpha (*Snca*), the synaptosome associated protein 23 (*Snap23*), and synaptotagmin 5 (*Syt5*); and dopamine signalling, such as tyrosine hydroxylase (*Th*), a rate limiting enzyme in dopamine synthesis, the dopamine receptor *Drd2*, and the vesicular acetylcholine transporter *Slc18a3*.

Therefore, although we cannot fully explain why some cognitive domains are more impacted than others nor the mechanisms for sexual dimorphism in these phenotypes, we hope that the results and data shared in the revised manuscript will enable further research in this area.

Reviewer #2.

Remarks to the Author

The study by Waugh et al. investigates the mechanism of interferon pathway hyperactivity in Down's syndrome (TS21). The same lab has shown earlier that TS21 cells are hypersensitive to Interferon stimulation. The authors now expand on this firstly by performing a comprehensive transcriptome and plasma marker analysis on whole blood of TS21 patients with matched controls. Integrated data analysis

reveals a strong association of elevated expression of Interferon Receptors (notably located on chromosome 21) with the DS signature and related activated pathways. These effects appear more directly connected to Interferon genes compared to other triplicated genes in DS. The authors then present a new mouse model of TS21, where they have deleted the Interferon-gene cluster and thereby rescue the elevated dosage caused by TS21. This mouse model is carefully characterized with regards to several phenotypic manifestations of DS patients. Restoration of IFNR gene dosage rescues from a malfunctioning immune system. It rescues embryonic heart defects. It ameliorates some of the cognitive features and craniofacial defects observed in TS21 and the DP16 model. Hence, several of the key physiological features of DS seem to be directly related to the INF-receptor dosage.

I really enjoyed reading this manuscript. The writing is excellent and I felt the interpretation of the data is balanced. The experiments, analyses and mouse model data are carefully performed/analyzed/characterized (e.g. a whole genome sequencing of the mouse model is provided and usually several approaches are presented before reaching a conclusion). The supplementary information and raw data (GEO) are complete and informative. The human dataset on its own will be a valuable resource for the field. Overall, I think the results are of high relevance, since they link a “pro-inflammatory” phenotype caused by increased gene dosage of the IFNR to multiple phenotypic (systemic) alterations (which perhaps one could simplistically assume are caused by the increased gene dosage of several individual genes on chr21 mattering for a given phenotype/organ but not another).

Major Point

I only have one major point, which I think the authors (given their strong transcriptome analysis in Figure 1 and Figure 3) can easily address: I missed some transcriptome characterizations of the mouse model beyond the heart (e.g. the brain relating to the cognitive experiments, perhaps the neural crest for the facial features or some immune cell type related to experiments presented in Figure 2). I think this would nicely make a point in that several tissues, organs and phenotypes – despite their very different developmental trajectories/origins – may be affected by the increased gene dosage in DP16 and accordingly the rescue in DP16-2xIFNR. This may also help to understand why certain features are completely restored, while others (e.g. cognitive) are only ameliorated, but do not fully rebalanced to WT.

Such datasets would also clarify the expression of DP16-2xIFNR in the different tissues compared to WT (two alleles on the two parental alleles). It is possible that because the two doses of the receptor are expressed in a slightly different way than in WT (excellent scheme in Extended Data Fig. 2e), the levels might not be exactly balanced (although the qPCR in Fig. 2b and the FACS in 2c would suggest this is the case). I would suggest that the authors provide a compiled qPCR and FACS, if possible, for all representative organs / cell-types studied (white blood cells, heart, brain, craniofacial (e.g. neural crest?)).

Response: We are grateful for the overall positive assessment from Reviewer #2 and for the constructive comments, which, along with comments from the other Reviewers, prompted us to

complete a comprehensive transcriptome analysis of different tissues (and in the case of the heart also at different time points). In the revised manuscript we describe the results of six new transcriptome analysis from tissues relevant to the phenotypes described: adult mesenteric lymph nodes, embryonic heart (two different time points, E12.5 and E18.5) and adult heart, embryonic neural crest-derived facial mesenchyme at E10.5, and adult brain tissue. Throughout the manuscript, as relevant to each chapter and figure, we describe the presence of gene expression programs dysregulated in the Dp16 model that are attenuated in Dp16^{2xlfnr} mice. As we describe each of these transcriptome studies, we followed the Reviewer's guidance and highlight how the 4 *lfnr*s are overexpressed in Dp16 but not so in the Dp16^{2xlfnr} strain. Additionally, we describe how the Dp16^{2xlfnr} strain still overexpresses the other triplicated genes, while displaying significant attenuation of the fold changes for differentially expressed genes (DEGs) encoded outside of the triplicated region. As we describe the transcriptome study that matches each chapter (e.g., immune hypersensitivity, congenital heart defects, cognitive impairments, craniofacial development), we highlight the gene signatures that are most sensitive to dosage of the *lfnr* locus and how these gene signatures may be connected to the observed phenotype.

Minor Points

The authors chose a q-value of 0.1 for their differential expression analysis. This seems fairly high in my view, as typically 0.05 or even 0.01 is chosen. Could the authors perhaps comment in methods of why this threshold was chosen and how many DE genes would be recovered at FDR of 0.05?

Response: Thanks for this comment. We have employed the DESeq2 informatics tool²⁰ to analyze all RNAseq data. **The DESeq2 method has been extensively benchmarked against other bioinformatics tools, having been cited more than 43,000 times since publication in 2014²⁰. During design and testing of DESeq2, a q-value of 0.1 was found to have high precision and sensitivity and is the standard cut-off recommended by the developers for differential gene expression analysis²⁰.**

To answer the Reviewer's question about how the q<0.1 cut-off compares to q<0.05, the answer varies from one transcriptome experiment to another. For example, in the comparison of Dp16 versus wild type lymph nodes, 569 DEGs would pass the cut-off at q<0.1 versus 401 DEGs at q<0.05. To give the Reviewer a sense of how stringent the multiple hypotheses correction is when using DESeq2, in this same experiment a q=0.1 is equivalent to an unadjusted p=0.0035.

Importantly, to address the Reviewer's concern, we provide all DESeq2 output tables in our Supplementary Tables, one for each transcriptome study, including all q values and unadjusted p values, so that readers can analyze these datasets with their cut-off value of choice.

The methods paragraph for the Gene Set Enrichment Analysis and Spearman Correlation is relatively short and refers to packages and previously published information. Could the authors expand this in order to better explain the analysis for a non-expert?

Response: Thanks for this constructive comment, which prompted us to better explain these two methods. Gene Set Enrichment Analysis (GSEA) is a bioinformatic tool that analyses the distribution of

genes in a given pathway or signature (or in any gene set really) within a larger **ranked** list, to define whether the genes in the pathway are ‘enriched’ toward one end or the other of the ranked list. For example, the IFN Gamma Response gene set has a very high positive enrichment score (i.e., normalized enrichment score or **NES** in **Fig. 1**) in the ranked list generated by DESeq2 analysis of the transcriptome changes observed in persons with trisomy 21 versus euploid controls. These results indicate that genes in the IFN gamma pathway are highly enriched among those elevated in trisomy 21. Just as well, GSEA can evaluate the distribution of the IFN gamma gene set in a ranked list of Spearman correlation values (or any other ranked list really). For example, the IFN Gamma Response gene set has very high NES values in the ranked lists of Spearman rho values calculated from correlations between any of the four *IFNRs* encoded on chromosome 21 versus all other mRNAs in the transcriptome (**Fig. 1c**). That is, genes in the IFN gamma pathway are highly enriched among mRNAs showing positive co-expression values (high Spearman rho) with *IFNRs*, something that is not true for most other genes encoded on chromosome 21.

Guided by the Reviewer’s comment, we have expanded this section of the Methods.

The authors should double check the references for Figures in their text. E.g. on page 5 (“... rho values by GSEA (see Methods, Extended Data Fig. 1b)”) I think this refers to Extended Data Fig. 1c and following.

Response: Thanks for this noticing this oversight, we have corrected.

Could the authors provide a supplementary list which other 120 genes are triplicated in the Dp16 mouse model?

Response: Thanks for this comment, we agree that this list will be useful for readers. We supply the list of triplicated genes detected in the Dp16 transcriptome studies in the updated **Supplementary Table 4 – Additional Methods Data**, tab E.

Do the authors have an explanation why the RFI of STAT1 in the stimulated condition for DP16xIFNR is lower than in WT (Extended Data Fig. 3f)?

Response: We have been intrigued by this observation ourselves. Our transcriptome analysis across multiple tissues shows that *Ifnr* expression is reduced to ‘normal levels’ in Dp16^{2xIfnrs}, not any lower. In the revised manuscript we doubled the sample size for this experiment and the results remain the same (new **Fig. 3b**). One favored hypothesis is that whereas both WT and Dp16^{2xIfnrs} mice have two copies of the *Ifnr* locus, other genes still triplicated in Dp16^{2xIfnrs} may exert some form of dampening of STAT1 phosphorylation in the stimulated condition. In other words, we envision some form of negative feedback loop on STAT1 phosphorylation that is only evident when the hypersensitivity driven by *Ifnr* triplication is undone by our genetic correction. In the revised manuscript we describe this tantalizing result and briefly mention this hypothesis.

Page 9 “... versus Dp16 transcriptomes revealed significant normalization of the gene signatures dysregulated in Dp16”. I am not sure what the authors mean with “normalization”, most likely because this a technical term referred to RNA-seq normalization. Perhaps the authors can rephrase this slightly?

Response: We agree with the Reviewing that ‘normalization’ could be misinterpreted in this context and have rephrased accordingly. What we meant in this passage, and what is now also evident in the many additional transcriptome experiments, is that many gene expression changes observed in Dp16 mice are ‘attenuated’, ‘rescued’, or ‘prevented’ in Dp16^{2xIfnrs}. We have now rephrased all references to “normalization” with the alternative wording.

Fig 3f: Can the authors, instead of the pathways, provide a simple heatmap of the 73 DEGs on the triplicated segment and how they change in the three conditions / genotypes analyzed? Is expression of other genes on DP16 (e.g. DYRK1A) restored by rebalancing IFNR to 2x? I would also like to see a heatmap of all 120 triplicated genes and their expression in the three conditions. Can they also provide a heatmap of the remaining DEGs (not on the triplicated segment)? Furthermore, can the volcano plot (3e) be provided for all comparisons (DP16/WT, DP162x/WT, DP162x/DP16) and the overlaps between this dataset (e.g. upset plot) be shown? I understand that some of these informations can be retrieved from the Supplementary Lists, but I feel these comparisons are key to the findings and should be provided as a Main Figure.

Response: We are grateful for this comment, which inspired us to try many different ways of representing the results from the six new transcriptome analyses for greater clarity. Guided by this comment, we have introduced several graphs that we hope the Reviewer will find informative and easy to interpret:

1. All three Volcano plots for each genotype comparison, side-by-side and on the same scales, color coding the position of the 4 *Ifnrs* (red), all triplicated genes (blue), all DEGs outside of the triplicated region (gray), and all other genes (black), as well as Manhattan plots of Mmu16 using the same color coding. We provide these two formats for the lymph node transcriptome experiment in new **Fig. 2c** (Volcano plots) and **Extended Data Fig. 2f** (Manhattan plots), which is the first transcriptome experiment described. These graphs look very similar for all other transcriptomes, so we do not think it is necessary to include the other 5 sets of Volcano plots and Manhattan plots in addition to the Supplemental Tables. However, the Reviewer will likely appreciate what we believe is a good alternative that addresses the Reviewer’s curiosity:
2. Scatter plots comparing the **fold changes** for the 4 *Ifnrs* (red), all triplicated genes (blue), and all DEGs outside of the triplicated region (gray) in the Dp16/WT versus Dp16^{2xIfnrs}/WT comparisons, including linear fits and slope values for the triplicated genes (in blue) versus all other DEGs (gray). We believe these graphs, which we show for every single transcriptome study (new **Fig. 2e**, **Fig. 4e**, **Extended Data Fig. 6d**) are an elegant way to display the differences in expression of all DEGs, triplicated or not, across the three genotypes, akin to the heatmaps envisioned by the Reviewer. Consistently, these scatter plots highlight three key results: i) The genomic deletion consistently corrects *Ifnr* expression across multiple tissues, ii) The genomic deletion largely preserves overexpression of other triplicated genes, and iii) Correction of *Ifnr* locus copy number attenuates many, but not all, changes affecting DEGs encoded elsewhere in the genome. The comparison of fold changes in triplicated genes versus other DEGs is also presented as sina plots in new **Fig. 2f** and **Extended Data Fig. S2e** for calculation of statistical differences.

- Heatmaps as envisioned by the Reviewer showing example genes accompanying every transcriptome study described in the manuscript, with an emphasis on gene signatures that are specially sensitive to *Ifnr* copy number, showing the relative expression values in WT, Dp16 and Dp16^{2xIfnrs}. Given space limitations within the figures and to ensure clarity, we have restricted the number of genes to show in these heatmaps to 10 or less.

Since this is a standard RNA-seq performed from mice, I would recommend choosing a FDR of 0.05.

Response: Please see above our response to a similar comment, where we explain our choice to follow the recommendation from DESeq2 developers²⁰.

Fig. 4a – Hard to distinguish the size of the square symbol that indicates the $-\log_{10}q$

Response: Thanks for the comment, we have modified this figure to give a greater dynamic range to the squares representing the varying statistical significance.

Fig. 4e – is the comparison between DP16 and DP16-2xIFNR statistically significant?

Response: In this figure, the comparison between Dp16 and Dp16^{2xIfnrs} is not statistically significant. In this experiment we tested if Dp16 or Dp16^{2xIfnrs} differ from the reference WT control group by one-way ANOVA with Dunnett’s test correction, which is the appropriate test for such comparison, revealing a significant difference in Dp16 but not in Dp16^{2xIfnrs}. Testing if Dp16 is different than Dp16^{2xIfnrs} would require testing all possible combinations of group comparisons and add an additional hypothesis to the correction, for which Tukey’s test would be more appropriate. Using Tukey’s test would bring the p value for the WT to Dp16 comparison barely above 0.05 (see Fig. R1 shown here comparing Dunnett’s versus Tukey’s corrections). We chose to display the results as currently presented (one-way ANOVA with Dunnett’s test) because these subtle yet significant differences in MWM probe trials are in line with previous publications of Dp16¹²⁻¹⁵. Otherwise, we would be concluding that Dp16 do not have differences in MWM performance, countering multiple published reports.

The paragraph explaining the results of the craniofacial anomalies is a bit hard to understand for a non-specialist. Perhaps this paragraph can be rewritten for a more general audience, while the details can be provided in a supplementary note?

Response: Thank you for this constructive critique. We have rewritten this section with more details and more general language where appropriate. We have taken extra effort to explain in plain language some of the terms used (e.g., synchondrosis, calvarium).

Fig. R1. Reversal probe results with correction for two comparisons.

Earlier work has provided crucial roles for DYRK1A dosage in Trisomy21, especially with regards to the neurological and cognitive features of the disease. Perhaps in the nervous system, IFNR together with DYRK1A dosage are the key drivers for the DS phenotype?

Response: We appreciate this comment, which we address in the revised text. As explained in the response to the next comment, *Dyrk1a* overexpression is preserved in various tissues of the Dp16^{2xifnrs} mice, including brain tissue, suggesting that this key gene could contribute to phenotypes in the absence of *Ifnr* triplication. As described in the manuscript, correction of *Ifnr* dosage does not fully rescue the cognitive impairments observed in Dp16, suggesting that other triplicated genes, including *Dyrk1a*, could be having important effects as well, which we acknowledge in the revised Discussion.

I noticed that in the heart RNA-seq, *Dyrk1a* expression was restored by deleting the third copy of Interferon receptors. Maybe this is not the case in the brain? I would encourage the authors to expand the results and discussion in that regard.

Response: Thanks for this comment, which encouraged us to use *Dyrk1a* as an example of a gene triplicated in Dp16 that is not affected by the genomic deletion across the various tissues studied. In the new transcriptome analyses of six different tissues, for which we employed a larger sample size of at least 5-6 animals per genotype, *Dyrk1a* and most other triplicated genes were not significantly affected by the genomic deletion. The new transcriptomes of heart tissue at E12.5, E18.5, and adult age replace the single, less powered experiment in the original submission at E15.5. Inspired by the Reviewer's comment, we highlighted the position of *Dyrk1a* relative to *Ifnar1* in various plots where the transcriptome data is presented. We also elaborate on the possibility that *Dyrk1a* may contribute to the cognitive phenotypes, which are not fully rescued in Dp16^{2xifnrs}, in the revised Discussion.

p.15 / L7 – "... provides much needed mechanistic insight about the ..." – I would remove the word "mechanistic" here, because at the mechanistic / molecular level the link between IFNR and the craniofacial features still remains unclear.

Response: We have removed the term "mechanistic" from this sentence.

Extended Data Fig 1a: The authors measured two inflammatory markers from plasma (CRP and IL-6). The scheme (and also one passage in the text stating "proteome") gives the impression many markers were tested. I would suggest adjusting the scheme and just write CRP/IL-6.

Response: We have modified the figure and text as suggested by the Reviewer.

Extended Data Fig 1c: Too small font / labels.

Response: We have increased the size of font and labels in Extended Data Fig 1c.

Reviewers #3 & #4: Experts on DS genetics, reviewed together.

Remarks to the Author

I read this manuscript with much interest because it addresses an important question in Down syndrome research, with biological and translational implications. Several prior studies, including from this group, have reported that individuals with Down syndrome have elevated inflammation and immune dysregulation, co-occurring with dysregulation of interferon-related genes. Since many things can influence these complex bodily systems, one could not assume that these effects are due to the presence of four interferon receptor genes on Chr21. Nonetheless, this is a compelling hypothesis, which Waugh et al. have now tested in a comprehensive study in a mouse model of Down syndrome. The heart of the study is the deletion of a cluster of four IFN receptor genes from a previously studied mouse model, Dp16, which carries a segmental duplication of ~120 genes orthologous to human Chr21.

While the study begins with pathway and cytokine analysis in human blood samples, the novel and impressive main part of the study is the engineering and analysis of trisomic mice corrected for triplication of the IFN receptor genes. The authors deleted a ~192kb segment containing the four interferon receptor genes in a WT mouse, then interbred these with the trisomic DP16 mouse, thereby correcting dosage for just those genes. They then assess and report that correcting IFN receptor gene dosage essentially prevents several major phenotypes of the DS mouse, some of which are remarkable and surprising.

The implications of these findings for humans remains to be directly established, but is plausible, and the findings in this DS mouse model alone are of high impact and significance. Overall, this study appears to be comprehensive, well-done, and well-presented, although I have some significant comments or questions that should be addressed.

Major comments

1. The first section analyzing the human blood transcriptome and cytokines is a reasonable addition; it is not novel nor strongly compelling, but provides context likely intended to strengthen the connection to human DS. The data serves to reinforce that inflammatory signatures are prevalent in human DS, although I didn't find the argument so compelling that the inflammatory pathways impacted correlate with the IFNR expression rather than other Chr21 genes. (ITSN1 seemed perhaps the strongest correlation.) It is also a bit counter-intuitive how one could discriminate this point clearly, given that expression of most Chr21 genes would, a priori, be elevated in concert due to trisomy. The authors emphasize the stronger correlation of IFNR mRNAs with inflammatory pathways in contrast to DYRK1A, a Chr21 gene which many studies report are connected to numerous phenotypes of DS. (It wasn't clear to me how the use of whole blood RNAseq here would complicate such comparisons.) It seems notable that seven of the top ten GSEA gene sets involve IFN signalling and inflammation, although it would be good for authors to indicate how many gene sets in blood could be considered related to IFNs and inflammation. Also, does triplication of six IFN-response genes (including MX1 and MX2) itself make it more likely that these pathways are near the top of the list? Or were the chr21 genes removed from the pathway analysis?

Response: We appreciate the overall positive assessment and the opportunity to elaborate on the results arising from the whole blood transcriptome analysis.

First, we agree with the Reviewers that the four *IFNRs* are not the only genes that show strong positive associations between their mRNA expression levels and expression of genes involved in inflammatory pathways and plasma levels of markers of inflammation (i.e., CRP, IL6). The Reviewers correctly notice that *ITSN1* (as well as other HSA21 genes) also correlate positively with inflammation signatures. However, the surprising and novel observation ('counter-intuitive' as the Reviewers stated) is that only a minor fraction of chr21 genes show such significant correlations with the inflammatory signatures elevated in Down syndrome, including the *IFNRs* and *ITSN1*. Therefore, a key finding in this part of the manuscript is that not all HSA21 genes are overexpressed in concert to the same degree in every person with Down syndrome, and that this 'variegated overexpression' correlates with different gene expression signatures dysregulated in trisomy 21. More specifically, individuals with Down syndrome who overexpress the *IFNRs* show signatures of inflammation but do not show signatures of increased oxidative phosphorylation, which are also elevated in trisomy 21 (**Fig. 1c-d**). In contrast, individuals with Down syndrome who overexpress *DYRK1A* show transcriptomic signatures of increased protein secretion and decreased heme metabolism, while those who overexpress *ATP5PO*, a subunit of the mitochondrial ATPase complex encoded on HSA21, show signatures of increase oxidative phosphorylation (**Fig. 1c**).

To make this observation more explicit to readers, in the revised manuscript we revised the text to enhance the explanation of these results, but we also created a new figure panel showing examples of HSA21 genes that are clearly co-expressed (e.g., *IFNAR1* vs *IFNGR2*, *DYRK1A* vs *ZBTB21*) versus those are not co-expressed and are actually anti-correlated (e.g., *IFNAR1* vs *ATP5PO*, *DYRK1A* vs *CSTB*). These results are shared in new **Extended Data Fig. 1e**. In the future, we hope to expand on these observations to elucidate the impacts of variegated overexpression of HSA21 genes in Down syndrome.

Regarding the comment about how many inflammatory signatures are related to IFN signalling, this is a difficult question to answer with precision, as our collective understanding of IFN-regulated pathways continues to evolve. Whereas many inflammatory pathways are considered to act downstream of IFN signalling (e.g., IL6 signalling, TNF α signalling), these same pathways can also be activated by other immune stimuli. In other words, it is formally possible that the inflammatory profile of Down syndrome contains both IFN-dependent and IFN-independent components, which was a key motivation for our experiments in the mouse model, leading to the important conclusion that the immune hypersensitivity to viral mimetics observed in Dp16 mice requires a third copy of the *Ifnr* locus. Nevertheless, we remain open minded about the possibility that individuals with Down syndrome also experience alternative, IFN-independent inflammatory processes.

Lastly, regarding the question as to whether the inflammatory signatures in the transcriptome are driven by genes encoded on HSA21, the answer is no, the bulk of these signatures is composed of genes encoded elsewhere in the genome. We published a comparative pathway analysis including or excluding

HSA21 genes back in 2016 using transcriptome data generated from 6 pairs of skin fibroblasts with or without trisomy 21²¹, showing that IFN signatures are still the top signature of the transcriptome changes even when removing HSA21 genes. This remains true for the much larger whole blood transcriptome analysis shared in this manuscript (400 samples), and we share here with Reviewers the full GSEA analysis with and without the HSA21 genes (Fig. R2).

2. What is most impressive is the creation of and results from the mouse model with normalized copy number for Interferon receptor gene cluster, which they report prevents several major DS-related hallmarks in Dp16 mice. In this context, it is important that the comparison of WT, Dp16, Dp16xIFNRs mice, for all these assays, was done between mice/embryos for which there is essentially no difference in genetic backgrounds. In reading the methods, I surmise that this is the case, and that the comparisons were done between littermates with different genotypes, not pups for different litters....and not with differences in genetic backgrounds. This is important to the whole study, and assuming my reading is correct, the comparison of mice from the same litter and background should be briefly highlighted in the text. (Should there be any caveats to this, or complicating factors, this should be acknowledged and important to address in revision.)

Response: The Reviewer is correct in the sense that all experiments were done between mice/embryos for which there is essentially no difference in genetic backgrounds, which was achieved by employing animals from the same litters to power each comparison arm. Nevertheless, naturally, the number of animals of each genotype varied somewhat from one litter to another. However, when multiple litters are employed and the littermates randomly combined to achieve enough numbers in each genotype arm, the impacts of any potential genetic drift are minimized. To make this clear to readers, we added the following sentence in the 'Animal Husbandry' section of the Methods:

"In order to power the comparison arms representing each genotype in this study (i.e., WT, Dp16, Dp16^{2xIfnrs}, WT^{1xIfnrs}), multiple litters were combined, where each litter contributed randomly to the sum of each cohort, thus minimizing the impact of any potential shift in genetic background."

With that being said, in the experiments investigating differences in timelines of reaching developmental milestones, which could be very sensitive to litter size and maternal care, we accounted

Fig. R2. Gene set enrichment analysis (GSEA) of transcriptome changes observed in the whole blood of individuals with Down syndrome versus age- and sex-matched euploid controls using all genes detected (left) or removing the genes encoded on

for potential impacts of litter in our statistical analysis, as described in the Methods section under ‘Developmental Milestones’. The exact description is pasted here:

*“To test for differences in the chance of success in achieving each developmental milestone, results were treated as time-to-event data and analysed using a mixed effects Cox regression approach using the survival (version 3.2-7²²), coxme (version 2.2-16²³), emmeans (version 1.5.1²⁴) and broom (version 0.7.9²⁵) packages in R. Models for each milestone were generated using the coxme() function from the coxme package with a time-to-event survival object as the outcome variable, genotype as the variable of interest, and with adjustment for sex and sex*genotype interaction as fixed effects and litter as a random effect.”*

3. In order to make the mice, the authors first deleted one copy of the IFNR gene cluster from WT mice..... thereby creating mice monosomic for the IFNR genes....but there is no mention of anything about these monosomic mice. Their diagram indicates monosomic mice would represent 25% of F1 pups..... so were these mice noticeably impacted in any way? This is interesting in general, but also has some relevance to the dosage sensitivity of IFNR genes....so there should be at least some brief summary about this. Do the authors think that monosomy for IFNR genes is more benign than trisomy for these genes?

Response: We agree with the Reviewers that the monosomic WT^{1xIfnrs} mice are worthy of further study, although probably for other scientific inquiries related to IFN signalling. For the purpose of our work, which was focused on understanding the impacts of correcting *Ifnr* copy number in the Dp16 model, we did not characterize them in great detail, being excluded from most experiments, and only use them as controls in a few experiments to monitor *Ifnr* expression (e.g., **Fig. 3a**). We look forward to being able to share these animals with the community to empower additional studies.

In response to the Reviewers question, we hypothesize that these animals will be immunocompromised (both WT^{0xIfnrs} and WT^{1xIfnrs}), as they should be unable to mount appropriate IFN responses across all three types of IFN signalling. This may be manifested as greater susceptibility to pathogens, particularly viruses. Whether or not they would present developmental phenotypes or other phenotypes not clearly related to immune function would warrant further investigation.

4. The evidence that the IFNR correction prevents the heart defects is particularly striking..... even in embryos in which I would assume there are not immune insults? I was pleased to see that the analysis of embryos was done blind to their genotypes. However, a significant question comes up in Figure 3F. In transcriptomic analysis of E15.5 heart tissues, why are the differences in DP16/WT comparisons almost the same as in Dp162xIFNRs/WT comparisons? The IFNR-corrected Dp16 mice transcriptomes should be more similar to WT if so much is driven by the IFNR genes, but these results suggest that very similar pathways are dysregulated by other mechanisms independent of trisomy for the IFNR cluster. I understand that the Dp16 versus Dp162xIFNRs comparison shows some difference, but it is surprising and important that this data shows that mice with normalized IFNR gene dosage still have dysregulation of the key pathways that include: IFN alpha response, IFN gamma response, inflammatory response, IL6

Jak Stat3 signalling, IL2 Jak Stat2 signalling. These pathways seem central to this paper, but this point is not discussed. Is it a discrepancy that warrants substantial softening of the impression that inflammation is driven by or mostly by the IFNR gene cluster, or how do the authors explain it?

Response: We appreciate this comment and welcome the opportunity to further elaborate on these results both here in the response to Reviewers and in the revised manuscript. The Reviewers may appreciate that in the revised manuscript we describe the results of six new transcriptome analysis from tissues relevant to the phenotypes described: adult mesenteric lymph nodes, embryonic heart (two different time points, E12.5 and E18.5) and adult heart, embryonic neural crest-derived facial mesenchyme at E10.5, and adult brain tissue. Throughout the manuscript, as relevant to each chapter and figure, we emphasize several key points that address the Reviewers comments:

First, correction of *Ifnr* locus copy number restores normal *Ifnr* expression while largely preserving overexpression of the other triplicated genes.

Second, as noted by the Reviewers, correction of *Ifnr* copy attenuates only some specific pathways in each tissue, and these vary from tissue to tissue, while preserving many other transcriptome changes. This is a very important result throughout the manuscript: a mere 50% reduction in the expression of only 4 of the ~120 triplicated genes suffices to rescue (totally or partially) many hallmarks of Down syndrome while attenuating only a small fraction of the transcriptome changes caused by the trisomy. This indicates that although the other ~120 triplicated genes continue to exert their effects on diverse signalling pathways, the relatively small contribution of the *Ifnr* locus to gene expression changes is key for full penetrance and/or severity of major phenotypes.

Third, the contribution of *Ifnr* triplication to the global transcriptome changes varies both from one tissue to another and also in the same organ at different developmental time points. For example, in new **Fig. 2c**, we highlight how the global attenuation of gene expression changes is minimal in lymph nodes, but very striking in brain tissue. Likewise, *Ifnr* triplication contributes to many global transcriptome changes in the developing heart (E12.5), but less so in the adult heart (see new **Fig. 4e**). Furthermore, the contribution of *Ifnr* locus triplication to the same pathway may vary drastically across tissues. For example, *Ifnr* triplication contributes to signs of **increased** cell proliferation in lymph nodes (new **Fig. 3c**), potentially indicative of elevated immune activity, but also to signs of **decreased** cell proliferation in the developing heart and facial mesenchyme (new **Fig. 4f-g** and **Fig. 6i-j**).

The Reviewers are correct in the sense that some inflammatory signatures are still elevated in the Dp16^{2xIfnrs} mice. For example, in the lymph nodes, IL2 STAT5 signalling is elevated in both Dp16 and Dp16^{2xIfnrs} mice relative to WT animals. However, the mild dampening in the dysregulation of these genes observed in Dp16^{2xIfnrs} suffices to produce a significant decrease in the score for this signature when comparing Dp16^{2xIfnrs} to Dp16 (**Fig. 3c**). Similar observations are made in the developing heart. Altogether, these results relate to our response above commenting about IFN-dependent versus IFN-independent inflammation in Down syndrome. Clearly, not all inflammatory signalling in Down syndrome depends on a third copy of the *Ifnr* locus, and we make this point explicit in the revised text.

Hypothetically, other triplicated genes may also contribute to immune dysregulation in Down syndrome, and there is also the possibility that the triplication of a large genomic fragment could trigger a form of inflammation independent of gene content. For example, cellular stress from extra DNA, even from a Robertsonian translocation²⁶, and resulting activation of Damage Associated Molecular Patterns (DAMPs)^{27, 28} could still be activating IFN signalling in the absence of increased *Ifnr* gene dosage by increasing levels of IFN ligands and/or alternative mechanisms. We have revised the Discussion to acknowledge these potential alternative mechanisms.

The Reviewers also highlight a very important observation: the developing heart shows signs of immune dysregulation even in the absence of any known immune stimuli. In these experiments, the mice were housed in specific pathogen free conditions. Of note, various IFN ligands are made throughout fetal development (e.g., Type III IFNs constitutively produced by *syncytiotrophoblasts*)^{8, 29}. We hope that future studies building on these data will focus on the source of IFN-hyperactivity *in utero* (e.g., tonic signalling or various IFN ligands) to aid in development of preventative and/or therapeutic treatments of phenotypes modulated by increase dosage of the *Ifnr* locus in Down syndrome.

5. The evidence for correction or mitigation of “cognitive” deficits and craniofacial changes are both surprising, with important implications. Analysis of memory and learning deficits in mice can be tricky or potentially subjective, so I was pleased to see this analysis was also done blind, perhaps worth a mention in the text. While not required here, it is generally the case that it is good to have independent analysis and corroboration from an independent lab.... which I assume will come in the future. If any of these findings are difficult to interpret, it would be fine to acknowledge any limitations in Discussion.

Response: Thanks for these comments, which prompted us to extend the description of blind analysis of the cognitive testing in adult mice. Experimenters were blinded to genotype, but of course Dp16 animals are slightly smaller and have differences in craniofacial morphology that may be noticed by some trained personnel. We hope to include independent analysis and corroboration from an independent lab in the future. We also hope that our detailed methods will help with reproducibility by other labs. For now, we are encouraged to see that our results align with previous publications reporting the benefits of broad anti-inflammatory agents in Dp16 and other comparable mouse models^{30, 31}, further reinforcing the notion that inflammation contributes to the cognitive profile of Down syndrome, and we made sure to cite these studies.

6. It is understandable that the emphasis is placed on phenotypes that have been ameliorated, but the authors should comment somewhat more about any phenotypes not mitigated by reducing IFNRs to disomy.... this of interest too. For example, Fig 2e does not show significant difference in weight loss between Dp16 and Dp162xIFNRs on exposure to viral mimetic. Similarly in Figure 3h-j, there isn't a significant difference between the Dp162xIFNRs, DP16 and WT mice. Is reducing IFNR dosage sufficient to maintain homeostasis in pathways regulating interferon alpha response, oxidative phosphorylation and Myc targets. Finally, it is noted that motor co-ordination in Dp162xIFNRs and DP16 mice do not

show much rescue. Please comment on other readily apparent phenotypes if present in Dp16, such as effects on longevity or size.

Response: We are grateful for this comment, which inspired us to include another Dp16 phenotype that was not rescued in Dp16^{2xlnfrs}: decreased neonatal size/growth. It is well established that Dp16 neonates are smaller than wild type littermates, and this is not different in Dp16^{2xlnfrs}. We share this result in new **Extended Data Fig. 5c** to accompany the results demonstrating the rescue of developmental delays, which are clearly not matched by increased neonate size/growth. Guided by the Reviewer's comments, we have highlighted in multiple places the instances where there is no rescue or partial rescue of phenotypes (or gene expression changes) in Dp16^{2xlnfrs}.

Minor comments

1. Figure 1c in legend is mislabeled as "b"

Response: Thanks for catching this oversight, which is now corrected.

2. Extended data figure 1e: There seems to be a red triangle on the figure above IFNAR1 gene which hasn't been explained in the legend. What does it imply for?

Response: Thanks for noticing this, the triangle in volcano plots is used when the q value (or p value) is smaller than what the code can calculate, basically 0, which in this case applies to the gene KCNJ15. We have clarified this in the figure legend.

References.

1. Crow Y, Keshavan N, Barbet JP, Bercu G, Bondet V, Boussard C, Dedieu N, Duffy D, Hully M, Giardini A, Gitiaux C, Rice GI, Seabra L, Bader-Meunier B, Rahman S. Cardiac valve involvement in ADAR-related type I interferonopathy. *J Med Genet.* 2020;57(7):475-8. Epub 2019/11/28. doi: 10.1136/jmedgenet-2019-106457. PubMed PMID: 31772029.
2. Feigenbaum A, Muller C, Yale C, Kleinheinz J, Jezewski P, Kehl HG, MacDougall M, Rutsch F, Hennekam RC. Singleton-Merten syndrome: an autosomal dominant disorder with variable expression. *American journal of medical genetics Part A.* 2013;161A(2):360-70. Epub 20130115. doi: 10.1002/ajmg.a.35732. PubMed PMID: 23322711.
3. Jang MA, Kim EK, Now H, Nguyen NT, Kim WJ, Yoo JY, Lee J, Jeong YM, Kim CH, Kim OH, Sohn S, Nam SH, Hong Y, Lee YS, Chang SA, Jang SY, Kim JW, Lee MS, Lim SY, Sung KS, Park KT, Kim BJ, Lee JH, Kim DK, Kee C, Ki CS. Mutations in DDX58, which encodes RIG-I, cause atypical Singleton-Merten syndrome. *Am J Hum Genet.* 2015;96(2):266-74. Epub 20150122. doi: 10.1016/j.ajhg.2014.11.019. PubMed PMID: 25620203; PMCID: PMC4320253.

4. Rutsch F, Buers I, Nitschke Y. Hereditary Disorders of Cardiovascular Calcification. *Arterioscler Thromb Vasc Biol.* 2021;41(1):35-47. Epub 20201112. doi: 10.1161/ATVBAHA.120.315577. PubMed PMID: 33176451.
5. Aicardi J, Goutieres F. A progressive familial encephalopathy in infancy with calcifications of the basal ganglia and chronic cerebrospinal fluid lymphocytosis. *Ann Neurol.* 1984;15(1):49-54. Epub 1984/01/01. doi: 10.1002/ana.410150109. PubMed PMID: 6712192.
6. Ye Z, Wang L, Yang T, Chen L, Wang T, Chen L, Zhao L, Zhang S, Zheng Z, Luo L, Qin J. Maternal Viral Infection and Risk of Fetal Congenital Heart Diseases: A Meta-Analysis of Observational Studies. *J Am Heart Assoc.* 2019;8(9):e011264. Epub 2019/04/19. doi: 10.1161/JAHA.118.011264. PubMed PMID: 30995883; PMCID: PMC6512143.
7. Begum NNF. Novel facial characteristics in congenital rubella syndrome: a study of 115 cases in a cardiac hospital of Bangladesh. *BMJ Paediatr Open.* 2020;4(1):e000860. Epub 2020/12/12. doi: 10.1136/bmjpo-2020-000860. PubMed PMID: 33305019; PMCID: PMC7692988.
8. Yockey LJ, Iwasaki A. Interferons and Proinflammatory Cytokines in Pregnancy and Fetal Development. *Immunity.* 2018;49(3):397-412. Epub 2018/09/21. doi: 10.1016/j.immuni.2018.07.017. PubMed PMID: 30231982; PMCID: PMC6152841.
9. Toubiana J, Okada S, Hiller J, Oleastro M, Lagos Gomez M, Aldave Becerra JC, Ouachee-Chardin M, Fouyssac F, Girisha KM, Etzioni A, Van Montfrans J, Camcioglu Y, Kerns LA, Belohradsky B, Blanche S, Bousfiha A, Rodriguez-Gallego C, Meyts I, Kisand K, Reichenbach J, Renner ED, Rosenzweig S, Grimbacher B, van de Veerdonk FL, Traidl-Hoffmann C, Picard C, Marodi L, Morio T, Kobayashi M, Lilic D, Milner JD, Holland S, Casanova JL, Puel A, International SG-o-FSG. Heterozygous STAT1 gain-of-function mutations underlie an unexpectedly broad clinical phenotype. *Blood.* 2016;127(25):3154-64. Epub 20160425. doi: 10.1182/blood-2015-11-679902. PubMed PMID: 27114460; PMCID: PMC4920021.
10. Li H, Edie S, Klinedinst D, Jeong JS, Blackshaw S, Maslen CL, Reeves RH. Penetrance of Congenital Heart Disease in a Mouse Model of Down Syndrome Depends on a Trisomic Potentiator of a Disomic Modifier. *Genetics.* 2016;203(2):763-70. Epub 2016/04/01. doi: 10.1534/genetics.116.188045. PubMed PMID: 27029737; PMCID: PMC4896192.
11. Lana-Elola E, Watson-Scales S, Slender A, Gibbins D, Martineau A, Douglas C, Mohun T, Fisher EM, Tybulewicz VL. Genetic dissection of Down syndrome-associated congenital heart defects using a new mouse mapping panel. *Elife.* 2016;5. Epub 2016/01/15. doi: 10.7554/eLife.11614. PubMed PMID: 26765563; PMCID: PMC4764572.
12. Li Z, Yu T, Morishima M, Pao A, LaDuca J, Conroy J, Nowak N, Matsui S, Shiraishi I, Yu YE. Duplication of the entire 22.9 Mb human chromosome 21 syntenic region on mouse

- chromosome 16 causes cardiovascular and gastrointestinal abnormalities. *Hum Mol Genet.* 2007;16(11):1359-66. Epub 2007/04/07. doi: 10.1093/hmg/ddm086. PubMed PMID: 17412756.
13. Goodliffe JW, Olmos-Serrano JL, Aziz NM, Pennings JL, Guedj F, Bianchi DW, Haydar TF. Absence of Prenatal Forebrain Defects in the Dp(16)1Yey/+ Mouse Model of Down Syndrome. *The Journal of neuroscience : the official journal of the Society for Neuroscience.* 2016;36(10):2926-44. doi: 10.1523/JNEUROSCI.2513-15.2016. PubMed PMID: 26961948; PMCID: 4783496.
 14. Yu T, Liu C, Belichenko P, Clapcote SJ, Li S, Pao A, Kleschevnikov A, Bechard AR, Asrar S, Chen R, Fan N, Zhou Z, Jia Z, Chen C, Roder JC, Liu B, Baldini A, Mobley WC, Yu YE. Effects of individual segmental trisomies of human chromosome 21 syntenic regions on hippocampal long-term potentiation and cognitive behaviors in mice. *Brain research.* 2010;1366:162-71. doi: 10.1016/j.brainres.2010.09.107. PubMed PMID: 20932954; PMCID: 3027718.
 15. Aziz NM, Guedj F, Pennings JLA, Olmos-Serrano JL, Siegel A, Haydar TF, Bianchi DW. Lifespan analysis of brain development, gene expression and behavioral phenotypes in the Ts1Cje, Ts65Dn and Dp(16)1/Yey mouse models of Down syndrome. *Dis Model Mech.* 2018;11(6). Epub 2018/05/03. doi: 10.1242/dmm.031013. PubMed PMID: 29716957; PMCID: PMC6031353.
 16. Chen W, Liu B, Li X, Wang P, Wang B. Sex Differences in Spatial Memory. *Neuroscience.* 2020;443:140-7. Epub 2020/07/28. doi: 10.1016/j.neuroscience.2020.06.016. PubMed PMID: 32710913.
 17. Yang Y, Conners FA, Merrill EC. Visuo-spatial ability in individuals with Down syndrome: is it really a strength? *Res Dev Disabil.* 2014;35(7):1473-500. Epub 2014/04/24. doi: 10.1016/j.ridd.2014.04.002. PubMed PMID: 24755229; PMCID: PMC4041586.
 18. Kittler P, Krinsky-McHale SJ, Devenny DA. Sex differences in performance over 7 years on the Wechsler Intelligence Scale for Children--Revised among adults with intellectual disability. *J Intellect Disabil Res.* 2004;48(Pt 2):114-22. Epub 2004/01/16. doi: 10.1111/j.1365-2788.2004.00500.x. PubMed PMID: 14723654.
 19. Del Hoyo Soriano L, Thurman AJ, Abbeduto L. Specificity: A Phenotypic Comparison of Communication-Relevant Domains Between Youth With Down Syndrome and Fragile X Syndrome. *Front Genet.* 2018;9:424. Epub 2018/10/18. doi: 10.3389/fgene.2018.00424. PubMed PMID: 30327664; PMCID: PMC6174242.
 20. Love MI, Huber W, Anders S. Moderated estimation of fold change and dispersion for RNA-seq data with DESeq2. *Genome biology.* 2014;15(12):550. doi: 10.1186/s13059-014-0550-8. PubMed PMID: 25516281; PMCID: 4302049.
 21. Sullivan KD, Lewis HC, Hill AA, Pandey A, Jackson LP, Cabral JM, Smith KP, Liggett LA, Gomez EB, Galbraith MD, DeGregori J, Espinosa JM. Trisomy 21 consistently activates the

- interferon response. *Elife*. 2016;5. Epub 2016/07/30. doi: 10.7554/eLife.16220. PubMed PMID: 27472900; PMCID: PMC5012864.
22. Terry M. Therneau PMG. *Modeling Survival Data: Extending the Cox Model*. . 1 ed: Springer, New York, NY; 2000. XIV, 350 p.
23. Therneau TM, Grambsch PM, Pankratz VS. Penalized Survival Models and Frailty. *Journal of Computational and Graphical Statistics*. 2003;12(1):156-75. doi: 10.1198/1061860031365.
24. Piepho H-P. An Algorithm for a Letter-Based Representation of All-Pairwise Comparisons. *Journal of Computational and Graphical Statistics*. 2004;13(2):456-66. doi: 10.1198/1061860043515.
25. David Robinson AHaSC. *Broom: Convert Statistical Objects into Tidy Tibbles*. R package version 0.7.9.
. <https://CRAN.R-project.org/package=broom2021>.
26. Williams BR, Prabhu VR, Hunter KE, Glazier CM, Whittaker CA, Housman DE, Amon A. Aneuploidy affects proliferation and spontaneous immortalization in mammalian cells. *Science*. 2008;322(5902):703-9. Epub 2008/11/01. doi: 10.1126/science.1160058. PubMed PMID: 18974345; PMCID: PMC2701511.
27. Krivega M, Stiefel CM, Karbassi S, Andersen LL, Chunduri NK, Donnelly N, Pichlmair A, Storchova Z. Genotoxic stress in constitutive trisomies induces autophagy and the innate immune response via the cGAS-STING pathway. *Commun Biol*. 2021;4(1):831. Epub 2021/07/04. doi: 10.1038/s42003-021-02278-9. PubMed PMID: 34215848; PMCID: PMC8253785.
28. Pradeu T, Cooper EL. The danger theory: 20 years later. *Front Immunol*. 2012;3:287. Epub 2012/10/13. doi: 10.3389/fimmu.2012.00287. PubMed PMID: 23060876; PMCID: PMC3443751.
29. Ding J, Maxwell A, Adzibolosu N, Hu A, You Y, Liao A, Mor G. Mechanisms of immune regulation by the placenta: Role of type I interferon and interferon-stimulated genes signalling during pregnancy. *Immunol Rev*. 2022;308(1):9-24. Epub 2022/03/21. doi: 10.1111/imr.13077. PubMed PMID: 35306673; PMCID: PMC9189063.
30. Guedj F, Siegel AE, Pennings JLA, Alsebaa F, Massingham LJ, Tantravahi U, Bianchi DW. Apigenin as a Candidate Prenatal Treatment for Trisomy 21: Effects in Human Amniocytes and the Ts1Cje Mouse Model. *Am J Hum Genet*. 2020;107(5):911-31. Epub 2020/10/26. doi: 10.1016/j.ajhg.2020.10.001. PubMed PMID: 33098770; PMCID: PMC7675036.
31. Pinto B, Morelli G, Rastogi M, Savardi A, Fumagalli A, Petretto A, Bartolucci M, Varea E, Catelani T, Contestabile A, Perlini LE, Cancedda L. Rescuing Over-activated Microglia Restores Cognitive Performance in Juvenile Animals of the Dp(16) Mouse Model of Down

Syndrome. *Neuron*. 2020;108(5):887-904 e12. Epub 2020/10/08. doi: 10.1016/j.neuron.2020.09.010. PubMed PMID: 33027640; PMCID: PMC7736620.

Decision Letter, first revision:

4th Jan 2023

Dear Dr Espinosa,

Your Article, "Interferon receptor locus dosage determines diverse hallmarks of Down syndrome" has now been seen by 4 referees (Reviewers #3 and #4 reviewed together). You will see from their comments below that while they find your work of interest, some important points are raised. We are interested in the possibility of publishing your study in *Nature Genetics*, but would like to consider your response to these concerns in the form of a revised manuscript before we make a final decision on publication.

You'll see that Reviewer #1 remains unconvinced by some of your genetics data, whereas Reviewers #3 and 4 voice issues with the transcriptomic data. At this stage, we are not asking for further experiments (although you are welcome to include new data if you wish), but we do think that you need to carefully go through the manuscript and tone down some of your conclusions (Reviewers #3,4 provide some suggestions).

We therefore invite you to revise your manuscript taking into account all reviewer and editor comments. Please highlight all changes in the manuscript text file. At this stage we will need you to upload a copy of the manuscript in MS Word .docx or similar editable format.

*2) If you have not done so already please begin to revise your manuscript so that it conforms to our Article format instructions, available

http://www.nature.com/ng/authors/article_types/index.html here.

*3) Include a revised version of any required Reporting Summary:

[redacted]

We hope to receive your revised manuscript within four to eight weeks but we can be flexible with this deadline. If you cannot send it within this time, please let us know.

Sincerely,

Safia Danovi
Editor
Nature Genetics

Reviewers' Comments:

Reviewer #1:
Remarks to the Author:

I would like to thank the authors for the additional experiments performed and for the explanations offered.

Whilst some of my concern were adequately addressed I am afraid that key aspect of the study which is the IFN genes are major drivers of cardiac and perhaps cognitive issues remains largely the same as in the first iteration.

For me to judge this in earnest as a cause of pathology I simply need more suggested experiments that would address this directly.

Reviewer #2:

Remarks to the Author:

I really enjoyed the initial manuscript and found the conclusions to be a potentially important contribution. In this revision, the authors added significant new data including comprehensive transcriptome analysis of their new mouse model that further strengthens their conclusions.

In addition, the authors addressed all of my other minor concerns.

In light of these changes, I think this manuscript should be published. I believe that it will be a major contribution to the field!

Reviewer #3:

Remarks to the Author:

Re-review of Waugh, Espinosa Interferon Dp16 Mouse

Overall, I continue to think that the main novelty and important part of this study is the creation and phenotypic analysis of Dp16 mice for which the IFNR gene cluster has been deleted (reduced to disomy). The data showing normalization of heart defects (now Figure 4 D) is particularly striking and appears to be done on a sufficient number of animals to support that triplication of the IFNR locus is a necessary component of the Dp16 congenital heart defects. Since the heart tissue is removed from mice and dissected, investigators would do this analysis blind. It is also an important finding that craniofacial abnormalities and cognitive defects are substantially mitigated, although I find analysis of those traits potentially a little less "concrete" because they cannot be done blind (given obvious size difference between mice). (So many studies of DS mice have reported correction of cognitive deficits, even though they target different mechanisms!) Nonetheless, this phenotypic part of the study, particularly the heart, shows very important, novel observations for the field with the high significance appropriate for Nature Genetics.

While I recognize that the new experimental work added was to expand the transcriptomics analysis, I regrettably must say that this part of the study is less conclusive and impactful in my opinion (and, independently, that of the co-reviewer). Some of these results do support a correlation between expression of one or more of the IFNR genes and other non-Chr21 DEGs and pathways, but this largely makes a similar point to prior studies from this group which focus on interferon-related genes in trisomic samples. These findings are correlative and there are substantial examples that do not fit this theme, so these results and interpretations are a less clear (even "muddy") part of the study.

With more data presented than readers can likely digest fully, the data and interpretation that best fits the "IFNR dosage" thesis is emphasized. It is only because of the mouse knock-out of the IFNR cluster that the link between phenotypes and IFNR dosage is persuasive, not because of the transcriptomics.

For example, the added data or responses do not really address the question posed in my last review. Point 1 in my prior review pointed to a discrepancy in Fig 3F (which is now Fig 4F):

"However, a significant question comes up in Figure 3F. In transcriptomic analysis of E15.5 heart tissues, why are the differences in DP16/WT comparisons almost the same as in Dp162xIFNRs/WT comparisons? The IFNR-corrected Dp16 mice transcriptomes should be more similar to WT if so much is driven by the IFNR genes, but these results suggest that very similar pathways are dysregulated by other mechanisms independent of trisomy for the IFNR cluster. I understand that the Dp16 versus Dp162xIFNRs comparison shows some difference, but it is surprising and important that this data shows that mice with normalized IFNR gene dosage still have dysregulation of the key pathways that include: IFN alpha response, IFN gamma response, inflammatory response, IL6 Jak Stat3 signaling, IL2 Jak Stat2 signaling. These pathways seem central to this paper, but this point is not discussed. Is it a discrepancy that warrants substantial softening of the impression that inflammation is driven by or mostly by the IFNR gene cluster, or how do the authors explain it?"

In response the authors state that they agree with this "important" point, but the caveat added to the text seems to understand the degree of disconnect (shown in Fig 4F) and its significance. "In Dp16 247 2xIfnrs mice, many, but certainly not all, of 248 these gene expression changes are attenuated....". However, Figure 4F seems to show that correction of the IFNR gene dosage in DP16 mice did not reduce the high inflammatory signatures much relative to wild-type levels. (Results seem to indicate that the DP16 inflammatory pathways based on RNAseq are largely due to other genes on Chr21.) Now that more space is devoted to transcriptomics, this point is made more central to the study, as it is framed as addressing the "mechanism" of how IFNR dosage impacts DS phenotypes. But the additional transcriptome analysis on different tissues mostly just further shows how variable (and unpredictable) these pathways and relationships are. Another important example of this is the inexplicable drop in Stat-phosphorylation in the IFNR corrected mice, well below wild-type levels (Figure 4b).

Specific Recommendations:

1. The authors have done an elegant experiment to delete the IFNR cluster from the DP16 mice and show mitigation of phenotypes. This is the heart of this study, and it is not necessary to claim that the transcriptomics shows IFNR genes control certain pathways, because the gene deletion connects IFNR dosage to phenotypes. I realize more transcriptomics may have been added in response to a reviewer, but transcriptomics will be impacted by differences in tissue samples, cell composition, or state of individual blood samples, which may explain the variation in chr21 gene expression, or the correlation of certain chr21 genes with certain DEGs etc. The tendency to interpret this part of the study in a way that minimizes the substantial caveats can detract from the more important and clearer findings. In final editing I recommend the authors go through the text with this in mind....the strong evidence that the IFNR genes cause part of the DS phenotype is the phenotypic analysis of the engineered mouse, not really the transcriptomics.
2. Related to above, in response the authors state that a central finding is that phenotypes are corrected by IFNR dosage even though much of the transcriptome differences are not. But this could be stated more clearly as a "finding" in results rather than just mention it as an "aside" after conveying that the main finding is that the IFNR genes are linked to the transcriptomic inflammatory pathways.
3. A sentence has now been added to the Discussion acknowledging that the results show IFNR triplication is a necessary contributor (for DS malformations), but not necessarily sufficient....other chr21 genes may well contribute. The title should be modified accordingly from "IFNR dosage determines" to "IFNR dosage is a determinant of" or "contributes to" or "is necessary for".
4. I did not readily find (in Results or Methods) the information as to how many mice were analyzed

for the craniofacial abnormalities. The number of mice analyzed for each phenotypic assay should be on the figure or in the results section (as well as Methods), as this is critical to the strength of the analysis. If the number for cranio-facial abnormalities was small, the conclusion should be softened accordingly.

5. The Discussion seems to advocate the potential of therapeutic interventions with anti-inflammatories based on these (and other) findings. It would be prudent to mention in this context that other studies have reported mitigation of DS mouse model phenotypes based on several different mechanisms and drugs, and it remains to be seen which may be relevant to humans.

Despite reservations about part of this study, I remain enthusiastic about the major novel findings that alone are worthy of publication.

Reviewer #4:

Remarks to the Author:

Overall, I continue to think that the main novelty and important point here is the creation and phenotypic analysis of Dp16 mice for which the IFNR gene cluster has been deleted. The data showing normalization of heart defects (now Figure 4 D) is particularly striking and appears to be done on a sufficient number of animals to support the important conclusion: that triplication of the IFNR locus is a necessary component of the Dp16 congenital heart defects (of similar type to human DS). It is also noteworthy that craniofacial abnormalities and cognitive defects are largely mitigated, although I find analysis of those traits potentially a little less "concrete" in that they cannot be done blind (given obvious size difference between mice). Nonetheless, very important observations for the field that have significance appropriate for Nature Genetics.

As stated in prior review, the part of the study focused on transcriptomic changes in human blood or mice is less novel and less persuasive to this reviewer (and independently also to the co-reviewer). The authors did significant work to add more transcriptomic data on different tissues and ages, which I realize may have been prompted by questions from one reviewer. However, for the most part the additional transcriptomics results reinforce a point made in my prior review, that the transcriptomic data was over-emphasized and interpreted to support that correction of the IFNR gene dosage corrected inflammatory pathways. It is hard for any reader to fully digest all these transcriptomic results but they appear to be "all over the place", in terms of which Interferon/inflammation genes/pathways are impacted. It also does not really address the question posed in the last review. Point 1 in my prior review pointed to a discrepancy in Fig 3F (which is now Fig 4F):

"However, a significant question comes up in Figure 3F. In transcriptomic analysis of E15.5 heart tissues, why are the differences in DP16/WT comparisons almost the same as in Dp162xIFNRs/WT comparisons? The IFNR-corrected Dp16 mice transcriptomes should be more similar to WT if so much is driven by the IFNR genes, but these results suggest that very similar pathways are dysregulated by other mechanisms independent of trisomy for the IFNR cluster. I understand that the Dp16 versus Dp162xIFNRs comparison shows some difference, but it is surprising and important that this data shows that mice with normalized IFNR gene dosage still have dysregulation of the key pathways that include: IFN alpha response, IFN gamma response, inflammatory response, IL6 Jak Stat3 signaling, IL2 Jak Stat2 signaling. These pathways seem central to this paper, but this point is not discussed. Is it a discrepancy that warrants substantial softening of the impression that inflammation is driven by or mostly by the IFNR gene cluster, or how do the authors explain it?"

In response the authors state that they agree with this and agree it is an important point, so they

state in the text that there are interferon-independent inflammatory signatures (presumably due to other chr21 genes, or extra DNA?). Expanded transcriptome analysis on different tissues further shows how variable (and unpredictable) , and which genes/pathways varies (unpredictably) in different tissues and ages etc. While they now mention this "caveat" in the text, it still seems that there is imbalanced emphasis (cherry-picking?) on examples that can support a relationship of some inflammation-related genes/pathway to IFNR dosage, even though there seems perhaps similar weight of evidence to the contrary. As noted in the first review, this point seems central to the thesis of this paper, particularly because the so much space and data is devoted to arguing that various transcriptome changes are due to IFNR gene dosage. This part of this study depends on more strained interpretations of complex data, and in our view is the "muddier" (and less novel) aspect of this study. Title should be reflect the change in claim (correct), that IFNR cluster is necessary, does not show that it is sufficient. So "determines" seems to suggest it is sufficient. Therefore, change to "is a determinant of" or "is necessary for"

1) First sub-heading "SOME or SEVERAL inflammatory signatures correlate with IFNR gene triplication

Author Rebuttal, first revision:

Response to Referees – Manuscript NG-A60025-T by Waugh et al.

Reviewer #1.

Remarks to the Author

I would like to thank the authors for the additional experiments performed and for the explanations offered.

Whilst some of my concern were adequately addressed I am afraid that key aspect of the study which is the IFN genes are major drivers of cardiac and perhaps cognitive issues remains largely the same as in the first iteration.

For me to judge this in earnest as a cause of pathology I simply need more suggested experiments that would address this directly.

Response: We are grateful for the overall positive assessment and constructive critiques by Reviewer #1.

First, we would like to note that the revised manuscript has been thoroughly revised to make sure that there is no description of individual '*Ifnr* genes' as the major drivers of the phenotypes studied, referring instead to the '*Ifnr* locus' as a whole. These modifications have been introduced to the title and abstract as well. With that being said, we believe that the most parsimonious interpretation of our results is that overexpression of one or more of the *Ifnrs* is contributing to the phenotypes studied. While we agree with the Reviewer that we cannot fully disregard other effects of the 192 kb genomic deletion (e.g., non-coding RNAs, cis-regulatory elements), such alternative explanations are, in our opinion, less likely.

With regards to additional experiments that would more clearly tie the individual *Ifnr* genes to the various phenotypes, we described in our previous response the difficulties of performing experiments

with anti-IFNR antibodies as originally proposed by the Reviewer. As the Reviewer noted initially, *'this might be difficult in the developing embryo'*.

The next logical step would be to employ knockout mice for the individual *Ifnr* genes (i.e., *Ifnra1*^{-/-}, *Ifnr2*^{-/-}, *Ifngr2*^{-/-}, *Il10rb*^{-/-}) using short deletions or point mutations. The results in our manuscript justify the pursuit of these additional experiments by others in the field and our own team in the near future, but we believe that embarking on these expensive and lengthy experiments at this point would cause unnecessarily delays in the publication of the important results in the current manuscript.

In a way, the Reviewer is reviving an important discussion that our team had nearly six years ago at the onset of the project, when we had to decide whether to knock out all four *Ifnr* genes at once, or to test the contributions of individual *Ifnr* genes. After much deliberation, the potential for cooperative and redundant effects of the four *Ifnrs* on downstream signaling pathways guided our decision that it would be more valuable to investigate the impacts of deleting the entire locus first. Then, if interesting results were obtained, to embark on subsequent testing of the individual *Ifnr* gene knockouts.

With regards to other additional experiments that could further dissect a cause-effect relationship between *Ifnr* overexpression and the diverse phenotypes studied, we believe the extensive transcriptome analysis provided in this manuscript will guide us and others in the design and completion of these follow up experiments. For example, in the developing heart, correction of *Ifnr* locus dosage attenuates gene expression changes indicative of decreased cell proliferation. These results can then help others in the field and our own team investigate this phenomenon in depth. For example, which IFNRs, IFN ligands, and downstream JAK kinases are involved in this phenomenon? What cell types are most affected by such impaired cell proliferation? What IFN-stimulated genes may cause decreased proliferation in the developing heart? Clearly, the possibilities here are many and fascinating, and we look forward to embarking on these investigations in the future.

While we empathize with the Reviewer's desire to see more experiments for some of the phenotypes described (e.g., cardiac, cognitive), and we share the same curiosities, we adhere to our decision to describe in a single manuscript all the phenotypes that we found were affected by *Ifnr* locus dosage, at the expense of fully characterizing each individual phenotype. We hope the Reviewer will agree that an extensive investigation of each phenotype affected by the *Ifnr* locus will be worthy of future investigations described in separate publications.

Prompted by the Reviewer's comment, we have revised the text, both in the Results and Discussion sections, to acknowledge the current limitations of the study and the need for additional research. For example, key passages that directly address the Reviewer's comments are:

Opening paragraph of the Discussion: *Nevertheless, it is also possible that some of the effects observed are due to other events affected by triplication of this 192 kb genomic locus, including potential contributions from non-coding RNAs and cis-regulatory elements.*

Later in the Discussion, when discussing the cardiac phenotypes: *Therefore, additional research is needed to illuminate the mechanisms by which the *Ifnr* locus contributes to CHD, including a dissection of the roles of specific *Ifnr* genes, IFN ligands, downstream signalling cascades, and ISGs in heart development.*

Once more in the Discussion, near the final closing statement: *An increasing appreciation about the role of maternal and foetal antiviral immunity in development of congenital disease supports the need for additional research to dissect which of the IFNRs, their specific ligands, and downstream kinases contribute to the diverse phenotypes of DS, as well as the safest and most effective therapeutic interventions.*

Reviewer #2.Remarks to the Author

I really enjoyed the initial manuscript and found the conclusions to be a potentially important contribution. In this revision, the authors added significant new data including comprehensive transcriptome analysis of their new mouse model that further strengthens their conclusions. In addition, the authors addressed all of my other minor concerns.

In light of these changes, I think this manuscript should be published. I believe that it will be a major contribution to the field!

Response: We are grateful for the overall positive assessment from Reviewer #2.

Reviewer #3Remarks to the Author

Overall, I continue to think that the main novelty and important part of this study is the creation and phenotypic analysis of Dp16 mice for which the INFR gene cluster has been deleted (reduced to disomy). The data showing normalization of heart defects (now Figure 4 D) is particularly striking and appears to be done on a sufficient number of animals to support that triplication of the IFNR locus is a necessary component of the Dp16 congenital heart defects. Since the heart tissue is removed from mice and dissected, investigators would do this analysis blind. It is also an important finding that craniofacial abnormalities and cognitive defects are substantially mitigated, although I find analysis of those traits potentially a little less “concrete” because they cannot be done blind (given obvious size difference between mice). (So many studies of DS mice have reported correction of cognitive deficits, even though they target different mechanisms!) Nonetheless, this phenotypic part of the study, particularly the heart, shows very important, novel observations for the field with the high significance appropriate for Nature Genetics.

While I recognize that the new experimental work added was to expand the transcriptomics analysis, I regrettably must say that this part of the study is less conclusive and impactful in my opinion (and, independently, that of the co-reviewer). Some of these results do support a correlation between expression of one or more of the IFNR genes and other non-Chr21 DEGs and pathways, but this largely makes a similar point to prior studies from this group which focus on interferon-related genes in trisomic samples. These findings are correlative and there are substantial examples that do not fit this theme, so these results and interpretations are a less clear (even “muddy”) part of the study. With more data presented than readers can likely digest fully, the data and interpretation that best fits the “IFNR dosage” thesis is emphasized. It is only because of the mouse knock-out of the IFNR cluster that the link between phenotypes and IFNR dosage is persuasive, not because of the transcriptomics. For example, the added data or responses do not really address the question posed in my last review.

Point 1 in my prior review pointed to a discrepancy in Fig 3F (which is now Fig 4F):

“However, a significant question comes up in Figure 3F. In transcriptomic analysis of E15.5 heart tissues, why are the differences in DP16/WT comparisons almost the same as in Dp162xIFNRs/WT comparisons? The IFNR-corrected Dp16 mice transcriptomes should be more similar to WT if so much is driven by the IFNR genes, but these results suggest that very similar pathways are dysregulated by other mechanisms independent of trisomy for the IFNR cluster. I understand that the Dp16 versus Dp162xIFNRs comparison shows some difference, but it is surprising and important that this data shows that mice with normalized IFNR gene dosage still have dysregulation of the key pathways that include: IFN alpha response, IFN gamma response, inflammatory response, IL6 Jak Stat3 signaling, IL2 Jak Stat2 signaling. These pathways seem central to this paper, but this point is not discussed. Is it a discrepancy that warrants substantial softening of the impression that inflammation is driven by or mostly by the IFNR gene cluster, or how do the authors explain it?”

In response the authors state that they agree with this “important” point, but the caveat added to the text seems to understand the degree of disconnect (shown in Fig 4F) and its significance. “In Dp16 247 2xIfnrs mice, many, but certainly not all, of 248 these gene expression changes are attenuated....”. However, Figure 4F seems to show that correction of the IFNR gene dosage in DP16 mice did not reduce the high inflammatory signatures much relative to wild-type levels. (Results seem to indicate that the DP16 inflammatory pathways based on RNAseq are largely due to other genes on Chr21.) Now that more space is devoted to transcriptomics, this point is made more central to the study, as it is framed as addressing the “mechanism” of how IFNR dosage impacts DS phenotypes. But the additional transcriptome analysis on different tissues mostly just further shows how variable (and unpredictable) these pathways and relationships are. Another important example of this is the inexplicable drop in Stat-phosphorylation in the IFNR corrected mice, well below wild-type levels (Figure 4b).

Response: We are grateful for Reviewer #3’s additional comments. As the Reviewer well notes, not all gene expression changes (including inflammatory gene signatures) observed in the transcriptome of the Dp16 mice are corrected by reduction of *Ifnr* locus dosage, pointing to the fact that other triplicated genes also contribute to the global changes generally and inflammatory signatures specifically.

Importantly, these results indicate that the phenotypes of Down syndrome could be ameliorated (even if partially) by interventions that only mildly attenuate gene expression changes in select dysregulated pathways. As the Reviewer noted, most of the gene expression changes caused by the genomic triplication in Dp16 are still observed in the Dp16^{2xIfnrs} mice, which is expected based on the fact that all other ~120 triplicated genes are still overexpressed in Dp16^{2xIfnrs} mice, but with noticeable attenuation in specific gene signatures. These results indicate that normalization of *Ifnr* locus dosage can affect phenotypes *without* a complete rescue of underlying gene expression changes caused by the trisomy, and even *without* fully rescuing inflammatory signatures. We believe the therapeutic implications of these findings are profound: the phenotypes of Down syndrome could be potentially ameliorated by targeting a few master regulators encoded on chromosome 21 (e.g., IFNRs, DYRK1A) without having to necessarily correct all processes dysregulated by the trisomy. We believe this conclusion is a source of optimism for the field and justifies the pursuit of therapies targeting key gene products encoded on chromosome 21.

Following Reviewer's guidance, we have revised the text to make these points more explicit, including the need to look beyond IFN signaling as a source of immune dysregulation in Down syndrome (please see tracked changes in the revised manuscript file).

With regards to the value of the extensive additional transcriptome studies included in the revised manuscript, which we completed following Reviewer #2's guidance, we believe these are an important addition to the manuscript, as they produce two key observations:

- 1) The impact of *Ifnr* locus dosage on global transcriptome changes varies strongly from tissue to tissue, but also across development within the same tissue, as demonstrated by the three transcriptome studies of embryonic and adult heart tissue.
- 2) The specific signaling pathways sensitive to *Ifnr* locus dosage also vary across tissues, an observation that will greatly aid future investigations to understand how elevated IFN signaling affects organ development and function in Down syndrome. For example, in the developing heart and embryonic facial mesenchyme, elevated IFN signaling may cause a decrease in cell proliferation, whereas in adult brain tissue, IFN signaling may cause dysregulation of synaptogenesis and dopamine signaling.

While we agree with the Reviewer that the wealth of transcriptome data in the revised manuscript creates challenges in terms of narrative ('muddy', as the Reviewer would say), we also believe that these datasets will provide a very important resource to the field. Even without the Dp16^{2xIfnrs} strain, these transcriptome studies will greatly aid the field in understanding which pathways are dysregulated by the trisomy in different tissues.

Specific Recommendations:

1. The authors have done an elegant experiment to delete the IFNR cluster from the DP16 mice and show mitigation of phenotypes. This is the heart of this study, and it is not necessary to claim that the transcriptomics shows IFNR genes control certain pathways, because the gene deletion connects IFNR dosage to phenotypes. I realize more transcriptomics may have been added in response to a reviewer,

but transcriptomics will be impacted by differences in tissue samples, cell composition, or state of individual blood samples, which may explain the variation in chr21 gene expression, or the correlation of certain chr21 genes with certain DEGs etc. The tendency to interpret this part of the study in a way that minimizes the substantial caveats can detract from the more important and clearer findings. In final editing I recommend the authors go through the text with this in mind....the strong evidence that the IFNR genes cause part of the DS phenotype is the phenotypic analysis of the engineered mouse, not really the transcriptomics.

Response: We appreciate this comment and agree with the Reviewer: it is the phenotypic analysis of the novel Dp16^{2xIfnrs} strain, not the accompanying transcriptome data, that constitutes the backbone of the manuscript. We have revised the text to highlight this fact.

2. Related to above, in response the authors state that a central finding is that phenotypes are corrected by IFNR dosage even though much of the transcriptome differences are not. But this could be stated more clearly as a “finding” in results rather than just mention it as an “aside” after conveying that the main finding is that the IFNR genes are linked to the transcriptomic inflammatory pathways.

Response: Thank you for this comment, we have modified the Results section to highlight this finding as suggested, rather than a commentary in the Discussion, along the lines of our response above. We have done this in multiple instances throughout the various chapters of the Results section (see tracked changes in the manuscript file).

3. A sentence has now been added to the Discussion acknowledging that the results show IFNR triplication is a necessary contributor (for DS malformations), but not necessarily sufficient....other chr21 genes may well contribute. The title should be modified accordingly from “IFNR dosage determines” to “IFNR dosage is a determinant of” or “contributes to” or “is necessary for”.

Response: Thank you. The new title now reads: *Interferon receptor locus dosage contributes to diverse hallmarks of Down syndrome.*

4. I did not readily find (in Results or Methods) the information as to how many mice were analyzed for the craniofacial abnormalities. The number of mice analyzed for each phenotypic assay should be on the figure or in the results section (as well as Methods), as this is critical to the strength of the analysis. If the number for cranio-facial abnormalities was small, the conclusion should be softened accordingly.

Response: Thank you. We have made sure that all numbers of animals are described in the corresponding figure legends as instructed by editorial formatting guidelines. The number for the analysis of cranio-facial abnormalities was 6-7 animals per group as indicated in Figure 6 legend.

5. The Discussion seems to advocate the potential of therapeutic interventions with anti-inflammatories based on these (and other) findings. It would be prudent to mention in this context that other studies have reported mitigation of DS mouse model phenotypes based on several different mechanisms and drugs, and it remains to be seen which may be relevant to humans.

Response: We agree with this comment, in the sense that anti-inflammatory strategies are just one potential avenue for therapeutic intervention in Down syndrome. We have revised the Discussion to acknowledge the existence of other valid strategies supported by results in animal models and pointed readers to important reviews summarizing the knowledge in this area.

Despite reservations about part of this study, I remain enthusiastic about the major novel findings that alone are worthy of publication.

Reviewer #4:

Remarks to the Author:

Overall, I continue to think that the main novelty and important point here is the creation and phenotypic analysis of Dp16 mice for which the IFNR gene cluster has been deleted. The data showing normalization of heart defects (now Figure 4 D) is particularly striking and appears to be done on a sufficient number of animals to support the important conclusion: that triplication of the IFNR locus is a necessary component of the Dp16 congenital heart defects (of similar type to human DS). It is also noteworthy that craniofacial abnormalities and cognitive defects are largely mitigated, although I find analysis of those traits potentially a little less “concrete” in that they cannot be done blind (given obvious size difference between mice). Nonetheless, very important observations for the field that have significance appropriate for Nature Genetics.

As stated in prior review, the part of the study focused on transcriptomic changes in human blood or mice is less novel and less persuasive to this reviewer (and independently also to the co-reviewer). The authors did significant work to add more transcriptomic data on different tissues and ages, which I realize may have been prompted by questions from one reviewer. However, for the most part the additional transcriptomics results reinforce a point made in my prior review, that the transcriptomic data was over-emphasized and interpreted to support that correction of the IFNR gene dosage corrected inflammatory pathways. It is hard for any reader to fully digest all these transcriptomic results but they appear to be “all over the place”, in terms of which Interferon/inflammation genes/pathways are impacted. It also does not really address the question posed in the last review.

Point 1 in my prior review pointed to a discrepancy in Fig 3F (which is now Fig 4F):

“However, a significant question comes up in Figure 3F. In transcriptomic analysis of E15.5 heart tissues, why are the differences in DP16/WT comparisons almost the same as in Dp162xIFNRs/WT comparisons? The IFNR-corrected Dp16 mice transcriptomes should be more similar to WT if so much is driven by the IFNR genes, but these results suggest that very similar pathways are dysregulated by other mechanisms independent of trisomy for the IFNR cluster. I understand that the Dp16 versus Dp162xIFNRs comparison shows some difference, but it is surprising and important that this data shows that mice with normalized IFNR gene dosage still have dysregulation of the key pathways that include: IFN alpha response, IFN gamma response, inflammatory response, IL6 Jak Stat3 signaling, IL2 Jak Stat2 signaling.

These pathways seem central to this paper, but this point is not discussed. Is it a discrepancy that warrants substantial softening of the impression that inflammation is driven by or mostly by the IFNR gene cluster, or how do the authors explain it?"

In response the authors state that they agree with this and agree it is an important point, so they state in the text that there are interferon-independent inflammatory signatures (presumably due to other chr21 genes, or extra DNA?). Expanded transcriptome analysis on different tissues further shows how variable (and unpredictable) , and which genes/pathways varies (unpredictably) in different tissues and ages etc. While they now mention this "caveat" in the text, it still seems that there is imbalanced emphasis (cherry-picking?) on examples that can support a relationship of some inflammation-related genes/pathway to IFNR dosage, even though there seems perhaps similar weight of evidence to the contrary. As noted in the first review, this point seems central to the thesis of this paper, particularly because the so much space and data is devoted to arguing that various transcriptome changes are due to IFNR gene dosage. This part of this study depends on more strained interpretations of complex data, and in our view is the "muddier" (and less novel) aspect of this study.

Response: We are grateful for Reviewer #4's additional comments, and our response is similar to that offered to Reviewer #3 above, with some additions. As the Reviewer well notes, not all gene expression changes (including inflammatory gene signatures) observed in the transcriptome of the Dp16 mice are corrected by reduction of *Ifrn* locus dosage, pointing to the fact that other triplicated genes may also contribute to the global changes generally and inflammatory signatures specifically. Please see our response to Reviewer #3 above for more on this topic.

With regards to the examples that we selected from the transcriptome analysis (the comment about 'cherry picking'), our choice of examples was guided by the results of the pathway analysis with an emphasis on pathways dysregulated in Dp16 but attenuated in Dp16^{2xifnr}. For example, in lymph nodes, this criteria led us to focus on gene signatures associated with cell proliferation (e.g., E2F targets), IL2 STAT5 signaling and oxidative phosphorylation, in the developing heart this led to the choice of genes involved in the Interferon Alpha Response and cell proliferation (e.g., MYC targets), and in the brain this criteria led us to focus on synaptogenesis, SNARE signaling and dopamine signaling. As the Reviewer appreciates already, we had to selected only key salient examples to prevent a rather overwhelming narrative.

Title should be reflect the change in claim (correct), that IFNR cluster is necessary, does not show that it is sufficient. So "determines" seems to suggest it is sufficient. Therefore, change to "is a determinant of" or "is necessary for"

Response: Thank you. Following guidance by Reviewer #3, the new title now reads: *Interferon receptor locus dosage contributes to diverse hallmarks of Down syndrome.*

1) First sub-heading "SOME or SEVERAL inflammatory signatures correlate with IFNR gene triplication"

Response: Thank you. The revised subheading now reads: *Specific inflammatory signatures correlate with IFNR overexpression in Down syndrome.*

Decision Letter, second revision:

22nd Feb 2023

Dear Dr Espinosa,

I hope you are well.

Your Article, "Interferon receptor locus dosage contributes to diverse hallmarks of Down syndrome" has now been seen by Reviewers #3 and #4 who (as per the previous round of review) have provided a joint report. You will see from their comments below that while they find your work of interest, some important points are raised. We are interested in the possibility of publishing your study in Nature Genetics, but would like to consider your response to these concerns in the form of a revised manuscript before we make a final decision on publication.

Overall, both reviewers agree that you have presented an important body of work that is likely to be of broad interest. However, they are concerned that the limitations of the work have not been appropriately defined and discussed. As such, their fear is that your data is open to over-interpretation by your readers. They have provided clear advice as to areas of the manuscript that need to be caveated and toned down and we would urge you to address their comments in full. Our intention would be to perform an in-house assessment of your changes when the revision comes back in but we might return to Reviewers #3 and #4 if necessary.

Please highlight all changes in the manuscript text file. At this stage we will need you to upload a copy of the manuscript in MS Word .docx or similar editable format.

*2) If you have not done so already please begin to revise your manuscript so that it conforms to our Article format instructions, available

http://www.nature.com/ng/authors/article_types/index.html here.

*3) Include a revised version of any required Reporting Summary:

[redacted]

We hope to receive your revised manuscript within four to eight weeks. If you cannot send it within this time, please let us know.

Sincerely,

Safia Danovi
Editor
Nature Genetics

Reviewers' Comments:

Reviewer #3:

Remarks to the Author:

Re-review of revised Nat Genet Jan 2023 IFNR genes in DS mouse model

This was a large study with a lot of dense data, so it took some time to carefully re-read the

manuscript and consider the prior reviews and whether responses were adequate, and a qualified colleague independently provided assessments. As requested by the editor, I also considered concerns of Reviewer 1 and responses to them. Both Reviewer 1 and our perspective agree that the most impactful and novel aspect of this study is the creation of a mouse model deleted for the IFNR cluster and the more significant and unexpected findings that this prevented congenital heart defects as well as many cognitive deficits (as measured in mice).

I previously commented that the transcriptomic data is quite inconsistent and rather “muddy” and that correlations do not provide strong evidence that the IFNR gene cluster causes the inflammatory signatures and immune-related pathways. The authors seem to agree that the findings are “muddy” but make the reasonable argument that it might be because the IFNR genes have different effects in different tissues at different times. This may be the explanation, and the authors have acknowledged these complications more in the manuscript, so they have sufficiently responded to those questions. I appreciate how much work this study represents, and I continue to feel it should be published in an impactful journal. However, in re-reading the study and various comments and responses, I have misgivings about this manuscript being published without more straightforward statements of the limitations. The phenotypic findings, if true in human Down syndrome, would be so important that the “take home” for this paper (or related news reports) will be read by families of people with DS, clinicians advising patients (about the proposed clinical trials!), biotech and pharmaceutical firms and investors, granting agencies etc. Even researchers in the DS field will not necessarily easily digest all the data and recognized substantial caveats. This concern might be most efficiently and effectively done with a “Limitations paragraph”, and some other edits, that make this caveats and questions clearer.

Regarding Reviewer 1 comments, I surmise that they were not satisfied with the answers to their initial questions and comments that current human findings, and certain mouse studies, do not support that the IFNR triplication/over-expression causes congenital heart defects. I thought these were good questions, and would have preferred clear acknowledgement of such questions and limitations. However, from what I could tell the authors seem to respond by citing a number of studies that very indirectly or loosely are consistent with their thesis about IFNR triplicationsuch as citing that rubella infection causing defects, or other types of IFN-related genes mutated causing effects. Reviewer 1 also cited a study which argued against this effect of IFNR triplication in a different DS mouse model and the authors explained this study may be underpowered (but didn’t acknowledge this significant question in the ms). I surmise that Reviewer 1 was not satisfied with these answers and thus in last round just briefly said they wanted more proof. Personally, I didn’t feel that knocking-out individual IFNR genes (as Reviewer 1 had suggested) should necessarily be required.....but on the other hand, I do think these major questions and caveats should be clearly acknowledged.

1. There should be a clear upfront statement (in limitations paragraph) that this evidence is for one of the trisomic mouse models..... and the title and abstract should not conflate “Down syndrome” (implying human) with results in a mouse model. I realize now that the way the study (and abstract) is constructed it begins with correlative human transcriptome evidence that in DS inflammatory signatures are correlated with IFNR over-expression, but this is not persuasive evidence that the immune/inflammatory pathway are caused by IFNR triplication. (IFNR expression would seem logically correlated with inflammatory/immune pathways.) So the mouse study is the key evidence on which they conclude that there findings now justify therapeutic intervention (in humans).....even prenatally”

2. The limitations paragraph should somehow convey, as I stated previously, that there have been numerous “cures” of DS mouse models, by various mechanisms, but thus far this has not lead to a significant human therapeutic. It is also the case that results in different mouse models or different studies may disagree. I am hopeful that this particular finding about IFNR cluster as underlying so

many main phenotypes will prove a lasting one and be reproduced by others. I agree that this group and some others have strongly shown that there is excess inflammation in people with DS, and it seems plausible that anti-inflammatories could have some modest effect, particularly on respiratory infections etc. So if they are known not to be harmful, this may be justified for testing for modest anti-inflammatory benefits. But this study will be read that there is a strong prospect that this will cure/mitigate so many phenotypes of DS.....it is important and fair to this population that any questions be made clear about whether IFNR triplication causes congenital heart and cognitive impact in humans.....before recommending prenatal treatments. I am concerned that the authors so clearly advocate that this mouse model finding justifies human therapeutics targeting IFNR pathways (JAK-stat etc.).....even when their results show how complicated these pathways are to predict.

3. In re-reading the manuscript, I noticed something related to a question I asked in my first review, regarding whether the monosomic IFNR mouse (used to breed and correct the trisomics) had any phenotype. I thought this interesting and relevant to whether IFNR genes are dosage sensitive. The authors responded that they didn't study those mice, and I didn't press this further, but in re-reading I see they don't even mention this in the manuscript, and, moreover, brief information appears to indicate that they first made a mouse with NULLISOMY for the IFNR cluster (double-knock out!!). This mouse was then bred to generate the monosomic WT mouse, which they then bred to the Dp16 mice to reduce IFNR copy number. Is this correct.....did they make and breed mice with complete loss of both IFNR cluster alleles? If so, that is remarkable.....and is totally relevant to a paper that is focused on how dosage sensitive is IFNR triplication. If they indeed began with a complete double IFNR knock-out, that should be in their diagram so it is not so hard to decipher in the manuscript. And something should be said that the monosomy or nullisomy for IFNR cluster does not have an obvious impact on development, phenotype or viability (but likely would I assume impact response to viruses). While this would not rule out a developmental impact of trisomy for IFNR, it does make such dosage sensitivity less likely or hard to understand and should be mentioned in this context.

4. A comment from my prior review stated: "In transcriptomic analysis of E15.5 heart tissues, why are the differences in DP16/WT comparisons almost the same as in Dp162xIFNRs/WT comparisons? The IFNR-corrected Dp16 mice transcriptomes should be more similar to WT if so much is driven by the IFNR genes, but these results suggest that very similar pathways are dysregulated by other mechanisms independent of trisomy for the IFNR cluster. I understand that the Dp16 versus Dp162xIFNRs comparison shows some difference, but it is surprising and important that this data shows that mice with normalized IFNR gene dosage still have dysregulation of the key pathways that include: IFN alpha response, IFN gamma response, inflammatory response, IL6 Jak Stat3 signalling, IL2 Jak Stat2 signalling. These pathways seem central to this paper, but this point is not discussed. Is it a discrepancy that warrants substantial softening of the impression that inflammation is driven by or mostly by the IFNR gene cluster, or how do the authors explain it.

I believe in revision the authors have addressed this in the text by stating the disconnect more minimally.... that "not all" of the transcriptomic dysregulated genes are caused by the IFNR pathways, hence naturally other Chr21 genes must contribute. But in their responses they acknowledged this more clearly that "only a small fraction of the transcriptome changes caused by the trisomy were due to IFNR" and that there was a "mild dampening in the dysregulation of these genes observed in Dp162xIfnrs". While I agree these findings can still be consistent with the severe phenotypic effects shown for IFNR, it does seem somewhat surprising..... and different than the human transcriptomics interpreted to be that IFNR genes correlate/cause these pathways. So, some reference to this question (with their possible explanation) might be also mentioned in a limitations paragraph.

Despite these significant questions, I support this study and the novel findings that might prove a

breakthrough relevant to human biology in DS. However my big caveat is that this study's big caveats should be made more transparently and clear for the audiences that will be impacted. The title should refer to a DS mouse model. The abstract should not say: "Here we demonstrate that triplication of the interferon receptor gene cluster on chromosome 21 is necessary for multiple hallmarks of DS." DS refers to humans and human Chr21 and DS mouse models and orthologous genes are not "DS" nor Chr21. Even for known single-gene defects in humans, precise mouse mutants don't always reflect the human biology. Here, establishing from a mouse model what Chr21 genes are key in human Down syndrome is much more indirect and complicated.

Reviewer #4:

Remarks to the Author:

Re-review of revised Nat Genet Jan 2023 IFNR genes in DS mouse model

This was a large study with a lot of dense data, so it took some time to carefully re-read the manuscript and consider the prior reviews and whether responses were adequate, and a qualified colleague independently provided assessments. As requested by the editor, I also considered concerns of Reviewer 1 and responses to them. Both Reviewer 1 and our perspective agree that the most impactful and novel aspect of this study is the creation of a mouse model deleted for the IFNR cluster and the more significant and unexpected findings that this prevented congenital heart defects as well as many cognitive deficits (as measured in mice).

I previously commented that the transcriptomic data is quite inconsistent and rather "muddy" and that correlations do not provide strong evidence that the IFNR gene cluster causes the inflammatory signatures and immune-related pathways. The authors seem to agree that the findings are "muddy" but make the reasonable argument that it might be because the IFNR genes have different effects in different tissues at different times. This may be the explanation, and the authors have acknowledged these complications more in the manuscript, so they have sufficiently responded to those questions. I appreciate how much work this study represents, and I continue to feel it should be published in an impactful journal. However, in re-reading the study and various comments and responses, I have misgivings about this manuscript being published without more straightforward statements of the limitations. The phenotypic findings, if true in human Down syndrome, would be so important that the "take home" for this paper (or related news reports) will be read by families of people with DS, clinicians advising patients (about the proposed clinical trials!), biotech and pharmaceutical firms and investors, granting agencies etc. Even researchers in the DS field will not necessarily easily digest all the data and recognized substantial caveats. This concern might be most efficiently and effectively done with a "Limitations paragraph", and some other edits, that make this caveats and questions clearer.

Regarding Reviewer 1 comments, I surmise that they were not satisfied with the answers to their initial questions and comments that current human findings, and certain mouse studies, do not support that the IFNR triplication/over-expression causes congenital heart defects. I thought these were good questions, and would have preferred clear acknowledgement of such questions and limitations. However, from what I could tell the authors seem to respond by citing a number of studies that very indirectly or loosely are consistent with their thesis about IFNR triplicationsuch as citing that rubella infection causing defects, or other types of IFN-related genes mutated causing effects. Reviewer 1 also cited a study which argued against this effect of IFNR triplication in a different DS mouse model and the authors explained this study may be underpowered (but didn't acknowledge this significant question in the ms). I surmise that Reviewer 1 was not satisfied with these answers and thus in last round just briefly said they wanted more proof. Personally, I didn't feel that knocking-out

individual IFNR genes (as Reviewer 1 had suggested) should necessarily be required....but on the other hand, I do think these major questions and caveats should be clearly acknowledged.

1. There should be a clear upfront statement (in limitations paragraph) that this evidence is for one of the trisomic mouse models..... and the title and abstract should not conflate "Down syndrome" (implying human) with results in a mouse model. I realize now that the way the study (and abstract) is constructed it begins with correlative human transcriptome evidence that in DS inflammatory signatures are correlated with IFNR over-expression, but this is not persuasive evidence that the immune/inflammatory pathway are caused by IFNR triplication. (IFNR expression would seem logically correlated with inflammatory/immune pathways.) So the mouse study is the key evidence on which they conclude that their findings now justify therapeutic intervention (in humans).....even prenatally"

2. The limitations paragraph should somehow convey, as I stated previously, that there have been numerous "cures" of DS mouse models, by various mechanisms, but thus far this has not led to a significant human therapeutic. It is also the case that results in different mouse models or different studies may disagree. I am hopeful that this particular finding about IFNR cluster as underlying so many main phenotypes will prove a lasting one and be reproduced by others. I agree that this group and some others have strongly shown that there is excess inflammation in people with DS, and it seems plausible that anti-inflammatories could have some modest effect, particularly on respiratory infections etc. So if they are known not to be harmful, this may be justified for testing for modest anti-inflammatory benefits. But this study will be read that there is a strong prospect that this will cure/mitigate so many phenotypes of DS.....it is important and fair to this population that any questions be made clear about whether IFNR triplication causes congenital heart and cognitive impact in humans.....before recommending prenatal treatments. I am concerned that the authors so clearly advocate that this mouse model finding justifies human therapeutics targeting IFNR pathways (JAK-stat etc.).....even when their results show how complicated these pathways are to predict.

3. In re-reading the manuscript, I noticed something related to a question I asked in my first review, regarding whether the monosomic IFNR mouse (used to breed and correct the trisomics) had any phenotype. I thought this interesting and relevant to whether IFNR genes are dosage sensitive. The authors responded that they didn't study those mice, and I didn't press this further, but in re-reading I see they don't even mention this in the manuscript, and, moreover, brief information appears to indicate that they first made a mouse with NULLISOMY for the IFNR cluster (double-knock out!!). This mouse was then bred to generate the monosomic WT mouse, which they then bred to the Dp16 mice to reduce IFNR copy number. Is this correct.....did they make and breed mice with complete loss of both IFNR cluster alleles? If so, that is remarkable.....and is totally relevant to a paper that is focused on how dosage sensitive is IFNR triplication. If they indeed began with a complete double IFNR knock-out, that should be in their diagram so it is not so hard to decipher in the manuscript. And something should be said that the monosomy or nullisomy for IFNR cluster does not have an obvious impact on development, phenotype or viability (but likely would I assume impact response to viruses). While this would not rule out a developmental impact of trisomy for IFNR, it does make such dosage sensitivity less likely or hard to understand and should be mentioned in this context.

4. A comment from my prior review stated: "In transcriptomic analysis of E15.5 heart tissues, why are the differences in DP16/WT comparisons almost the same as in Dp162xIFNRs/WT comparisons? The IFNR-corrected Dp16 mice transcriptomes should be more similar to WT if so much is driven by the IFNR genes, but these results suggest that very similar pathways are dysregulated by other mechanisms independent of trisomy for the IFNR cluster. I understand that the Dp16 versus Dp162xIFNRs comparison shows some difference, but it is surprising and important that this data shows that mice with normalized IFNR gene dosage still have dysregulation of the key pathways that

include: IFN alpha response, IFN gamma response, inflammatory response, IL6 Jak Stat3 signalling, IL2 Jak Stat2 signalling. These pathways seem central to this paper, but this point is not discussed. Is it a discrepancy that warrants substantial softening of the impression that inflammation is driven by or mostly by the IFNR gene cluster, or how do the authors explain it.

I believe in revision the authors have addressed this in the text by stating the disconnect more minimally.... that "not all" of the transcriptomic dysregulated genes are caused by the IFNR pathways, hence naturally other Chr21 genes must contribute. But in their responses they acknowledged this more clearly that "only a small fraction of the transcriptome changes caused by the trisomy were due to IFNR" and that there was a "mild dampening in the dysregulation of these genes observed in Dp162xIfnrs". While I agree these findings can still be consistent with the severe phenotypic effects shown for IFNR, it does seem somewhat surprising.... and different than the human transcriptomics interpreted to be that IFNR genes correlate/cause these pathways. So, some reference to this question (with their possible explanation) might be also mentioned in a limitations paragraph.

Despite these significant questions, I support this study and the novel findings that might prove a breakthrough relevant to human biology in DS. However my big caveat is that this study's big caveats should be made more transparently and clear for the audiences that will be impacted. The title should refer to a DS mouse model. The abstract should not say: "Here we demonstrate that triplication of the interferon receptor gene cluster on chromosome 21 is necessary for multiple hallmarks of DS." DS refers to humans and human Chr21 and DS mouse models and orthologous genes are not "DS" nor Chr21. Even for known single-gene defects in humans, precise mouse mutants don't always reflect the human biology. Here, establishing from a mouse model what Chr21 genes are key in human Down syndrome is much more indirect and complicated.

Author Rebuttal, second revision:

Response to Referees – Manuscript NG-A60025-T by Waugh et al.

Reviewers' Comments:

Remarks to the Author:

Re-review of revised Nat Genet Jan 2023 IFNR genes in DS mouse model.

This was a large study with a lot of dense data, so it took some time to carefully re-read the manuscript and consider the prior reviews and whether responses were adequate, and a qualified colleague independently provided assessments. As requested by the editor, I also considered concerns of Reviewer 1 and responses to them. Both Reviewer 1 and our perspective agree that the most impactful and novel aspect of this study is the creation of a mouse model deleted for the IFNR cluster and the more significant and unexpected findings that this prevented congenital heart defects as well as many cognitive deficits (as measured in mice).

I previously commented that the transcriptomic data is quite inconsistent and rather "muddy" and that correlations do not provide strong evidence that the IFNR gene cluster causes the inflammatory signatures and immune-related pathways. The authors seem to agree that the findings are "muddy" but make the reasonable argument that it might be because the IFNR genes have different effects in different tissues at

different times. This may be the explanation, and the authors have acknowledged these complications more in the manuscript, so they have sufficiently responded to those questions.

I appreciate how much work this study represents, and I continue to feel it should be published in an impactful journal. However, in re-reading the study and various comments and responses, I have misgivings about this manuscript being published without more straightforward statements of the limitations. The phenotypic findings, if true in human Down syndrome, would be so important that the “take home” for this paper (or related news reports) will be read by families of people with DS, clinicians advising patients (about the proposed clinical trials!), biotech and pharmaceutical firms and investors, granting agencies etc. Even researchers in the DS field will not necessarily easily digest all the data and recognized substantial caveats. This concern might be most efficiently and effectively done with a “Limitations paragraph”, and some other edits, that make these caveats and questions clearer.

Regarding Reviewer 1 comments, I surmise that they were not satisfied with the answers to their initial questions and comments that current human findings, and certain mouse studies, do not support that the IFNR triplication/over-expression causes congenital heart defects. I thought these were good questions, and would have preferred clear acknowledgement of such questions and limitations. However, from what I could tell the authors seem to respond by citing a number of studies that very indirectly or loosely are consistent with their thesis about IFNR triplication . . . such as citing that rubella infection causing defects, or other types of IFN-related genes mutated causing effects. Reviewer 1 also cited a study which argued against this effect of IFNR triplication in a different DS mouse model and the authors explained this study may be underpowered (but didn't acknowledge this significant question in the ms). I surmise that Reviewer 1 was not satisfied with these answers and thus in last round just briefly said they wanted more proof. Personally, I didn't feel that knocking-out individual IFNR genes (as Reviewer 1 had suggested) should necessarily be required. . . . but on the other hand, I do think these major questions and caveats should be clearly acknowledged.

1. There should be a clear upfront statement (in limitations paragraph) that this evidence is for one of the trisomic mouse models. . . . and the title and abstract should not conflate “Down syndrome” (implying human) with results in a mouse model. I realize now that the way the study (and abstract) is constructed it begins with correlative human transcriptome evidence that in DS inflammatory signatures are correlated with IFNR over-expression, but this is not persuasive evidence that the immune/inflammatory pathway are caused by IFNR triplication. (IFNR expression would seem logically correlated with inflammatory/immune pathways.) So the mouse study is the key evidence on which they conclude that these findings now justify therapeutic intervention (in humans). . . . even prenatally”

Response: We are once again grateful for the work of the Reviewers and the constructive feedback. Following the Reviewers' guidance, we have addressed comment #1 in the revised manuscript as follows:

First, we have created a ‘limitations of this study’ paragraph in the Discussion where we very clearly acknowledge the caveats raised by the Reviewers. This paragraph reads as follows (sentence relevant for comment #1 highlighted in red):

*This study has several limitations. First, the contribution of *Ifnr* locus dosage to DS phenotypes was defined in a single mouse model of DS, and caution should be exercised when extrapolating these results*

*to the human condition. Second, there are many examples of experimental interventions that reversed phenotypes in mouse models of DS¹⁻⁵, but none of these has been translated yet into an approved therapy for DS. In this regard, the notion that pre-natal interventions targeting IFN signalling could ameliorate CHD or cognitive impairments would need support from additional pre-clinical and clinical research. Third, the phenotypic differences observed upon correction of *Ifnr* locus dosage are accompanied by only partial attenuation of global gene expression changes, and it is not clear which of the pathways affected by *Ifnr* triplication contribute to the rescued phenotypes. Furthermore, these results indicate that many effects of the trisomy are likely to be independent of IFNR dosage, including full induction of inflammatory pathways.*

Second, the title and the abstract have been modified so as to not conflate ‘Down syndrome’ with the results obtained in the mouse model of Down syndrome. The new title and paragraph read as follows (key changes in red):

Title:

Interferon receptor locus contributes to hallmarks of Down syndrome **in a mouse model**

Abstract:

Down syndrome (DS), the genetic condition caused by trisomy 21 (T21), is characterized by cognitive impairment, immune dysregulation, dysmorphogenesis, and increased prevalence of multiple co-occurring conditions. Despite significant efforts, the mechanisms by which T21 causes these effects remain largely unknown. Here we demonstrate that triplication of the interferon receptor (IFNR) gene cluster on chromosome 21 is necessary for multiple phenotypes **in a mouse model of DS**. Whole blood transcriptome analysis demonstrated that IFNR overexpression associates with chronic interferon hyperactivity and inflammation in people with T21. To define the contribution of this locus to DS phenotypes, we used genome editing to correct its copy number **in a mouse model of DS**, which attenuated lethal antiviral responses, prevented heart malformations, ameliorated developmental delays, improved cognition, and normalized craniofacial anomalies. Therefore, *Ifnr* locus dosage modulates major traits of DS **in mice**, suggesting that T21 elicits an interferonopathy **potentially** amenable to therapeutic intervention.

With regards to the comment about Reviewer 1’s questions about the CHD phenotype and its placement in the broader context of the literature, this has been clearly addressed in the Discussion as follows (key segments in red):

*Our findings define a role for the *Ifnr* locus during embryonic heart development, even in the absence of obvious immune triggers. In mice, several regions orthologous to HSA21 were shown to contribute to increased rate of heart malformations, only some of which include the *Ifnr* gene cluster (Extended Data Fig. 7)⁶⁻¹¹. These efforts have not yet identified a single triplicated gene that is sufficient to cause CHD, supporting instead the notion of a polygenic basis for this complex phenotype¹². Although our results do not demonstrate that *Ifnr* triplication is sufficient to cause CHD, they indicate that the *Ifnr* locus contributes to this trait, likely potentiating the*

effects of other necessary genes.

2. The limitations paragraph should somehow convey, as I stated previously, that there have been numerous “cures” of DS mouse models, by various mechanisms, but thus far this has not lead to a significant human therapeutic. It is also the case that results in different mouse models or different studies may disagree. I am hopeful that this particular finding about IFNR cluster as underlying so many main phenotypes will prove a lasting one and be reproduced by others. I agree that this group and some others have strongly shown that there is excess inflammation in people with DS, and it seems plausible that anti-inflammatories could have some modest effect, particularly on respiratory infections etc. So if they are known not to be harmful, this may be justified for testing for modest anti-inflammatory benefits. But this study will be read that there is a strong prospect that this will cure/mitigate so many phenotypes of DS.....it is important and fair to this population that any questions be made clear about whether IFNR triplication causes congenital heart and cognitive impact in humans.....before recommending prenatal treatments. I am concerned that the authors so clearly advocate that this mouse model finding justifies human therapeutics targeting IFNR pathways (JAK-stat etc.).....even when their results show how complicated these pathways are to predict.

Response: We agree with the Reviewers’ call for caution and moderation in the interpretation of our results and their potential therapeutic applications. The new ‘limitations paragraph’ clearly conveys this message (relevant sentence for comment #2 in red):

*This study has several limitations. First, the contribution of *Ifnr* locus dosage to DS phenotypes was defined in a single mouse model of DS, and caution should be exercised when extrapolating these results to the human condition. Second, there are many examples of experimental interventions that reversed phenotypes in mouse models of DS¹⁻⁵, but none of these has been translated yet into an approved therapy for DS. In this regard, the notion that pre-natal interventions targeting IFN signalling could ameliorate CHD or cognitive impairments would need support from additional pre-clinical and clinical research. Third, the phenotypic differences observed upon correction of *Ifnr* locus dosage are accompanied by only partial attenuation of global gene expression changes, and it is not clear which of the pathways affected by *Ifnr* triplication contribute to the rescued phenotypes. Furthermore, these results indicate that many effects of the trisomy are likely to be independent of IFNR dosage, including full induction of inflammatory pathways.*

Additionally, we have softened our language about anti-inflammatory strategies in DS, making amply explicit that additional research will be needed to define the potential value of these therapies. These changes are tracked in the revised manuscript.

3. In re-reading the manuscript, I noticed something related to a question I asked in my first review, regarding whether the monosomic IFNR mouse (used to breed and correct the trisomics) had any phenotype. I thought this interesting and relevant to whether IFNR genes are dosage sensitive. The authors responded that they didn’t study those mice, and I didn’t press this further, but in re-reading I see they don’t even mention this in the manuscript, and, moreover, brief information appears to indicate that

they first made a mouse with NULLISOMY for the IFNR cluster (double-knock out!!). This mouse was then bred to generate the monosomic WT mouse, which they then bred to the Dp16 mice to reduce IFNR copy number. Is this correct.....did they make and breed mice with complete loss of both IFNR cluster alleles? If so, that is remarkable.....and is totally relevant to a paper that is focused on how dosage sensitive is IFNR triplication. If they indeed began with a complete double IFNR knock-out, that should be in their diagram so it is not so hard to decipher in the manuscript. And something should be said that the monosomy or nullisomy for IFNR cluster does not have an obvious impact on development, phenotype or viability (but likely would I assume impact response to viruses). While this would not rule out a developmental impact of trisomy for IFNR, it does make such dosage sensitivity less likely or hard to understand and should be mentioned in this context.

Response: We welcome the opportunity to clarify how the genome editing and subsequent crosses were done to correct *Ifnr* locus dosage in the Dp16 mouse model of DS. The initial CRISPR-mediated genome editing event created founder F0 **heterozygote** *Ifnr*^{-/+} mice, referred to in the paper as WT^{1xIfnr}, which were then crossed to wild type (WT) mice to generate progeny that was either WT (*Ifnr*^{+/+}) or WT^{1xIfnr} (*Ifnr*^{-/+}). We then crossed the heterozygote WT^{1xIfnr} mice to Dp16 mice, leading to progeny that was either WT, WT^{1xIfnr}, Dp16, or Dp16^{2xIfnr}, as shown in Figure 2B. **We did not employ homozygotes *Ifnr*^{-/-} anywhere in this process.** However, homozygotes *Ifnr*^{-/-} (nullisomic) could be generated by crossing heterozygote males and females.

Prompted by the Reviewers' comment, in re-reading the Results and Methods sections, we noticed instances that may have created confusion in this regard, and we have edited accordingly in the revised manuscript for greater clarity. Additionally, in response to the Reviewers' comments, we added commentaries both in the Results and Methods to highlight that the heterozygote WT^{1xIfnr} mice are viable and fertile and without any obvious phenotypes, and that further characterization would be required to identify potential phenotypes associated with *Ifnr* monosomy (or nullisomy).

The edited sections in the Results and Methods read as follows (edits highlighted here in red):

In the Results:

The Ifnr locus contributes to global transcriptome changes in a mouse model of Down syndrome.

... To define if increased dosage of the *Ifnr* locus contributes to DS phenotypes, we used CRISPR/Cas9 genome editing to delete **one copy of the entire gene cluster**. Given that all four *Ifnrs* employ JAK/STAT signalling and that overexpression of each of them associates with inflammatory signatures (**Fig. 1**), creating the potential for genetic redundancy, we designed a strategy to delete **one copy of the 192 kb genomic segment on MMU16 encoding all four *Ifnrs* in wild type (WT) C57BL/6 mice (see **Methods, Fig. 2a, Supplementary Table 4**). **Heterozygous** knockout was confirmed in potential founders by PCR and Sanger sequencing (**Extended Data Fig. 2a-c**). Whole genome sequencing (WGS) confirmed the **heterozygous** deletion and did not reveal any other substantial genomic alterations (**Fig. 2a, Extended Data Fig. 2d**). Heterozygous progeny of this strain (WT^{1xIfnr}) was then intercrossed with Dp16 to correct *Ifnr* copy number from three to two in a portion of Dp16 offspring (Dp16^{2xIfnr}) (**Fig. 2b**). **WT^{1xIfnr} mice were viable and fertile, with no obvious phenotypes, but additional characterization will be valuable to define the impacts of monosomy (or nullisomy) of the *Ifnr* locus.** ...**

In the Methods:

Generation of mice with *heterozygous* deletion of the interferon receptor locus on mouse chromosome 16.

*One copy of the 192 kb genomic segment containing the four interferon receptor (Ifnr) genes orthologous to those on human chromosome 21 (HSA21) was deleted in mice using CRISPR/Cas9 gene editing as previously described for smaller regions of the genome¹³. Briefly, CRISPR/Cas9 target sites were identified using <http://crispr.mit.edu/> with scores of 84-96 to predict specific deletions. Two guide RNAs (gRNAs) were synthesized per target site flanking the Ifnr gene cluster on mouse chromosome 16 (MMU16) using the MEGAshortscriptTM T7 Transcription Kit (LifeTechnologies, Cat# AM1354) and MEGAcleanTM Transcription Clean-Up Kit (Life Technologies, Cat# AM1908) (**Supplementary Table 4, Tab A**). C57BL/6NTac zygotes (Taconic) were microinjected with 25 ng/ μ L Cas9 mRNA (Sigma, Cat# CAS9MRNA-1EA) and four total gRNAs at 7 ng/ μ L each, then implanted into pseudopregnant females. **Heterozygote $WT^{lxIfnrs}$ ($Ifnr^{-/+}$) mutant mice were made in collaboration with Dr. Jennifer Matsuda and James Gross of the Genetics Core Facility at National Jewish Health, CO.***

Sanger sequencing of mutant mice.

*Potential F0s lacking **one copy of the Ifnr gene cluster ($WT^{lxIfnrs}$)** without additional large chromosomal rearrangements on MMU16 were bred to wild type (WT) C57BL/6NTac (Taconic) mice to generate **additional heterozygous F1 progeny ($WT^{lxIfnrs}$)**. PCR products spanning the deleted region were generated and subjected to Sanger sequencing using a 3730xl DNA Analyzer (ThermoFisher Scientific) to identify transmission of a single modified allele to progeny. Sequence-verified F1 $WT^{lxIfnrs}$ mice with identical deletion events were then selected to maintain the line started by a single F0 male $WT^{lxIfnrs}$.*

Animal Husbandry.

*All animal experiments were approved by the Institutional Animal Care and Use Committee (IACUC) at the University of Colorado Anschutz Medical Campus, under Protocol #00111. One candidate F1 male progeny of the validated F0 $WT^{lxIfnrs}$ was backcrossed to WT C57BL/6J (The Jackson Laboratory) for at least 3 generations before female $WT^{lxIfnrs}$ were intercrossed with males from a mouse model of Down syndrome (DS), also of the C57BL/6J background. This $Dp(16Lipi-Zbtb21)1Yey/J$ (hereafter “Dp16”) mouse model of DS¹⁴ was originally purchased from The Jackson Laboratory (Cat# JAX:013530, RRID:IMSR_JAX013530) as well as gifted from Drs. Diana Bianchi and Faycal Guedj (National Institutes of Health, NIH) then intermixed and maintained on the C57BL/6J background. **$WT^{lxIfnrs}$ mice were viable and fertile, with no obvious phenotypes, but further characterization would be valuable to define the impacts of monosomy (or nullisomy) of the Ifnr locus in mice.** After intercrossing female $WT^{lxIfnrs}$ with male Dp16, mice were confirmed to be at least 87.5% C57BL/6J via Transnetyx automated PCR services using single nucleotide polymorphisms from 48 alleles (Cordova, TN) before use in experiments with the remaining mixed background on C57BL/6N (Taconic).*

4. A comment from my prior review stated: “In transcriptomic analysis of E15.5 heart tissues, why are the differences in DP16/WT comparisons almost the same as in Dp162xIFNRs/WT

comparisons? The IFNR-corrected Dp16 mice transcriptomes should be more similar to WT if so much is driven by the IFNR genes, but these results suggest that very similar pathways are dysregulated by other mechanisms independent of trisomy for the IFNR cluster. I understand that the Dp16 versus Dp162xIFNRs comparison shows some difference, but it is surprising and important that this data shows that mice with normalized IFNR gene dosage still have dysregulation of the key pathways that include: IFN alpha response, IFN gamma response, inflammatory response, IL6 Jak Stat3 signalling, IL2 Jak Stat2 signalling. These pathways seem central to this paper, but this point is not discussed. Is it a discrepancy that warrants substantial softening of the impression that inflammation is driven by or mostly by the IFNR gene cluster, or how do the authors explain it.

I believe in revision the authors have addressed this in the text by stating the disconnect more minimally.... that “not all” of the transcriptomic dysregulated genes are caused by the IFNR pathways, hence naturally other Chr21 genes must contribute. But in their responses they acknowledged this more clearly that “only a small fraction of the transcriptome changes caused by the trisomy were due to IFNR” and that there was a “mild dampening in the dysregulation of these genes observed in Dp162xIfnrs”. While I agree these findings can still be consistent with the severe phenotypic effects shown for IFNR, it does seem somewhat surprising..... and different than the human transcriptomics interpreted to be that IFNR genes correlate/cause these pathways. So, some reference to this question (with their possible explanation) might be also mentioned in a limitations paragraph.

Response: Thank you for the opportunity to further revise the text to highlight the fact that the phenotypic rescue observed is accompanied by partial dampening of the transcriptome changes observed. Guided by the Reviewers’ comments, we have revised the text in the Results section and also added a disclaimer in the ‘limitations of this study’ paragraph in the Discussion. The revised sections read as follows (key segments in red):

In the Results section:

Triplication of the Ifnr locus exacerbates immune responses in a mouse model of trisomy 21.

... Notably, key inflammatory signatures are still elevated in Dp16^{2xIfnrs} relative to WT mice, such as IL2/STAT5 Signalling and IL6/JAK/STAT3 signalling (**Fig. 3c**), *indicating that much immune dysregulation occurs without a third copy of the Ifnr locus in this setting, consistent with additional mechanisms driving immune dysregulation in DS, including potential roles for other triplicated genes...*

Reduction of Ifnr locus dosage rescues heart malformations in a mouse model of Down syndrome.

...In Dp16^{2xIfnrs} mice, many, but certainly not all, of these gene expression changes are *attenuated*, with some variation across time points. For example, at E12.5, a critical time point for septation of the heart, Dp16^{2xIfnrs} mice show lesser dysregulation of ISGs (e.g., *Irf9*, *Ifih1*), EMT genes (e.g., *Col4a2*, *Tgfb1*), and MYC target genes (e.g., *Rcf4*, *Srsf7*) (**Fig. 4f-g**). *Thus, the decreased incidence of CHD observed in Dp16^{2xIfnrs} mice is accompanied by modest effects on*

the concurrent global transcriptome changes, whereby only a small fraction of the gene expression changes observed are due to Ifnr triplication...

Triplication of the Ifnr locus delays development and impairs cognition.

...As observed in other tissues, the effects of correcting Ifnr locus dosage on gene expression changes are partial and selective for specific signalling pathways, once again revealing that phenotypic differences can be observed without full correction of underlying transcriptome changes.

Reduction of Ifnr locus copy number attenuates craniofacial anomalies.

...These results indicate that cell proliferation in this embryonic tissue is negatively impacted by an extra copy of the Ifnr locus. As for other traits sensitive to dosage of the Ifnr locus, Dp16^{2xIfnrs} mice show amelioration of craniofacial phenotypes even with only mild dampening of dysregulated gene expression patterns...

In the Discussion, first acknowledgement of this caveat and alternative mechanisms:

Importantly, correction of Ifnr locus copy number does not fully rescue transcriptome signatures of inflammation and immune dysregulation, many of which are still observed in tissues of Dp16^{2xIfnrs} mice (e.g., IL2 signalling in lymph nodes, IFN signalling in the developing heart). This indicates the presence of additional mechanisms contributing to immune dysregulation in DS. For example, cellular stress from extra DNA, even from a Roberstonian translocation¹⁵, and resulting activation of Damage Associated Molecular Patterns (DAMPs)^{16,17} could still be activating IFN signalling in the absence of increased Ifnr gene dosage by elevating IFN ligands and/or alternative mechanisms.

Later in the Discussion, second acknowledgement of this caveat and alternative mechanisms:

Correction of Ifnr locus copy number did not fully rescue the cognitive impairments characteristic of Dp16, nor the global gene expression changes observed, indicating that other triplicated genes could also contribute to these phenotypes. For example, we observed that Dyrkl1a, a gene with documented roles in brain development and function in DS and other genetic disorders¹⁸, is overexpressed in Dp16^{2xIfnrs} tissues, including the brain, where it could exert additional effects independent of Ifnr copy number.

Further down in the Discussion, within the new ‘limitations of this study paragraph’ (relevant sentence for comment #4 in red):

This study has several limitations. First, the contribution of Ifnr locus dosage to DS phenotypes was defined in a single mouse model of DS, and caution should be exercised when extrapolating these results to the human condition. Second, there are many examples of experimental interventions that reversed phenotypes in mouse models of DS¹⁻⁵, but none of these has been translated yet into an approved therapy for DS. In this regard, the notion that pre-natal interventions targeting IFN signalling could ameliorate CHD or cognitive impairments would need support from additional pre-clinical and clinical research. Third, the phenotypic differences observed upon correction of Ifnr locus dosage are accompanied by only partial attenuation of global gene expression changes, and it is not clear which of the pathways affected by Ifnr triplication contribute to the rescued phenotypes. Furthermore, these results indicate that many

effects of the trisomy are likely to be independent of IFNR dosage, including full induction of inflammatory pathways.

Despite these significant questions, I support this study and the novel findings that might prove a breakthrough relevant to human biology in DS. However my big caveat is that this study's big caveats should be made more transparently and clear for the audiences that will be impacted. The title should refer to a DS mouse model. The abstract should not say: "Here we demonstrate that triplication of the interferon receptor gene cluster on chromosome 21 is necessary for multiple hallmarks of DS." DS refers to humans and human Chr21 and DS mouse models and orthologous genes are not "DS" nor Chr21. Even for known single-gene defects in humans, precise mouse mutants don't always reflect the human biology. Here, establishing from a mouse model what Chr21 genes are key in human Down syndrome is much more indirect and complicated.

Response: We are grateful for the additional guidance. Following these Reviewers' comments, we have modified the title and abstract as follows:

Title:

Interferon receptor locus contributes to hallmarks of Down syndrome **in a mouse model**

Abstract:

Down syndrome (DS), the genetic condition caused by trisomy 21 (T21), is characterized by cognitive impairment, immune dysregulation, dysmorphogenesis, and increased prevalence of multiple co-occurring conditions. Despite significant efforts, the mechanisms by which T21 causes these effects remain largely unknown. Here we demonstrate that triplication of the interferon receptor (IFNR) gene cluster on chromosome 21 is necessary for multiple phenotypes **in a mouse model of DS**. Whole blood transcriptome analysis demonstrated that IFNR overexpression associates with chronic interferon hyperactivity and inflammation in people with T21. To define the contribution of this locus to DS phenotypes, we used genome editing to correct its copy number **in a mouse model of DS**, which attenuated lethal antiviral responses, prevented heart malformations, ameliorated developmental delays, improved cognition, and normalized craniofacial anomalies. Therefore, *Ifnr* locus dosage modulates major traits of DS **in mice**, suggesting that T21 elicits an interferonopathy **potentially** amenable to therapeutic intervention.

References.

- 1 Pinto, B. *et al.* Rescuing Over-activated Microglia Restores Cognitive Performance in Juvenile Animals of the Dp(16) Mouse Model of Down Syndrome. *Neuron* **108**, 887-904 e812 (2020). <https://doi.org:10.1016/j.neuron.2020.09.010>
- 2 Guedj, F. *et al.* Apigenin as a Candidate Prenatal Treatment for Trisomy 21: Effects in Human Amniocytes and the Ts1Cje Mouse Model. *Am J Hum Genet* **107**, 911-931 (2020). <https://doi.org:10.1016/j.ajhg.2020.10.001>

- 3 Hunter, C. L., Bachman, D. & Granholm, A. C. Minocycline prevents cholinergic loss in
a mouse model of Down's syndrome. *Ann Neurol* **56**, 675-688 (2004).
<https://doi.org/10.1002/ana.20250>
- 4 de la Torre, R. & Dierssen, M. Therapeutic approaches in the improvement of cognitive
performance in Down syndrome: past, present, and future. *Prog Brain Res* **197**, 1-14
(2012). <https://doi.org/10.1016/B978-0-444-54299-1.00001-7>
- 5 Rueda, N., Florez, J., Dierssen, M. & Martinez-Cue, C. Translational validity and
implications of pharmacotherapies in preclinical models of Down syndrome. *Prog Brain
Res* **251**, 245-268 (2020). <https://doi.org/10.1016/bs.pbr.2019.10.001>
- 6 Antonarakis, S. E. *et al.* Down syndrome. *Nature reviews. Disease primers* **6**, 9 (2020).
<https://doi.org/10.1038/s41572-019-0143-7>
- 7 Li, Z. *et al.* Duplication of the entire 22.9 Mb human chromosome 21 syntenic region on
mouse chromosome 16 causes cardiovascular and gastrointestinal abnormalities. *Hum
Mol Genet* **16**, 1359-1366 (2007). <https://doi.org/10.1093/hmg/ddm086>
- 8 Liu, C. *et al.* Genetic analysis of Down syndrome-associated heart defects in mice. *Hum
Genet* **130**, 623-632 (2011). <https://doi.org/10.1007/s00439-011-0980-2>
- 9 Liu, C. *et al.* Engineered chromosome-based genetic mapping establishes a 3.7 Mb
critical genomic region for Down syndrome-associated heart defects in mice. *Hum Genet*
133, 743-753 (2014). <https://doi.org/10.1007/s00439-013-1407-z>
- 10 Lana-Elola, E. *et al.* Genetic dissection of Down syndrome-associated congenital heart
defects using a new mouse mapping panel. *Elife* **5** (2016).
<https://doi.org/10.7554/eLife.11614>
- 11 Zhang, H., Liu, L. & Tian, J. Molecular mechanisms of congenital heart disease in down
syndrome. *Genes Dis* **6**, 372-377 (2019). <https://doi.org/10.1016/j.gendis.2019.06.007>
- 12 Li, H. *et al.* Penetrance of Congenital Heart Disease in a Mouse Model of Down
Syndrome Depends on a Trisomic Potentiator of a Disomic Modifier. *Genetics* **203**, 763-
770 (2016). <https://doi.org/10.1534/genetics.116.188045>
- 13 Boroviak, K., Doe, B., Banerjee, R., Yang, F. & Bradley, A. Chromosome engineering in
zygotes with CRISPR/Cas9. *Genesis* **54**, 78-85 (2016). <https://doi.org/10.1002/dvg.22915>
- 14 Li, Z. *et al.* Duplication of the entire 22.9 Mb human chromosome 21 syntenic region on
mouse chromosome 16 causes cardiovascular and gastrointestinal abnormalities. *Human
molecular genetics* **16**, 1359-1366 (2007).
- 15 Williams, B. R. *et al.* Aneuploidy affects proliferation and spontaneous immortalization
in mammalian cells. *Science* **322**, 703-709 (2008).
<https://doi.org/10.1126/science.1160058>
- 16 Krivega, M. *et al.* Genotoxic stress in constitutive trisomies induces autophagy and the
innate immune response via the cGAS-STING pathway. *Commun Biol* **4**, 831 (2021).
<https://doi.org/10.1038/s42003-021-02278-9>
- 17 Pradeu, T. & Cooper, E. L. The danger theory: 20 years later. *Front Immunol* **3**, 287
(2012). <https://doi.org/10.3389/fimmu.2012.00287>

- 18 Duchon, A. & Herault, Y. DYRK1A, a Dosage-Sensitive Gene Involved in Neurodevelopmental Disorders, Is a Target for Drug Development in Down Syndrome. *Front Behav Neurosci* **10**, 104 (2016). <https://doi.org:10.3389/fnbeh.2016.00104>

Decision Letter, third revision:

Our ref: NG-A60025R2

8th Mar 2023

Dear Dr. Espinosa,

Thank you for submitting your revised manuscript "Interferon receptor locus contributes to hallmarks of Down syndrome in a mouse model" (NG-A60025R2). It has now been seen by the original referees and their comments are below. The reviewers find that the paper has improved in revision, and therefore we'll be happy in principle to publish it in Nature Genetics, pending minor revisions to satisfy the referees' final requests and to comply with our editorial and formatting guidelines.

Sincerely,

Safia Danovi
Editor
Nature Genetics

Final Decision Letter: